



# Estimates of the remote sensing retrieval errors by GRASP algorithm: application to ground-based observations, concept and validation

Milagros E. Herrera[1], Oleg Dubovik[1], Benjamin Torres[1], Tatyana Lapyonok[1], David Fuertes[2], Anton Lopatin[2], Pavel Litvinov[2], Cheng Chen[2], Jose Antonio Benavent-Oltra[3], Juan L. Bali[4], and Pablo R. Ristori[5]

[1]Laboratoire d'Optique Atmosphérique, CNRS/Université de Lille, Villeneuve d'Ascq, France
[2]GRASP-SAS, Remote Sensing Developments, Université de Lille, Villeneuve d'Ascq, 59655, France
[3]Department of Electrical, Electronical and Automatic Control Engineering and Applied Physics, Escuela Técnica Superior de Ingeniería y Diseño Industrial, Universidad Politécnica de Madrid, 28012 Madrid, Spain
[4]National Scientific and Technical Research Council (CONICET), Buenos Aires, Argentina
[5]CEILAP-UNIDEF (MINDEF - CONICET) - CITEDEF, Buenos Aires, Argentina

**Correspondence:** Milagros E. Herrera (milagros.herrera@univ-lille.fr)

**Abstract.**

The understanding of the uncertainties in the retrieval of the aerosol and surface properties is very important for an adequate characterization of the processes that occur in the atmosphere. However, the reliable characterization of the error budget of the retrieval products is a very challenging aspect that currently remains not fully resolved in most remote sensing approaches. The level of uncertainties for the majority of the remote sensing products relies mostly on post-processing validations and inter comparisons with other data while the dynamic errors are rarely provided. Therefore, implementations of fundamental approaches for generating dynamic retrieval errors and the evaluation of their practical efficiency remains of high importance.

This study describes and analyses the dynamic estimates of uncertainties of aerosol retrieved properties by GRASP (Generalized Retrieval of Atmosphere and Surface Properties) algorithm. GRASP inversion algorithm described by Dubovik et al. (2011, 2014, 2021) is designed on the concept of statistical optimization and provides dynamic error estimates for all retrieved aerosol and surface properties. The approach takes into account the effect of both random and systematic uncertainties propagations. The algorithm provides error estimates both for directly retrieved parameters included in the retrieval state vector and for the characteristics derived from these parameters. For example, in the case of the aerosol properties, GRASP retrieves directly the size distribution and the refractive index that are used afterwards to provide phase function, scattering, extinction, single scattering albedo, etc. Moreover, GRASP algorithm provides full covariance matrices, i.e. not only variances of the retrieval errors and also correlations coefficients of these errors. The analysis of correlation matrix structure can be very useful for identifying unobvious retrieval tendencies that appear to be a useful approach for optimizing observation schemes and retrieval setups.

In this study we analyse the efficiency of GRASP error estimation approach for applications to ground-based observations by sun/sky photometer and lidar. Specifically, diverse aspects of the errors generations and their evaluations are discussed and





illustrated. The studies rely on the series of comprehensive sensitivity tests when simulated sun/sky photometer measurements and lidar data are perturbed by random and systematic errors and inverted. Then, the results of the retrievals and their error estimations are analyzed and evaluated. The tests are conducted for different observations of diverse aerosol types including biomass burning, urban, dust and their mixtures. The study considers observations of AERONET sun/sky photometer measure-

ments at $440$, $675$, $870$ and $1020\,nm$ and multi-wavelength elastic lidar measurements at $355$, $532$ and $1064\,nm$. The sun/sky photometer data are inverted alone or together with lidar data.

The analysis shows overall successful error retrievals for different aerosol characteristics including aerosol size distribution, complex refractive index, single scattering albedo, lidar ratios, aerosol vertical profiles, etc. Also, the main observed tendencies in the error dynamic agree with known retrieval experience. For example, the main accuracy limitations for retrievals of all

aerosol types relate to the situations with low optical depth. Also, in situations with multi-component aerosol mixtures, the reliable characterization of each component is possible only in limited situations, for example from radiometric data obtained for low solar zenith angle observations or from a combination of radiometric and lidar data. At the same time, total optical properties of aerosol mixtures are always retrieved satisfactorily.

In addition, the study includes the analysis of the detailed structure of correlation matrices for the retrieval errors of mono-

and multi- component aerosols. The conducted analysis of error correlation appears to be a useful approach for optimizing observations schemes and retrieval setups. The application of the approach to real data is provided.

# 1   Introduction

Remote sensing is one major tool for monitoring atmosphere and surface properties at large scales. These observations have non-destructive character and allow for dynamic local, regional or global monitoring of the ambient atmosphere. Correspond-

ingly, diverse remote sensing observations are employed for routine observations and characterization of the Earth atmosphere. One of the key challenges in implementing remote sensing is the development of the retrieval algorithms. While remote sensing retrievals have substantially evolved during the last decades, a significant need for further advancing various aspects of the retrieval algorithms remains. One of the most challenging and important, while underdeveloped, aspects is the evaluation of the errors in the retrieval products. For example, the review by Sayer et al. (2020) emphases that for most aerosol satellite

retrievals, the quality of the retrieval uncertainty estimates has not been routinely assessed.

Here we discuss and analyze the approach implemented in GRASP (Generalized Retrieval of Atmosphere and Surface Properties) retrieval algorithm. GRASP concept is based on the statistically optimization fitting designed for retrieval of detailed aerosol properties from diverse observations ((Dubovik et al., 2011, 2014, 2021)). This algorithm uses statistical estimates of random error propagation and provides the dynamic error estimates for both retrieved parameters (such as size distribution,

refractive index, etc.) and characteristics derived from those parameters (as total scattering, extinction, single scattering albedo, etc). Specifically, in this work, we discuss and analyze the error estimates of the aerosol properties by GRASP for aerosol retrievals from ground-based observations by sun/sky radiometers and lidars.



One of the most visible data sets of ground-based radiometric observations is provided by AERONET (AErosol RObotic NETwork, Holben et al. (1998)) network of more than 500 operational sites distributed over the world. AERONET network
provides column integrated aerosol properties of the different sites distributed over the world. AERONET provides column integrated aerosol properties originally provided by Nakajima et al. (1996) algorithm and later by Dubovik and King (2000) retrieval. Several studies accessing the AERONET retrieval errors. First, Dubovik et al. (2000) provided a rather comprehensive analysis of retrieval uncertainties caused by both random measurements errors and systematic errors originated from potential biases in the measurements and imperfections in the modeling aerosol properties. This analysis was revisited by Torres et al.
(2017) studies that overall confirmed most of the uncertainty tendencies revealed by Dubovik et al. (2000). Recently, Sinyuk et al. (2020) published a concept for aerosol retrieval error estimates that have been adapted in Version 3 aerosol operational product of AERONET (Giles et al., 2019). In frame of this concept the uncertainties are estimated using the spread of retrieved parameters generated by 27 distinct combinations of retrieval implemented with perturbed the input data (AOD, sky radiances, solar spectral irradiance and surface reflectance). Somewhat similar concept for the estimating error was earlier
employed in the LiRIC (Lidar and Radiometer Inversion Code) approach for the synergy processing of co-located lidar and AERONET sun/sky photometer observations (Chaikovsky et al., 2016). LiRIC provided some uncertainty simulated using a series of retrievals with the perturbed input data. A large number of factors affect the retrievals, whose variation is complex and non-linear. Indeed, modeling all the factors and circumstances that can affect the retrieval in all situations is theoretically impossible, and practically challenging, within the limited series of perturbed runs especially for the situations when a large
number of parameters is retrieved. In these regards, the error propagations approaches based on statistical estimation theory and described in numerous textbooks (e.g., Edie et al., 1971; Fourgeaud and Fuchs, 1967; Rodgers, 2000) provide asymptotically comprehensive estimates for random retrieval errors. At the same time, it should be noted that both the result of perturbation tests and statistical estimates of propagated error rely on the forward model employed may not fully represent inaccuracies related to the limitations of this model. Some additional evaluations and considerations are always desirable for accessing the
adequacy of chosen forward model and its potential limitations.

The GRASP is a state-of-art highly versatile inversion algorithm of new generation that can be applied to a variety of remote sensing and laboratory observations. For example, GRASP has been applied for several satellite instruments (Dubovik et al., 2019, 2021; Li et al., 2019; Chen et al., 2018, 2019, 2020; Puthukkudy et al., 2020) and demonstrated the capability to provide accurate information about detailed properties of aerosol and underlying surface reflectance. The GRASP has been successfully
used for retrieving aerosol properties from polar nephelometer data (Espinosa et al., 2017, 2019; Schuster et al., 2019). The applications of GRASP algorithm for retrieving aerosol properties from photometer sun/sky radiometer observations alone or in combination with lidar observations have been employed and discussed in numerous studies (e.g., Torres et al., 2017), synergy of sun/sky photometer plus lidar (Lopatin et al., 2013, 2021; Torres et al., 2017; Benavent-Oltra et al., 2017, 2019, 2021; Hu et al., 2019; Tsekeri et al., 2017; Román et al., 2018; Herreras et al., 2019; Titos et al., 2019). The successful application
of GRASP has been illustrated for the interpretation of terrestrial observations with a celestial camera (Román et al., 2017). Recent studies by Torres et al. (2017) and Torres and Fuertes (2021) demonstrated the high potential of GRASP retrieval concept for inverting only direct-Sun photometric observations.





The evaluation of the retrieval accuracy in all those studies was realized by comparing and validating the retrieval results with independent reliable data. It should be noted however that practically none of these studies discuss the retrieval error estimates while the formalism of error estimate is realized in GRASP since a while. As a result, the validity of the retrieval error estimates provided by GRASP approach remains unattended and unverified. Therefore, in order to address this issue, the current work proposes a discussion of the main aspects of GRASP error generation and attempts to provide evaluation of the retrieval error estimates provided by GRASP. The study is focused on the considerations of aerosol retrieval from sun/sky photometers and lidar ground-based observations.

## 2  Modeling of error estimates in the GRASP algorithm

As mentioned in the introduction, in this work we make use of the GRASP (Generalized Retrieval of Atmosphere and Surface Properties) algorithm. It is a rigorous, versatile and open-source algorithm capable of providing information of the aerosol properties from measurements of different instruments and dynamic error estimates (Dubovik et al., 2011, 2014, 2021). It is a flexible, generalized algorithm that relies on two independent modules: the forward model and the numerical inversion. The forward contains the full description of the physical model including various interactions of electromagnetic solar radiations, such as aerosol scattering, surface reflectance and gaseous absorption. The multiple scattering interactions in the atmosphere are accounted by solving the vector of radiative transfer equation. Thus, GRASP forward model is capable of simulating diverse measurements in laboratory and atmosphere remote sensing including passive and active observation from space and ground. On the other hand, numerical inversion is not directly related to any physical problem and realizes formal inversion of the measurements using statistical estimation approach. Specifically, GRASP employs the Multi-term Least Square Method (LSM) that allows for a flexible utilization of multiple a priori constraints. This approach is very convenient for designing diverse remote sensing retrievals as discussed in details by Dubovik et al. (2021).

The retrieval error estimates in GRASP are calculated by modeling propagation of measurement errors based on statistical estimation approach. In addition, the formulation used for estimating errors account for some contribution of systematic errors that could be originated from biases in the measurement or some modifications implemented in the algorithm for improving retrieval convergence of non-linear solutions. Below the description of the overall concept and specific key implementations of the errors estimation in GRASP are provided.

### 2.1  The numerical inversion based on statistical optimization concept

The Multi-term LSM employed in GRASP searches for the solution using statistically optimized fitting under multiple a priori constraints (Dubovik, 2004; Dubovik et al., 2011, 2021). It considers both measurements and a priori data in similar manner considering them as a data from different and independent data sources i.e.:

$$\mathbf{f}_k^* = \mathbf{f}_k(\mathbf{a}) + \Delta \mathbf{f}_k^* \tag{1}$$





where $k$ denotes different data sets, that are not correlated and may have different levels of uncertainties described by different covariance matrices $\mathbf{C}_k$. Such explicit differentiation of the input data makes the retrieval more transparent because it clearly identifies the different used data sets. Correspondingly joint probability density function (PDF) of independent data sets $\mathbf{f}_1^*, \mathbf{f}_2^*, \cdot, \mathbf{f}_K^*$ can be obtained by the simple multiplication of the PDFs of data from all $K$ sources:

$$\mathbf{P}(\mathbf{f}(\mathbf{a})|\mathbf{f}^*) = \mathbf{P}(\mathbf{f}_1(\mathbf{a}), \ldots, \mathbf{f}_k(\mathbf{a})|\mathbf{f}_1^*, \ldots, \mathbf{f}_k^*) = \prod_{k=1}^{K} \mathbf{P}(\mathbf{f}_k(\mathbf{a})|\mathbf{f}_k^*) \tag{2}$$

It can be noted that Eq. (1) not assume any relations between forward models $\mathbf{f}_k(\mathbf{a})$, i.e. forward models $\mathbf{f}_k(\mathbf{a})$ can be the same or different. In the frame of LSM approach, i.e. under the assumptions of a normal PDF of the error $\Delta \mathbf{f}_k^*$, the solution of the Eq. (1) corresponds to the minimum of the following functional:

$$2\hat{\Psi}(\mathbf{a}) = \frac{1}{2} \sum_{k=1}^{K} (\mathbf{f}_k(\mathbf{a}) - \mathbf{f}_k^*)^T \mathbf{C}_k^{-1} (\mathbf{f}_k(\mathbf{a}) - \mathbf{f}_k^*) = \min \tag{3}$$

For the general case, of non-linear functions $\mathbf{f}_k(\mathbf{a})$ the solution of Eq. (3) is sought iteratively:

$$\mathbf{a}^{p+1} = \mathbf{a}^p - \Delta \mathbf{a}^p \tag{4}$$

where $\Delta \mathbf{a}^p$ is the solution that can be found by solving the system of so-called normal equations, i.e.:

$$\left( \sum_{k=1}^{K} \mathbf{K}_{k,p}^T (\mathbf{C}_k)^{-1} \mathbf{K}_{k,p} \right) \Delta \mathbf{a}^p = \sum_{k=1}^{K} \mathbf{K}_{k,p}^T (\mathbf{C}_k)^{-1} \Delta \mathbf{f}_k^p \tag{5}$$

where $\Delta \mathbf{f}^p = \mathbf{f}(\mathbf{a}^p) - \mathbf{f}^*$, and $\mathbf{K}_p$ is Jacobean matrix at p-th iteration of the functions $\mathbf{f}_k(\mathbf{a})$ in the vicinity of $\mathbf{a}^p$ with the elements $\{\mathbf{K}_{k,p}\}_{j,i} = \frac{\partial \mathbf{f}_{k,j}(\mathbf{a})}{\partial \mathbf{a}_i} \Big|_{\mathbf{a}=\mathbf{a}^p}$.

The asymptotic limit of the minimized quadratic form, for most applications, can be written as:

$$2\Psi(\mathbf{a}) = min \rightarrow \sum_{k=1}^{K} N_k - n \tag{6}$$

It should be noted that the LSM solution defined by Eq. (3) corresponds to the minimum of quadratic form $\hat{\Psi}(\mathbf{a})$ and does not depend in any way from the value of this minimum. Considering this fact, in practical application is often convenient to renormalize the minimized quadratic $\hat{\Psi}(\mathbf{a})$, in situations when only one data set is inverted it is convenient to weighting matrix $\mathbf{W} = \mathbf{C}/\varepsilon_1^2$ and minimize the quadratic form $\Psi'(\mathbf{a}) = \varepsilon_1^2 \Psi(\mathbf{a})$. In such approach one does not need to know the exact value of $\varepsilon_1^2$. Moreover, $\varepsilon_1^2$ can be estimated from asymptotic LSM expectations provided by Eq. (6).





In frame of Multi-term approach the use of weighting matrices additionally allows for making more explicit the contribution of different data sources. Indeed, using the weighting matrices $\mathbf{W}_k$ instead of covariance matrices $\mathbf{C}_k$ the Eq. (5) can be written as:

$$\left(\sum_{k=1}^{K}\gamma_k\mathbf{K}_k^T(\mathbf{W}_k)^{-1}\mathbf{K}_k\right)\Delta\mathbf{a}^p = \sum_{k=1}^{K}\gamma_k\mathbf{K}_k^T(\mathbf{W}_k)^{-1}\Delta\mathbf{f}_k^p \tag{7}$$

    In this formulation the relative contribution of the data from different data sources are scaled by the corresponding Lagrange

parameters $\gamma_i$, defined as:

$$\mathbf{W}_i = \frac{1}{\varepsilon_i^2}\mathbf{C}_i \quad\text{and}\quad \gamma_i = \frac{\varepsilon_1^2}{\varepsilon_i^2} \tag{8}$$

    where $\varepsilon_i^2$ is the first diagonal element of $\mathbf{C}_i$, i.e. $\varepsilon_i^2 = \mathbf{C}_{i\,11}$ and $\gamma_i$ is the ratio of the variances of scattered radiances and variances of the corresponding data set. Evidently, that $\gamma_1 = 1$ as discussed by Dubovik and King (2000); Dubovik (2004); Dubovik et al. (2011), etc. This renormalization strategy is especially convenient on Multi-term LSM approach once some

of data sets correspond to a priori information. In addition, the renormalized definition of the minimized quadratic function (or 'residual') as $\Psi'(\mathbf{a}) = \varepsilon_1^2\Psi(\mathbf{a})$, the measurement error $\varepsilon_1^2$ can be estimated from the residual of the fit. Indeed, once the weighting matrices used in the solution, Eq. (7) minimizes quadratic with the limit depending on $\varepsilon_1^2$:

$$2\Psi'(\mathbf{a}) = 2\varepsilon_1^2\Psi(\mathbf{a}) = min \rightarrow \varepsilon_1^2\left(\sum_{k=1}^{K}N_k - n\right) \quad\text{and}\quad \hat{\varepsilon}_1^2 \approx \frac{2\Psi'(\mathbf{a}^p)}{\sum_{k=1,\dots,K}(N_{\mathbf{f}_i}) - N_\mathrm{a}} \tag{9}$$

### 2.1.1   A priori constraints in Multi-term LSM approach and in GRASP algorithm

As discussed in detail by Dubovik et al. (2021) the Multi-term LSM concept has been proposed as methodologically convenient approach for integrating different types of a priori constraints in remote sensing applications (Dubovik, 2004; Dubovik and King, 2000; Dubovik et al., 1995, 2000, 2008, 2011). In the Multi-term LSM a priori estimates are considered to be 'equivalent' to the measurements, i.e. characterized by their PDF and treated equivalently to the actual measurements. In these regards, Eqs. (1-7) do not show any distinction between different $\mathbf{f}_k(\mathbf{a})$.

At the same time, in practice there are always two different types of data sets: measurements and a priori constraint on the unknowns $\mathbf{a}$. Therefore, the vector of the measurement $(\mathbf{f}^*)^T = (\mathbf{f}_1^*, \mathbf{f}_2^*, \cdot, \mathbf{f}_k^*)^T$ can be written as:

$$(\mathbf{f}^*)^T = (\mathbf{f}_1^*, \mathbf{f}_2^*, \cdot, \mathbf{f}_k^*, \mathbf{f}_1^\mathrm{a}, \mathbf{f}_2^\mathrm{a}, \cdot, \mathbf{f}_k^\mathrm{a})^T \tag{10}$$

    where $\mathbf{f}_i^* = \mathbf{f}_i^*(\mathbf{a})$ represent directly measured characteristics and $\mathbf{f}_i^\mathrm{a} = \mathbf{f}_i^\mathrm{a}(\mathbf{a})$ represent a priori known characteristics of unknowns $\mathbf{a}$. Correspondingly the right side of Eq. (2) can be formally split in two groups:





$$P(\mathbf{f}(\mathbf{a})|\mathbf{f}^*) = \prod_{k=1}^{K} P(\mathbf{f}_k(\mathbf{a})|\mathbf{f}_k^*) \prod_{n=1}^{N} P(\mathbf{f}_n^{\mathrm{a}}(\mathbf{a})|\mathbf{f}_n^{\mathrm{a},*}) \tag{11}$$

Therefore, the Eq. (7) can also be formally arranged to identify the contribution of measurements and a priori terms:

$$\left( \sum_{k=1}^{K} \gamma_k \mathbf{K}_{k,p}^T (\mathbf{W}_k)^{-1} \mathbf{K}_{k,p} + \sum_{n=1}^{N} \gamma_{\mathrm{a},n} \mathbf{K}_{\mathrm{a},n,p}^T (\mathbf{W}_{\mathrm{a},n})^{-1} \mathbf{K}_{\mathrm{a},n,p} \right) \Delta \mathbf{a}^p =$$
$$= \sum_{k=1}^{K} \gamma_k \mathbf{K}_{k,p}^T (\mathbf{W}_k)^{-1} \Delta \mathbf{f}_k^{*,p} + \sum_{n=1}^{N} \gamma_{\mathrm{a},n} \mathbf{K}_{\mathrm{a},n,p}^T (\mathbf{W}_{\mathrm{a},n})^{-1} \Delta \mathbf{f}_n^{\mathrm{a}^*,p} \tag{12}$$

where two groups of the terms in left and right parts of the equation represent the contributions of the set of $K$ measured characteristics $\mathbf{f}_k(\mathbf{a})$ and the set of $N$ a priori $\mathbf{f}_n^{\mathrm{a}}(\mathbf{a})$ characteristics, and the Lagrange parameters are defined as:

$$\gamma_k = \frac{\varepsilon_{k=1}^2}{\varepsilon_k^2} \quad \text{and} \gamma_{\mathrm{a},n} = \frac{\varepsilon_{n=1}^2}{\varepsilon_{\mathrm{a},n}^2} \tag{13}$$

As discussed by Dubovik (2004) and Dubovik et al. (2021) the Multi-term approach is a simple rearranging the base LSM formulation, while the resulting Eq. (7) provides a solid basis for unifying many known formulas of constrained inversion in a single formalism and practically convenient and efficient for developing remote sensing algorithms using diverse complimentary observations and a priori constrains.

While, the Multi-term LSM concept allows flexible utilizations of nearly arbitrary a priori constraints, GRASP algorithm is fully adapted for using the most popular and physically transparent a priori constraints such as direct a priori estimates of unknowns $\mathbf{a}$ and, smoothness constraints in situations, when the unknown vector $\mathbf{a}$ or any group of unknowns included in this vector, represent continuous smooth function. For example, if vector $\mathbf{a}$ represents aerosol size distribution, that is known to be rather smooth, the system given by Eq. (1) can be explicitly written as follows:

$$\begin{cases} \mathbf{f}_{k=1}^* = \mathbf{f}_{k=1}^*(\mathbf{a}) + \Delta \mathbf{f}_{k=1}^* \\ \mathbf{f}_{n=1}^{\mathrm{a},*} = \mathbf{f}_{n=1}^{\mathrm{a},*}(\mathbf{a}) + \Delta \mathbf{f}_{n=1}^{\mathrm{a},*} \\ \mathbf{f}_{n=2}^{\mathrm{a},*} = \mathbf{f}_{n=2}^{\mathrm{a},*}(\mathbf{a}) + \Delta \mathbf{f}_{n=2}^{\mathrm{a},*} = \end{cases} \begin{cases} \mathbf{f}_1^* = \mathbf{f}_{k=1}^*(\mathbf{a}) + \Delta \mathbf{f}_1^* \\ \mathbf{f}_1^{\mathrm{a},*} = \mathbf{f}_1^{\mathrm{a},*}(\mathbf{a}) + \Delta \mathbf{f}_1^{\mathrm{a},*} \\ \mathbf{f}_2^{\mathrm{a},*} = \mathbf{f}_2^{\mathrm{a},*}(\mathbf{a}) + \Delta \mathbf{f}_2^{\mathrm{a},*} = \end{cases} \begin{cases} \mathbf{f}_1^* = \mathbf{f}_{k=1}^*(\mathbf{a}) + \Delta \mathbf{f}_1^* \\ \mathbf{a}^* = \mathbf{a} + \Delta_{\mathbf{a}^*} \\ \mathbf{0}^* = \mathbf{G}_m^{\mathbf{a}} + \Delta_{\mathbf{g}^*} \end{cases} \tag{14}$$

The a priori constraints defined by the second line $\mathbf{a}^* = \mathbf{a} + \Delta_{\mathbf{a}^*}$ represents the most common of constraints of solution by direct a priori estimates of unknowns $\mathbf{a}^*$, where $\Delta_{\mathbf{a}^*}$ are the uncertainties of the estimates $\mathbf{a}^*$ and are generally considered to be unbiased random errors within the covariance matrix $\mathbf{C}_{\mathbf{a}^*}$. These constraints can be easily included in Eq. 12 by defining: $\mathbf{K}_{\mathrm{a}} = \mathbf{1}$ - unity matrix; i.e. $\mathbf{K}_{\mathrm{a}}^T \mathbf{W}_{\mathrm{a}}^{-1} \mathbf{K}_{\mathrm{a}} = \mathbf{W}_{\mathrm{a}}^{-1}$ and $\mathbf{K}_{\mathrm{a}}^T \mathbf{W}_{\mathrm{a}}^{-1} \mathbf{f}_1^{\mathrm{a},*} = \mathbf{W}_{\mathrm{a}}^{-1} \mathbf{a}$. Utilization of a priori estimates $\mathbf{a}^*$ was introduced





in the pioneering studies by Twomey (1963) and later evolved and discussed in detail in the Rodgers (2000) textbook on inversion. The third line represents another common type of a priori constraint known as smoothness constraints that limit the variability of retrieved functions by using a priori knowledge about limitations on derivatives of those functions. For example, a priori knowledge limits high frequency variations of continuous functions $v(x)$ , such as the aerosol size distribution. In GRASP, the smoothness constraints are related to a priori known limited values of the derivatives, i.e. with their m-th derivative

deviations from zero:

$$\frac{\partial^m v(x)}{\partial x^m} \approx 0 \tag{15}$$

For the vector of unknowns $\mathbf{a} = (a_1, a_2, \cdot, a_n)^T$ that contains discrete elements describing the continuous function $v(x)$, the knowledge on the smoothness of function $v(x)$ can be defined using a vector-matrix linear system (e.g., see Dubovik et al. 2021): $\mathbf{0}^* = \mathbf{G}_m^{\mathbf{a}} + \Delta_{\mathbf{g}^*}$, where $\mathbf{G}_m$ is the Jacobean matrix of the matrix of the $m$-th derivatives. In practice, these are often

approximated by matrices of the m-th finite difference estimated in point $\mathbf{a}$. The errors $\Delta_{\mathbf{g}^*}$ reflect the uncertainty in the knowledge of the deviations of $y(x)$ from the assumed constant ($m = 1$), straight line ($m = 2$), parabola ($m = 3$), and so on. Under assumption that the $\Delta_{\mathbf{g}^*}$ have a normal distribution, with the unbiased covariance matrix $\mathbf{C}_g$, these constraints can be easily included in Eq. 12 by defining: $\mathbf{K}_{a,2} = \mathbf{G}_m$ and $\mathbf{f}_2^{a,*} = \mathbf{0}^*$, i.e. $\mathbf{K}_a^T \mathbf{W}_a^{-1} \mathbf{K}_a = \mathbf{G}_m^T \mathbf{W}_{\Delta g}^{-1} \mathbf{G}_m^T$ and $\mathbf{K}_2^T \mathbf{W}_2^{-1} \Delta \mathbf{f}_2^{a,*} = \mathbf{G}_m^T \mathbf{W}_{\Delta g}^{-1} (\mathbf{a}^p - \mathbf{0}^*) = \mathbf{G}_m^T \mathbf{W}_{\Delta g}^{-1} \mathbf{a}^p$. Utilization of such smoothness constraints was suggested by one of the first formulations

of constrained inversion by Phillips (1962) and was also considered in article by Tikhonov (1963) and Tikhonov's later studies.

Thus, for a case where only a direct a priori estimates and smoothness constraints are used, Eq. (12) can be explicitly written via weighting matrices as:

$$\left( \mathbf{K}^T \mathbf{W}_f^{-1} + \gamma_a \mathbf{W}_a^{-1} + \gamma_g \Omega_m \right) \Delta \mathbf{a}^p = \mathbf{K}^T \mathbf{W}_f^{-1} \Delta \mathbf{f}^p + \gamma_a \mathbf{W}_a^{-1} (\mathbf{a}^p - \mathbf{a}^*) + \gamma_g \Omega_m \mathbf{a}^p \tag{16}$$

where $\Omega_m$ denotes the smoothness matrix defined as:

$$\mathbf{G}_m^T \mathbf{W}_{\Delta g}^{-1} \mathbf{G}_m^T = \Omega_m \tag{17}$$

the explicit formulation of $\Omega_m$ can be found in the paper by Dubovik et al. (2011). The Eq. (14) generalizes the commonly used base equations of constrained inversion by Phillips (1962), Twomey (1975, 1977), Tikhonov (1963) and Rodgers (1976, 1990, 2000). It should be noted that Eq. (14) is written for the simplest situation when the vector $\mathbf{a}$ represents only one continues function $v(x)$, while in many GRASP applications the vector of unknowns includes several components $\mathbf{a}^T =$

$(\mathbf{a}_{sd}^T, \mathbf{a}_{n(\lambda)}^T, \mathbf{a}_{k(\lambda)}^T, \mathbf{a}_h^T, ...)^T$, where each component is relevant to continues functions representing such physical characteristics as aerosol particle size distribution ($\mathbf{a}_{sd}$), spectral dependence of real ($\mathbf{a}_{n(\lambda)}$) and complex ($\mathbf{a}_{k(\lambda)}$) parts of refractive index, vertical distribution ($\mathbf{a}_h$), etc. Each of those characteristics is continues function and therefore in retrieval the smoothness constraints can be applied on each of corresponding component of the vector of unknowns. Evidently, direct a priori constraints





can be applied to each single element of the vector $\mathbf{a}$, while from practical view point separating outlining the contribution of a priori estimates for each component, e.g. $(\mathbf{a}^*)^T = ((\mathbf{a}_{sd}^*)^T, (\mathbf{a}_{n(\lambda)}^*)^T, (\mathbf{a}_{k(\lambda)}^*)^T, (\mathbf{a}_h^*)^T, ...)^T$. Similarly, the inverted measurements may come from different sources and therefore have different levels of accuracy and different weighting matrices. As a result, in practice, all the first, second and third terms in Eq. (14) may have many quite different components, and therefore actual formulation of the solution can be significantly more complex. Some of explicit equations can be found in the paper by Dubovik et al. (2011).

The realization of the inversion in GRASP, in principle, is based on general Eq. (7), while for used convenience there is a logical separation as indicated in Eq. (12) into actual measurements and a priori constraints. For each measurement data set $\mathbf{f}_k^*$ two types of errors can be set: relative or absolute and the magnitude of the errors is defined by the standard deviation and a weighting matrix $\mathbf{W}_i$. The standard deviation is used inside of the code to calculate corresponding Lagrange parameters $\gamma_i$. The weighting matrix $\mathbf{W}_i$ is assumed as the unity matrix by default, while it can also be set diagonal with different values at the diagonal, as well, in more general way with non-zero non-diagonal values too. For applying the a priori constraints, as discussed above, there are two main possibilities: using direct a priori constraints or applying smoothness constraints for the parameters that define continues functions.

The direct a priori estimates $\mathbf{a}_i^*$ for each of value $\mathbf{a}_i$ in the vector of unknows $\mathbf{a} = (\mathbf{a}_1, \mathbf{a}_2, ..., \mathbf{a}_n)^T$ can be provided with the corresponding Lagrange parameters $\gamma_{\mathbf{a}_i}$. There is also a possibility to assume a vector $\mathbf{a}^*$ of a priori estimates for all the retrieved parameters or for selected groups (e.g., parameters describing size distribution) with common Lagrange parameter $\gamma_{\mathbf{a}}$. In this case weighting matrix $\mathbf{W}_{\mathbf{a}}$ is also provided that is assumed as the unity matrix by default, or can be set diagonal with different values at the diagonal or in more general way with non-zero non-diagonal values.

The smoothness a priori constraints can be applied for each group of parameters describing a continuous function (e.g., $\mathbf{a}_{sd}^T, \mathbf{a}_{n(\lambda)}^T, \mathbf{a}_{k(\lambda)}^T, \mathbf{a}_h^T, ...$, etc) by defining the order $m$ of limited derivatives ($m = 0$ is a constant; $m = 1$ is a straight line; $m = 2$ is a parabola, etc.) and the strength of the applied a priori smoothness constraints is defined by Lagrange parameters $\gamma_n$. The smoothness matrix $\mathbf{\Omega}_m$ is defined as in Eq. 15 where weighting matrix $\mathbf{W}_{\Delta g}$ is unity matrix by default and be set diagonal with different values on the diagonal in case if the retrieved continues function has different level of variability for different ordinates.

It should be noted that GRASP considers two types of a priori constraints: the 'single pixel' a priori constraints for the retrieved parameters that corresponds to simultaneous and co-located observations and 'multi-pixel' constraints that limit variability for unknowns in different groups of similar parameters when several such groups of unknowns are retrieved simultaneously from coordinated but not fully co-incident or not fully co-located observations (see details in Dubovik et al. (2011, 2021)). In current paper only single pixel constraints are used.

## 2.2 Non-linear inversion in GRASP and used Levenberg-Marquardt optimization

Since most of atmospheric remote sensing applications are strongly non-linear, the Levenberg-Marquardt optimization (Press et al., 1992; Ortega and Rheinboldt, 1970) is realized to optimize convergence of GRASP solutions. Specifically, as described by Dubovik et al. (2021) in GRASP it is assumed that the correction of the solution at p-th iteration $\Delta \mathbf{a}^p$ should be limited,



especially at the initial iterations when the linearization error is the largest. For such cases, in GRASP, for the determination of $\Delta\mathbf{a}^p$ in the iterative procedure an additional constrained on the correction $\Delta\mathbf{a}^p$ is added at each iteration:

$$\Delta\mathbf{a}^{p,*} = \mathbf{0}^* + \Delta\mathbf{a} \tag{18}$$

Correspondingly, using this additional requirement, an additional term will be introduced in Eq. (7):

$$\left(\sum_{k=1}^{K}\gamma_k\mathbf{K}_k^T(\mathbf{W}_k)^{-1}\mathbf{K}_k + \mathbf{D}_{\Delta\mathbf{a}}^p\right)\Delta\mathbf{a}^p = \sum_{k=1}^{K}\gamma_k\mathbf{K}_k^T(\mathbf{W}_k)^{-1}\Delta\mathbf{f}_k^p \tag{19}$$

where matrix $\mathbf{D}_{\Delta\mathbf{a}}$ is diagonal matrix with the elements:

$$\mathbf{D}_{\Delta\mathbf{a}ii} = \gamma_{\Delta\mathbf{a}_i} = \frac{\varepsilon_1^2}{\varepsilon_{\Delta\mathbf{a}_i}^2} \tag{20}$$

The variance $\varepsilon_{\Delta\mathbf{a}_i}^2$ can be determined, for example, assuming that whole known range of each parameter $\mathbf{a}_i$ variability should be covered by $3\varepsilon_{\Delta\mathbf{a}_i}$, i.e. $\mathbf{a}_{i,max} - \mathbf{a}_{i,min} \approx 3\varepsilon_{\Delta\mathbf{a}_i}$.

Also, following common Levenberg-Marquardt procedure the impact of the correction $\Delta\mathbf{a}^p$ is always scaled by a factor $t_p$ in Eq. (4) as follows:

$$\mathbf{a}^{p+1} = \mathbf{a}^p - t_p\Delta\mathbf{a}^p \tag{21}$$

where $t_p$ is in the range $0 < t_p \leq 1$. It is selected empirically to provide convergence, by decreasing $t_p = t_p/2$ until decrease of the residual $\Psi'(\mathbf{a}^p) \leq \Psi'(\mathbf{a}^{p-1})$ is achieved (see Dubovik et al. (2011)).

Thus, in case of non-linear $\mathbf{f}_k(\mathbf{a})$ and/or $\mathbf{f}_i^a(\mathbf{a})$ the inversion in GRASP includes Levenberg-Marquardt like optimizations and implemented in the frame of Eqs. (4) and (5). While this optimization certainly helps to achieve successful convergence of the solution in practice, it also should be considered as one of possible sources of uncertainties, as pointed by Dubovik et al.

(2021) and will be discussed below.

## 2.3 Error propagation estimates in GRASP

Estimations of the retrieval errors in GRASP are based on LSM equations expressed for the case of multi-term solutions written via weighting matrixes (Dubovik et al., 2021). Both the contribution of random and systematic error components are estimated as following:

$$\mathbf{C}_{\hat{\mathbf{a}}} = \mathbf{C}_{\Delta\hat{\mathbf{a}}_{(ran)}} + (\hat{\mathbf{a}}_{bias})(\hat{\mathbf{a}}_{bias})^T \tag{22}$$





where

$$\mathbf{C_{\hat{a}}} = <\Delta\hat{\mathbf{a}}_{(ran)}\Delta\hat{\mathbf{a}}^T_{(ran)}> \approx \left(\sum_{k=1}^{K}\gamma_k\mathbf{K}_k^T\mathbf{W}_k^{-1}\mathbf{K}_k\right)^{-1}\hat{\varepsilon}_1^2 \tag{23}$$

$$\hat{\mathbf{a}}_{bias} = \left(\sum_{k=1}^{K}\gamma_k\left(\mathbf{K}_k^T\mathbf{W}_k^{-1}\mathbf{K}_k\right)\right)^{-1}\left(\sum_{k=1}^{K}\gamma_k\left(\mathbf{K}_k^T\mathbf{W}_k^{-1}\mathbf{b}_k^*\right)\right) \tag{24}$$

where $\mathbf{b}_k^*$ denotes the bias vector in the $k-th$ data set $\mathbf{f}_k$ and $\hat{\varepsilon}_1^2$ is estimated from the resulting miss-fit of the data using Eq.
(9).

The estimation of not only random retrieval error but also error retrieval bias $\Delta\mathbf{a}_{sys}$ is important for the adequate evaluation
of retrieval uncertainty, especially in the case when multiple a priori constraints are used. For example, for the case of the
retrieval given by Eq. (14) $\mathbf{C}_{\Delta\hat{\mathbf{a}}_{ran}}$ is expressed as:

$$\mathbf{C}_{\Delta\hat{\mathbf{a}}_{ran}} \approx \left(\mathbf{K}^T\mathbf{W}^{-1}\mathbf{K} + \gamma_{\mathbf{a}}\mathbf{W}_{\mathbf{a}}^{-1} + \gamma_g\mathbf{\Omega}_m\right)^{-1}\hat{\varepsilon}_1^2 \tag{25}$$

A rather obvious tendency can be seen from the analysis of this equation: the higher the contributions of the second and the
third terms the smaller the random errors are, i.e. the stronger a priori constraints are used the lower the random errors of the
retrieval. However, in practice a priori constraints can be unintentionally inadequate and therefore introduce some systematic
uncertainties, i.e. biases. In principle, there is no guaranteed approach for detecting those biases unless comprehensive analysis
and validation of the retrievals have been done. Nonetheless, some biases can manifest themselves via misfit of measurements
$\Delta\mathbf{f}_k^{bias} = \mathbf{f}_k(\mathbf{a}^{solution}) - \mathbf{f}_k^*$ or misfit of a priori constraints. For example, for Eq. (14) the bias can be introduced by a priori
estimate $\mathbf{a}_{bias}^* = \mathbf{a}^{solution} - \mathbf{a}^*$ or unsmooth features in the retrieved solution: $\mathbf{a}_{bias}^{smooth} = \Omega_m\mathbf{a}^{solution} \neq \mathbf{0}$. Correspondingly,
the bias for single-pixel retrieval is estimated as:

$$\hat{\mathbf{a}}_{bias} \approx \left(\mathbf{K}^T\mathbf{W}^{-1}\mathbf{K} + \gamma_{\mathbf{a}}\mathbf{W}_{\mathbf{a}}^{-1} + \gamma_g\mathbf{\Omega_m}\right)^{-1}\left(\mathbf{K}^T\mathbf{W}^{-1}\Delta\mathbf{f}^{bias} + \gamma_{\mathbf{a}}\mathbf{W}_{\mathbf{a}}^{-1}\mathbf{a}_{bias}^* + \gamma_g\mathbf{\Omega_m}\mathbf{a}_{bias}^{smooth}\right) \tag{26}$$

In this equation the contribution of a priori estimates to bias is probably the most significant in many applications since it
is never possible to have fully accurate a priori values (widely used in optimum estimation approaches) for constraining. In a
similar way, the a priori biases are estimated in the case when multi-pixel a priori constraints are used.

The Levenberg-Marquardt optimization of the convergence, discussed in Section 2.2 may also introduce a bias. Indeed, this
optimization makes the iterations converge from given initial guess to fit the data even if the basic linear system is singular.
Therefore, once Levenberg-Marquardt optimization is used there is an evident dependence on the initial guess that can bias the
solution. In order to take this into account the Eqs. (23) and (24) are modified as the following:

$$\mathbf{C}_{\Delta\hat{\mathbf{a}}_{ran}} \approx \left(\sum_{k=1}^{K}\gamma_k\mathbf{K}_k^T\mathbf{W}_K^{-1}\mathbf{K}_k + \mathbf{D}_{\Delta\mathbf{a}}^p\right)^{-1}\hat{\varepsilon}_1^2 \tag{27}$$





and

$$\hat{\mathbf{a}}_{bias} = \left( \sum_{k=1}^{K} \gamma_k \mathbf{K}_k^T \mathbf{W}_K^{-1} \mathbf{K}_k + \mathbf{D}_{\Delta\mathbf{a}}^p \right)^{-1} \left( \sum_{k=1}^{K} \gamma_k \mathbf{K}_k^T \mathbf{W}_K^{-1} \mathbf{b}_k^* + \mathbf{D}_{\Delta\mathbf{a}}^p (\mathbf{a}^{solution} - \mathbf{a}^{p=0}) \right) \tag{28}$$

These equations allow to obtain the error estimates for the retrieved parameters. That is, for example when the configuration
is from sun photometer and lidar measurements, the expected retrieved parameters are $dV/dln(r)$, real and imaginary part of
refractive index, sphericity fraction and aerosol volume concentration vertically distributed.

Also, in the practice the users may not need directly the retrieved parameters $\hat{\mathbf{a}}$ but their functions $m(\hat{\mathbf{a}})$ that can be calculated
from the retrieved parameters. For example, GRASP retrieves parameters of aerosol microphysics (particle sizes, refractive
indices, etc.) but users need aerosol optical depth, AOD. For such situation, GRASP provides a set of such diverse indirect
characteristics with the possibilities of providing the uncertainties calculated as:

$$\begin{aligned}
\mathbf{C}_{\Delta\hat{m}} &\approx \mathbf{M} \left( \mathbf{C}_{\Delta\hat{\mathbf{a}}_{ran}} + \hat{\mathbf{a}}_{bias} \hat{\mathbf{a}}_{bias}^T \right) \mathbf{M}^T \\
&= \mathbf{M} \mathbf{C}_{\Delta\hat{\mathbf{a}}_{ran}} \mathbf{M}^T + \mathbf{M} \hat{\mathbf{a}}_{bias} (\mathbf{M} \hat{\mathbf{a}}_{bias})^T \\
&= \mathbf{C}_{\Delta\hat{m}_{ran}} + \hat{\mathbf{m}}_{bias} \hat{\mathbf{m}}_{bias}^T
\end{aligned} \tag{29}$$

where $\mathbf{M}$- is the is the matrix of first derivatives $\mathbf{M}_{ji} = \left. \frac{\partial m_j}{\partial a_i} \right|_{\mathbf{a}^{solution}}$
Finally, the effect of biases in the measurements on the solution bias $\hat{\mathbf{a}}_{bias}$ is accounted for in Eq. (26) based on the assumption that the presence of biases is manifested in the non-zero misfits $\Delta\mathbf{f}_k^{bias}$. Indeed, in many cases when systematic errors
are present in the inverted measurements or the accurate fit of inverted data can't be achieved (e.g., see illustrations provided
by numerical sensitivity tests for AERONET retrievals by Dubovik et al. (2000)). At the same time, there are many situations
when biases in the measurements may not significantly affect the residual (Eq. 9) and the misfits $\Delta\mathbf{f}_k^{bias}$. For example, the
retrievals of aerosol SSA from AERONET ground-based measurements are highly sensitive to the calibration biases in the
direct Sun measurements, while the fitting of these direct measurements is always quite accurate (see discussion by Dubovik
et al. (2000)). The effects of such measurement biases can be estimated by implementing proxy numerical tests applied to the
measurements perturbed by possible biases. For example, the recent approach for evaluation retrieval errors of AERONET operational products is estimated using a series of ~27 numerical proxy inversion tests with the sets of perturbations in both input
measurements and auxiliary input parameters (Sinyuk et al., 2020). Similar strategy can be used for evaluation of potential
effects of undetected biases. Specifically, the bias term $(\hat{\mathbf{a}}_{bias})(\hat{\mathbf{a}}_{bias})^T$ in Eq. (22) can be estimated as:

$$(\hat{\mathbf{a}}_{bias})(\hat{\mathbf{a}}_{bias})^T \rightarrow \left\langle (\hat{\mathbf{a}}_{bias})(\hat{\mathbf{a}}_{bias})^T \right\rangle_{\text{bias proxy set}} \tag{30}$$

where the values of the retrieval biases are estimated as an average effect from a preselected set of possible biases in
measurements and auxiliary inputs. Therefore, if we assume positive and negative bias in the equation for systematic component
the contribution to Eq. (26) can be written as follow:





$$\hat{\mathbf{a}}_{bias} \approx \left(\mathbf{K}^T \mathbf{W}^{-1} \mathbf{K} + \gamma_{\mathbf{a}} \mathbf{W}_{\mathbf{a}}^{-1} + \gamma_g \mathbf{\Omega}_{\mathbf{m}}\right)^{-1} \left(\mathbf{K}^T \mathbf{W}^{-1} \mathbf{b}_{\mathbf{f}_{bias}} + \gamma_{\mathbf{a}} \mathbf{W}_{\mathbf{a}}^{-1} \mathbf{a}_{bias}^* + \gamma_g \mathbf{\Omega}_{\mathbf{m}} \mathbf{a}_{bias}^{smooth}\right) \tag{31}$$

where the vectors $\mathbf{b}_{\mathbf{f}_{bias}}$ represent the new bias related to the measurement.

In addition, in this work we also study the structure of the covariance matrix for different aerosols and configurations. Apparently, such matrix provides interesting information about the error estimates (focusing in the diagonal elements) and the relation between the retrieval parameters (from the covariance values, i.e. non-diagonal elements). The representation of the covariance matrix for the parameters has the following structure:

$$Cov(\mathbf{a}) = \begin{pmatrix} \sigma_1^2 & \sigma_1 \sigma_2 \rho_{12} & \sigma_1 \sigma_3 \rho_{13} & \cdots \\ \sigma_2 \sigma_1 \rho_{21} & \sigma_2^2 & \sigma_2 \sigma_3 \rho_{23} & \cdots \\ \sigma_3 \sigma_1 \rho_{31} & \sigma_3 \sigma_2 \rho_{32} & \sigma_3^2 & \cdots \\ \vdots & \vdots & \vdots & \ddots \end{pmatrix} \tag{32}$$

where in the diagonal are the variance of each element and the non-diagonal elements represent the covariance of each retrieved element $a_i$ with the others. The variances, i.e. diagonal elements are always used for estimating retrieval errors and providing the error bars. The non-diagonal elements are rarely considered, while they provide the very interesting and non-obvious information about error correlations.

In order to study the error correlation structure of the error, the following correlation matrix will be considered in this work, that can be obtained from the covariance matrix (Eq. 32):

$$Corr(\mathbf{a}) = \begin{pmatrix} 1 & \rho_{12} & \rho_{13} & \cdots \\ \rho_{21} & 1 & \rho_{23} & \cdots \\ \rho_{31} & \rho_{32} & 1 & \cdots \\ \vdots & \vdots & \vdots & \ddots \end{pmatrix} \tag{33}$$

where each diagonal element corresponds to the correlation with itself which is equal to 1 and the non-diagonal elements are the correlations related to each parameter that can vary between -1 and 1.

## 3 Methodology of error analysis

The calculation of retrieval estimates in GRASP is based on rigorous formulations of statistical estimations described above. At the same time, practical evaluation of developed error formalism and possible tuning is desirable for comprehensive evaluating of the approach and gaining full confidence in the practical efficiency of the approach. In this regard, one can probably state that the errors estimate always tend to be less accurate than the retrievals themselves. Indeed, in remote sensing, the retrieval relies on formalism of electromagnetic light interaction theory that is fundamentally very accurate and well established while





the factors contributing to the uncertainties can be very diverse, not fully formalized and often not even fully understood. For example, the forward model is non-linear, while the error propagations are usually (and in this work specifically) estimated in linear approximations as commented previously, the retrieval can be affected by not fully predicted biases in the measurements or by imperciptent of aerosols or surface models (in our applications). Therefore, the important part of establishing error estimates is their evaluation and validation.

As mentioned above, in this study, we attempt to evaluate the GRASP estimations based on an extensive series of the numerical tests with added random noise that covers a wide range of practical situations. Moreover, we complement this study assuming different biases in order to see how the error estimates are represented in the cases with both: random noise and bias. This section describes the design of the numerical experiment including:

- the instruments and retrieval scenarios used,

- the description of the overall experiment,

- assumed atmospheric properties and covered distinct specific situations of interest,

- the assumptions made for generated random errors, the considered retrieved parameter

- the considered error characteristics, etc.

As mentioned earlier this study evaluates of the GRASP errors estimates produced for the aerosol properties retrieved from ground-based observations. The details of used observations and considered aerosol retrieval scenarios are provided in next sections.

## 3.1 Observations and aerosol retrieval approaches considered

The analysis is focused on two widely known, and probably most popular, retrieval scenarios used for deriving detailed aerosol optical properties:

i) Retrieval of columnar properties of aerosol from the measurements by ground-based sun/sky-scanning radiometer alone;

ii) Simultaneous retrieval of both columnar aerosol properties and their vertical distribution from the combined observations by Sun/sky-scanning radiometer and multi-wavelength lidar.

### 3.1.1 Aerosol retrieval from Sun/sky radiometer alone

All the tests and analyses in this study include the spectral observations by the ground-based Sun/sky-scanning radiometers. These radiometers were used for more than 30 years by the worldwide AERONET project (https://aeronet.gsfc.nasa.gov, Holben et al. (1998)) that unities a federation of large number of ground-based remote sensing aerosol international networks. At present AERONET observations are widely recognized as a benchmark validation data set for satellite aerosol retrieval, as well as, as source of some unique information about detailed aerosol properties that are used in diverse studies on monitoring and





predicting regional and global pollution evolution and climate change. The standard set of AERONET observations includes spectral direct Sun measurements and spectral measurements of angular sky-radiance obtained from sky-scans by the radiometer. The direct Sun observations provide, with rather straightforward processing, spectral Aerosol Optical Depth (AOD) that

itself is highly valuable for satellite product validation and diverse aerosol studies (e.g., Eck et al., 1999). The combination of spectral AOD and sky-radiances at four wavelengths: $440, 675, 870$ and $1020\,nm$ (see Table 1) are used for the retrieval of detailed aerosol size distribution (the aerosol concertation in 22 logarithmically equidistant size bins in the range from 0,05 to 15 microns) together with spectral dependence complex index of refraction (Dubovik and King, 2000). The retrieval also provides aerosol absorption characterized by Single Scattering Albedo (SSA), parameters of fine and coarse mode size distri-

bution and other diverse detailed properties of columnar aerosol (e.g., see Dubovik et al., 2002b). In addition, based on the concept developed by Dubovik et al. (2002a, 2006), AERONET retrieval considers aerosol as a mixture of two components, spherical and non-spherical and provide a fraction of spherical particles as an additional parameter. The non-spherical fraction is modelled as a mixture of randomly oriented spheroids using fixed axis ration distribution equal to the one retrieved by Dubovik et al. (2006) by inverting full phase matrices of Feldspar dust sample measured in laboratory by Volten et al. (2001).

A rather complete description of this retrieval concept is provided also in the paper by Dubovik et al. (2011). The set of the aerosol parameters retrieved in AERONET standard operational processing is shown in Table 1. The AERONET operational retrieval were mainly provided for Solar almucantar geometry mainly, while since recently the AERONET has also provide retrievals for more complex hybrid observational geometries (Sinyuk et al., 2020). These studies are focused on retrieval in Solar almucantar only, while GRASP algorithm provides the error calculations for any geometry. Thus, in this study, the direct

sun measurements and sky radiances both at 4 different wavelengths $440, 675, 870$ and $1020\,nm$ are used in the inversion tests. These sky radiances measurements are measured in the Solar almucantar (fixed view zenith angle equal to the solar zenith angle, SZA) with a varying azimuth angle ranging from $\pm3.5$ degrees to $\pm180$ degrees (Table 1).

The detailed aerosol properties in the total atmospheric column provided by AERONET inversion of Sun/sky-scanning radiometers has been widely recognized as rather unique reliable data. For example, AERONET retrievals provided first reliable

data about aerosol spectral absorption and other detailed aerosol optical characteristics (e.g., see Dubovik et al., 2002b; Giles et al., 2012, etc.). These detailed data are of vital importance for evaluating the impact of aerosol on such important aspects as a climate change and diverse pollution effects, and can be reliable estimated nearly uniquely from remote sensing observations (Kaufman et al., 2002). Therefore, this retrieved aerosol information has been proven to be very useful for assessment of climate change dynamics in IPCC reports (Boucher et al., 2013; Masson-Delmotte et al., 2021) and other high profiles analyses.

As mentioned in earlier sections the evaluations of the accuracy of retrieved aerosol parameters was mainly relied on extensive sensitivity studies by Dubovik et al. (2000). The results were used for providing quality assurance criteria and expected accuracy estimation (see Dubovik et al., 2002b; Holben et al., 2006). Sinyuk et al. (2020) recently presented the approach to estimate retrieval uncertainties used in AERONET Version 3 data. The approach estimates the error using the variability in retrieved parameters generated by 27 perturbations in both input measurements and auxiliary input parameters. In comparisons

with these previous efforts this study evaluates the dynamic error estimates generated by GRASP approach for each retrieval based on the measurement error propagation and bias estimations. This study estimates the complete covariance matrices of





retrieval errors. In addition, the analysis of retrieval errors and their correlations conducted here is also aimed for demonstrate the value of obtained estimates for understanding the retrieval error tendencies and optimizing the retrieval approaches.

### 3.1.2 Aerosol retrieval from a combination of Sun/sky radiometer and lidar data

The inversion of co-located observation by Sun/sky radiometer and lidar is another popular retrieval approach in aerosol community. Indeed, radiometer direct Sun and multi-angular polarimetric observations of diffuse Sun radiation transmitted through the atmosphere have significant sensitivity to the atmospheric aerosol amount, its particles size, shape and morphology; however, they have practically no sensitivity to the vertical variability of aerosols. The lidar observations on the other hand provide the information about vertical distribution of aerosol while their sensitivity to other aerosol properties is more limited compare
to radiometer observations. Therefore, the information from collocated photometric measurements and lidar systems is complementary and always desirable for enhanced characterization of aerosol properties. This complementarity is well recognized by the research community and large number of joint observational sites with both radiometer and lidar observations have been established in last decade. In these regards, European ACTRIS (Aerosols, Clouds, and Trace gases Research Infrastructure Network) infrastructure (https://www.actris.eu) is one of the good examples of networks emphasizing the acquisition of
diverse complementary observations at each site. All ACTRIS observational supersites possess both Sun/sky radiometric and complex multi-wavelength lidar systems.

GRASP retrieval has been successfully adapted by Lopatin et al. (2013) for processing such combined observations in algorithm initially known as Generalized Aerosol Retrieval from Radiometer and Lidar Combined data (GARRLiC). This algorithm has been used in numerous studies (e.g., Granados-Muñoz et al., 2014, 2016; Tsekeri et al., 2017; Benavent-Oltra
et al., 2017, 2019, 2021, etc.) and was also adapted for operational processing of lidar/radiometer observation in frame of ACTRIS infrastructure. Later the capabilities of GARRLiC/GRASP were significantly extended by Lopatin et al. (2021) for processing diverse vertically resolved observation alone in diverse combination with radiometric observations.

In these studies, we consider the aerosol retrieval from the base GARRLiC/GRASP input data set that includes AERONET Sun/sky scanning observations in solar almucantar at four wavelengths and lidar back-scattering attenuation profile at three
wavelengths at $355$, $532$ and $1064\,nm$ (see Table 1). Thus, for the synergy of sun/sky radiometer and lidar measurements we considered the same sun/sky radiometer input data combined with the correlative range corrected signal (RCS) values, at $355$, $532$ and $1064\,nm$. The lidar signal provided in GRASP as input data is normalized at 60 log-spaced bins at different heights, as in Lopatin et al. (2013, 2021), giving a minimum and maximum heights. It is because all lidars provide observations within a certain distance range, which varies from instrument to instrument and it is limited by emitter/receiver field of view overlap in
the lower part as well as by the signal-to-noise ratio in the upper part. The GRASP inversion of these data derives in addition to columnar aerosol properties provided from radiometer only inversion, the vertical profile of aerosol concentration. Moreover, the aerosol can be considered as an external mixture of two aerosol components (fine and coarse). In such case, all retrieved parameters are provided for both aerosol modes as shown in the Table 1.





## 3.2 Structure of different error parameters analysis

As was already mentioned, GRASP has a capability to provide the full covariance matrix of the retrieval errors and this study is aimed to evaluate and illustrate the efficiency of these estimated covariance matrices. At the same time, retrieval error evaluations in most of practical applications rely on consideration of mainly diagonal elements of the covariance matrices while non-diagonal elements of covariance matrices are much less common. Indeed, in spite of the fact that non-diagonal elements of covariance matrices provide valuable and interesting information about retrieval errors correlations, these non-diagonal

elements are not often available in practice and the analysis of error correlations requires more sophisticated considerations compare to straightforward analysis diagonal elements only, and therefore it is less popular. Considering these aspects, in present study, as a first step, make more detailed and extensive analysis of error variances and then, as a second step, we illustrate the usefulness of obtained non-diagonal elements.

The performance of GRASP error variances estimated provided by Eqs. (25-27) is studied using a series of numerical tests.

Figure 1 illustrates the general scheme of these test organization.

First, as showed in Fig. 1, the parameters $\mathbf{a}_{assumed}$ for assumed detailed aerosol properties ($dV(r_i)/d\ln r_i$, $n(\lambda_i)$, $k(\lambda_i)$, $C_{sph}$ and $C_V(h)$ in the case of lidar) are used to obtain the synthetic observations using the GRASP forward model. These synthetic observations include the spectral AOD, sky radiances and range corrected signals (RCS) of lidar. These data are used then in the inversion tests where the aerosol parameters and their errors are estimated from these synthetic observations using

GRASP algorithm. In order to study the effects of the different uncertainties both random and systematic errors are added to the synthetic measurements before the inversion, then the retrieved parameters $\hat{\mathbf{a}}_{retr}$ are compares with $\mathbf{a}_{assumed}$ (see Fig. 1)). Therefore, from the retrieved parameters, the retrieval errors provided by GRASP algorithm and 'actual' retrieval errors can be compared. Thus, these actual errors are calculated comparing $\mathbf{a}_{assumed}$ and $\hat{\mathbf{a}}_{retr}$ as follow:

$$\begin{cases} \Delta\hat{\mathbf{a}}_{abs} = \hat{\mathbf{a}}_{retr} - \mathbf{a}_{assumed} \\ \Delta\hat{\mathbf{a}}_{rel} = \dfrac{\Delta\hat{\mathbf{a}}_{abs}}{\mathbf{a}_{assumed}} \cdot 100\% \end{cases} \tag{34}$$

where $\hat{\mathbf{a}}_{retr}$ is the retrieved parameter by GRASP algorithm and $\mathbf{a}_{assumed}$ is the parameter assumed in the input data for generation of the synthetic observation. Equation (34) is used for each retrieved parameter including the size distribution value at each size bin, the values of complex refractive index at each wavelength, the values of aerosol vertical profile at each altitude and the values of spherical particle fraction. We also implemented the evaluations of the errors for aerosol SSA, and other parameters, that are not part of the directly retrieved parameter while a function of the retrieved parameters and it is estimated

based on $\hat{\mathbf{a}}_{retr}$. Thus, the retrieval error variances estimated by GRASP can be compared with the calculated actual retrieval errors. It should be noted here that we have always verified that the errors of the retrieval realized by GRASP from the 'error free' synthetic data (i.e. with no error specifically added) are negligibly small.





GRASP generated variances of the retrieval errors are evaluated in the presence of random errors and analyzed using a series on the numerical tests conducted for statistically representative set of random error realizations. These results are then summarized for the whole series of the tests by figures and tables. The tests with added systematic errors are discussed for most of separate systematic error type while some overall summaries are also provided.

As was mention before, in addition to the standard deviation the non-diagonal elements of covariance matrices provide additional important inside about retrieval quality. This additional information mainly relates with non-zero correlation coefficients. Therefore, in order to illustrate the correlation structure, in this work we also analyzed the correlation matrix that contains the covariance matrix elements normalized by the respective variances as shown by Eq. (33). Our studies are not attempting to evaluate correlation matrices provided by GRASP algorithm since this would require the efforts exceeding the scope of this paper. Instead, we try to provide several demonstrations of how the structure of the correlation matrix may help to understand several interesting observations in existing retrieval experience.

### 3.3 Aerosols models and realizations used in the tests

The synthetic tests were performed for several preselected realizations of aerosol in the atmosphere. These realizations were selected based on extensive experience with aerosol retrieval from Sun/sky radiometer data and their combination with co-located lidar data. It is expected that the selected aerosol realization scenarios are representative of the majority of distinct actual observations of atmospheric aerosols.

Two main observational scenarios are considered:

    i) single aerosol, such as biomass burning (BB), urban and dust for different aerosol loads $\tau(440) = 0.3, 0.6$ and $0.9$; and

    ii) the mixture of dust with BB and with urban (BB-Dust and Urban-Dust). For each mixture, we have selected nine different scenarios that correspond to three different aerosol loads, $\tau(440) = 0.2, 0.5$ and $1.0$, where the different cases of partition between fine and coarse mode were: $\tau_f/\tau_c = 4.0$, $\tau_f/\tau_c = 1$ and $\tau_f/\tau_c = 0.25$.

The single aerosol and aerosol mixture observational scenarios are used in generation of synthetic tests with sun/sky photometer-only observations. By considering both single and two aerosol types, in this work we evaluated how the accuracy of the retrieved evolve once larger number of parameters are derived from the same information content. In a contrast, the retrieval based on the synergy between lidar and sun/sky photometer is aimed for the retrieval of the properties of two fine and coarse mode aerosol components, therefore the numerical tests for this type of the retrieval rely on mixed aerosol observation scenario. At the same time, the error estimation is also checked in the case when the joint radiometer and lidar observation of single aerosol are analyzed. The properties of each aerosol type were modeled using the climatology of aerosol retrievals from AERONET observations described by Dubovik et al. (2002a) and Torres et al. (2017). The dynamic climatological model from Mongu (Zambia) was used for BB aerosol, model from GSFC (Maryland, USA) for urban aerosol and model from Solar Village (Saudi Arabia) for dust aerosol. The real refractive index (RRI) and imaginary refractive index (IRI) for $\lambda = 355$ nm, $532\,nm$ and $1064\,nm$ (lidar measurements) were obtained by the extrapolation of the values from Dubovik et al. (2002a) as was suggested by Torres et al. (2017). All the scenarios were simulated assuming a solar zenith angle (SZA) equal to $75$ degrees.





The retrieval settings were used similar to those that conventionally used in retrieval of aerosol from AERONET Sun/sky-radiometer observations by Dubovik and King (2000) and from combined observations by Sun/sky-radiometer and lidar by Lopatin et al. (2013, 2021). Specifically, in the retrievals from Sun/sky-radiometer only observations the size distribution (SD) was simulated using 22 logarithmically equidistant size bins between 0.05 to $15\,\mu m$. In retrieval of aerosol mixture

from combined observations by Sun/sky-radiometer the size distribution is modelled using 10 logarithmically equidistant bins between 0.05 and $0.58\,\mu m$ for the fine mode and 15 logarithmically equidistant bins between 0.33 and $15\,\mu m$ for the coarse mode. Similar approach was employed in the retrieval from Sun/sky-radiometer only observations in attempt when bi-component aerosol model was retrieved.

## 4 Test results

Several tests were realized to evaluate the error estimates reliability and usefulness in the presence of both random and systematic uncertainties for aerosol retrievals from the observations of Sun/sky-radiometer alone and in a combination with lidar. In this section are presented the results for two scenarios: (i) as simpler case when only one type of aerosol present and (ii) more complex case then when two distinct types of aerosol present at the same time. Moreover, we estimates the correlation matrices for both scenarios and illustrate their usefulness for understanding retrieval error tendencies optimizing retrieval approach.

### 4.1 Random error analysis

In series of these tests, to all inverted the synthetic measurements, we added random noise with standard deviation of $\varepsilon_{\Delta\tau}(\lambda) = 0.01$ for AOD, $\varepsilon_{\frac{\Delta I}{I}}(\lambda) = 5\%$ for radiances in order to model realistic uncertainties of AERONET observations (Holben et al., 1998; Eck et al., 1999; Dubovik et al., 2000; Sinyuk et al., 2020), and $\varepsilon_{355} = 0.2$, $\varepsilon_{532} = 0.15$ and $\varepsilon_{1064} = 0.1$ for lidar attenuation measurements that vary with the altitude as explained by Lopatin et al. (2013, 2021).

### 4.1.1 Retrieval of single aerosol component from radiometer measurements

This section describes the evaluation of the error estimates assuming presence of only one type of aerosol: BB, urban or dust. As we mentioned before, the retrieval aerosol properties under assumption of the presence of single aerosol type composed by homogeneous particles is well-established approach for deriving detailed aerosol properties from ground-based observations by Sun/sky-radiometer that is adapted in operational AERONET retrievals by Dubovik and King (2000). The detailed error

analysis of AERONET inversion aerosol product was provided by Dubovik et al. (2000), Torres et al. (2017) and by the recent study of Sinyuk et al. (2020) that described the uncertainty approach adapted in for AERONET Version 3 retrieval products.

Figures 2 - 4 illustrate the error variances estimated by GRASP for all retrieved aerosol parameters in the selected synthetic tests for observation of BB, urban and dust with different aerosol loads: $\tau(440) = 0.3$, $0.6$ and $0.9$. The displayed error bars standard deviation are calculated from the diagonal elements of the covariance matrix (Eq. 32). Some tendencies can be seen

from these illustrations. For example, the errors of SD in the extremes (for the largest and smallest particles) are the biggest.

This is an expected tendency since these particles have typically a lower contribution to the measured signal (radiances and aerosol optical depths) compared to the particles of intermediate radius.

The retrievals improve and the error decrease when the aerosol load increases, specially for IRI and SSA (absorption information). For BB and urban, SSA error increases with the wavelength. On the other hand, SSA error decreases with the wavelength for dust. This is an expected behavior since the scattering efficiency is more pronounced at short wavelengths for small particles while it is somewhat increasing with wavelength for large particles. Furthermore, as shown in Fig. 2a, the observed underestimation in the SD fine mode seems to be related with an overestimation in RRI.

To evaluate the error estimates in presence of random noise, a set of the simulations for 300 different realizations of noise modelled using random numbers generator has been analyzed in this work. The results of such numerical tests conducted with statistically representative set of random errors which are summarized and illustrated using boxplots of the errors as demonstrated in Fig. 5 for SSA(675) values. In the upper part of the figure, the box represents $50\%$ of the data with the whiskers representing $5th$ and $95th$ percentiles of the data, the solid line in the boxplot representing the median and the points are the mean values.

Figure 6 shows the distributions of the error estimates provided by GRASP (in red) and the calculated errors (in blue) for the cases when $\tau(440) = 0.6$ and the following convergence criteria are satisfied: $\Delta\tau \leq 0.01$ and $\Delta I/I \leq 5\%$. It can be seen that overall the error estimates provided by GRASP show a capture quite well the 'actual' error tendencies with some overestimation of their values. Thus, the retrieved errors can be considered as upper estimates of actual errors. This observed general overestimation can be, at least partially, explained by the fact that the error estimates by Eqs. 23 and 24 rely on linear approximation. In this respect it is known from practice that non-linear effects often lead to some saturation while that cannot be captured by linear estimates. Some interesting tendencies can be appreciated in the obtained illustrations. For example, errors in SSA increase with the wavelength for BB and urban, and decreases for dust.

On the other hand, the RRI errors to be similar at the different wavelengths. This is likely related to the fact that spectrally RRI retrievals rely on rather strong smoothness constraints on spectral variability of RRI (e.g., Dubovik and King, 2000). In contrast to the RRI, a large variability of the calculated errors is observed in the distribution of the errors for the IRI. Indeed, in order to capture possible real spectral variability of IRI as that of dust (e.g., see Dubovik et al., 2002a) the IRI is retrieved under milder smoothness constraints on spectral variability (see Dubovik and King, 2000)). Some of the fore-mentioned and other tendencies in the retrieval errors will be further discussed and evaluated in the section which deals with error correlation matrices (Section 4.3.1).

Table 2 summarizes the evaluation of the error estimates represented in the boxplots. It provides the mean values for each parameter (RRI, IRI and SSA) at different wavelengths. These values correspond to the situation with a solar zenith angle equal to 75 degrees. The obtained estimates compare reasonably with the corresponding values provided by the Table 4 of the paper by Dubovik et al. (2006). Specifically, RRI error at $440\,nm$ provided by GRASP for BB is 0.079 (0.04), where the values in parenthesis are from Dubovik et al. (2000) and for urban it is 0.056. The IRI error at $440\,nm$ is $24\%$ ($30\%$) for BB, $54.1\%$ for urban and $24.4\%$ ($50\%$) for dust. The values of the SSA errors 0.028 (0.03) for BB, 0.013 for urban and 0.014 (0.03) for dust. At the same time, RRI error at $440\,nm$ provided by GRASP for dust it is 0.201 (0.04) is quite different, though Dubovik





et al. (2000) considered only spherical particles. Moreover, the error estimates for SSA are consistent with the U27 estimates provided by Sinyuk et al. (2020). For example, at $440\,nm$ for AOD $= 0.6$ the corresponding value for GSFC is $0.017$ while grasp provides values of $0.013$; for Mongu is $0.023$ while grasp provides error of $0.028$.

### 4.1.2 Retrieval of mixed aerosol properties from measurements of radiometer only

As it was already mentioned, most conventional aerosol retrievals from ground-based radiometer measurements (e.g., Dubovik and King, 2000; Nakajima et al., 2020) assumes that aerosol is represented by homogeneous polydisperse particles with the size independent refractive index. At the same time, this condition is not always correct in reality. Moreover, it is likely somewhat incorrect in majority of the cases. Dubovik et al. (2000) showed in that cases the retrieval assuming homogeneous particles would provide effective index of refraction that allows to reproduce the scattering properties of mixed aerosol rather

adequately. Nonetheless, the assumption of homogeneous particles is often questioned and revisited (Xu et al., 2015), therefore considerations of aerosol inhomogeneity is included in present studies also. In this regard, while the retrieval of the multi-component aerosol is not a part of the standard AERONET inversion, GRASP algorithm allows the retrieval of several aerosol components from diverse remote sensing observations including the case of aerosol retrieval from radiometer measurements only.

At the same time, since the retrieval of multi-component aerosol from radiometer only is not often used and not employed for operational retrievals, the tests in this section are limited only to several illustrations and no statistical evaluation is performed. The illustrations are produced for the observations of a mixture of Urban-Dust and BB-Dust (see Section 3.3) for three cases of total $\tau(440) = 0.2, 0.5$ and $1.0$. In particular, we illustrate the case for $\tau(440) = 1.0$, with $\tau_f = 0.8$ and $\tau_c = 0.2$, $\tau_f = \tau_c = 0.5$ and $\tau_f = 0.2$ and $\tau_c = 0.8$, anticipating more potential for adequate retrieval of multi-component aerosol since the effect of

AOD errors decreases for higher AOD. The analysis is focused on possibilities of the differentiation between the properties of fine and coarse aerosol mode parameters such as complex refractive indices, size distributions, single scattering albedo.

Figures 7 and 8 illustrate results of bi-component retrievals and their error estimates from observations of Sun/sky-radiometer of mixed aerosol. In the same figures, we also show a zoomed plot for the effective RRI and IRI and the total SSA with their errors. Several retrieval tendencies are evident from the figures. For example, in the presence of one mode dominated in optical

thickness, the retrievals and error estimates of dominating component are more accurate. For example, in Fig. 7c for $\tau_f = 0.2$ and $\tau_c = 0.8$, the retrievals of the coarse mode properties are more accurate. An opposite behavior can be seen in Fig. 7a for $\tau_f = 0.8$ and $\tau_c = 0.2$ when the predominance is in the fine mode. The clear trend can be observed in spectral dynamic of the error values for SSA: the error increases with the wavelengths in the fine mode and decreases for the coarse mode.

The most obvious difficulties in the separation of modes are evident when the properties of each mode are not very different.

For example, the such situation can be seen for IRI of Urban-Dust mixture (Fig. 7) and for RRI of BB-Dust mixture (Fig. 8). In such situations the error variances of each parameter are large and likely correlated (more details provided in discussion of covariance matrices). However, it is very important to note that while the discrimination of some parameters of each component separately is not evident, most of the total and effective properties (zoomed plots) can be estimated rather accurately.



### 4.1.3 Retrieval of mixed aerosol properties from measurements of radiometer in combination with lidar

The GRASP aerosol retrieval from were combined Sun/sky radiometer and lidar observations were always designed for retrieval of bi-component aerosol (Lopatin et al., 2013) and the approach is employed for operational processing in frame of ACTRIS activities. Therefore, the evaluation of the random error effect in the aerosol retrieval from radiometer and lidar observations of aerosol mixtures include both analysis of the selected illustrations and the statistically representative series of numerical tests with random errors. The considered synthetic data include synthetic observations produced for the same

examples of aerosols mixture (Urban-Dust and BB-Dust) as used in the Section 4.1.2.

Figures 9 to 12 illustrate the retrievals and their error estimates obtained for aerosol properties of both fine and coarse aerosol modes. The good agreement of actually retrieved parameters (solid lines) with the assumed values (dashed lines) can be seen for all cases. From comparison of Figs. 7-8 with Figs. 9-10, it is easy to see that the retrieval error estimate is lower when lidar data also used. The improvements (compared to the results from radiometer only retrievals) are especially evident in the

615 separation of the retrieved aerosols properties, especially when the contribution of the aerosol load is lower (this will be shown in Section 4.2 when also bias is assumed). At the same time, the total properties are accurately estimated in both retrieval scenarios.

Figure 11 shows the lidar ratio of fine mode, coarse mode and total aerosol for the three cases aforementioned. In general, good agreements of retrieved and assumed values are obtained, specially for the total LR. However, there are some discrepan-

620 cies at short wavelengths for fine mode lidar ratios.

Figure 12 illustrates the retrieval of aerosol vertical profile for each case. The agreement between retrieved and assumed values of vertical profiles is good mainly for the coarse mode at the altitudes where it has maximum and dominates. At the altitudes where there is a superposition of aerosol layers with comparable presence of both aerosols the retrieval struggles to discriminate the contribution of both modes and a clear overestimation of fine mode (and a consequently underestimation of

625 coarse mode) can be seen.

In order to evaluate the error estimates in presence of random errors, a set of simulations with adding 300 realization of random noise values is analyzed. Figures 13 to 19 show the comparisons of all retrieved aerosol parameters separately for fine and coarse aerosol modes. In addition, the retrieval of total SSA and LR are shown. The case for total $\tau(440) = 1.0$ is shown more extensively, as in previous Section, due to the interest the retrieval in the situation with higher aerosol loads. The main

result that can be gained from illustrations is that GRASP error estimates are typically higher than actual errors; this same result was obtained for the retrieval of only photometer data.

Figures 13 and 14 illustrate the comparisons of distributions of the GRASP error estimates and actual errors for RRI and IRI of fine and coarse aerosol modes for situations when mixtures of Urban-Dust and BB-Dust are observed. It can be seen that the accuracy of the refractive index retrievals for each mode depends strongly on the contribution of the mode to the signal, as was

635 observed by Lopatin et al. (2013). For example, if we analyse the performance of the fine mode, the higher the contribution of fine optical thickness, the better the accuracy in the retrievals of fine mode aerosol parameters.





Figures 15 and 16 show the situation for error distribution for SSA of fine mode, coarse mode and total. Similarly as observed in earlier tests, for the retrieval of fine mode parameter, the errors increase with the wavelength while for the retrieval of the coarse mode parameters the errors decrease with the wavelengths. Also, the results show the error in the total SSA are rather
small even if the SSA of fine and coarse modes are quite high.

The error evaluation for LR is represented in the Figs. 17 and 18. In most of the cases we see good agreements between the error estimations and actual error. The only exception is the errors of LR of fine mode at short wavelengths where the actual errors are higher than the errors provided by GRASP. This tendency seems to be anticorrelated with the results found for the coarse mode LR error estimates, where the GRASP error estimates are notably higher than actual values.

The results illustrated by the figures are summarized in Tables 3 to 5. These tables show the mean values of the GRASP error estimates for the cases when the total $\tau(440) = 1.0$ and $\tau_f = \tau_c = 0.5$, i.e. when there is no predominance of either modes. The values are provided for the aerosol parameter considered at different wavelengths both for simulation of Urban-Dust and BB-Dust observations calculated for a case of the SZA= 75 degrees.

For retrieval error of fine mode in the case of urban aerosol parameters, the mean values for RRI are around $0.05$, and the
values do not present much variability with the wavelength. For retrieval of IRI, the mean values of the GRASP retrieval errors are at the level of around $73\%$, showing a pronounced underestimation respect to the actual error, at short wavelengths. With respect to SSA errors provided by GRASP a clear tendency is observed: the error increases with the wavelengths from $0.024$ to $0.061$. Finally, the mean values of LR errors provided by GRASP decrease with the wavelength between $15\%$ to $10\%$, with notable underestimations respect to the actual errors at short wavelengths. In the case of the retrieval of fine mode BB
parameters, mean values for RRI errors provided by GRASP are around $0.05$. Some underestimations respect to the actual errors are observed at short wavelengths. The mean values for IRI errors are around $60\%$ and the errors for SSA show a clear tendency to increase with the wavelengths between $0.04$ to $0.09$. Mean values of LR errors provided by GRASP decrease with the wavelength between $18\%$ to $14\%$.

The mean values of error estimates provided by GRASP for dust present good agreement in case of both mixtures. In
general, the mean values for RRI error estimates vary between $0.07$ to $0.09$ and they do not present much variability with the wavelength, while the smaller values of errors are seen for Urban-Dust mixture case. The mean values of IRI error estimates are around $50\%$, while for BB-Dust mixture we observe some underestimations of actual errors by GRASP calculations. The errors of SSA show a clear tendency that decrease with the wavelengths between $0.04$ to $0.009$. The mean values for LR retrievals increase with the wavelength from $37\%$ to $60\%$, with bigger errors observed for BB-Dust mixture.

Once again, it is important to note that the errors of the parameters characterizing total aerosol are generally accurately estimated. For both cases of Urban-Dust and BB-Dust mixtures, the mean values of total SSA error estimates vary between $0.02$ to $0.009$ and the mean values of total LR error estimates are in the range of $23\%$ to $55\%$.

Figure 19 shows the relative errors of AVP retrievals for fine and coarse aerosol modes for Urban-Dust and BB-Dust aerosols mixture. The errors estimated by GRASP are a bit higher that the errors obtained by simulations of random errors, corre-
spondingly the GRASP errors can be safely used as upper estimates of actual retrieval uncertainties. Table 5 summarizes the





evaluation of the errors estimates for all the scenarios discussed above. The GRASP estimates of the retrieval errors for both mixtures are between $50 - 70\%$ for the fine mode and $50 - 57\%$ for the coarse mode.

Finally, a lower sensitivity to the retrieval of fine mode properties can be observed as a clear tendency in the evaluation of the retrieval errors for the cases when mixed aerosols are analyzed. In particular, quite high errors were obtained for the complex refractive index. Then, these errors consequently propagate to the errors of other optical properties such as SSA of fine mode, as was found in the earlier study by Lopatin et al. (2013).

### 4.2 The analysis of the retrieval in presence of the systematic uncertainties

In the Section 4.1 was presented the evaluation and validation of the errors of the different aerosol properties considering propagation of the random noise from measurements into retrieval. The analysis confirmed rather satisfactory performance of the approach adapted in GRASP for the estimation of retrieval errors in the presence of random noise. This section discusses the approach for estimating possible contributions of the systematic errors in the retrieval uncertainties. In principle, each retrieval methodology assumes that there is no systematic uncertainties neither in measurements nor in the used forward model. If any systematic bias is identified it is corrected in measurements or in their interpretation. However, in practice the systematic uncertainties may remain unidentified and make significant contribution in the retrieval uncertainties.

As mentioned above, in Eq. 24 the apparent misfit was used as an indicator of bias, however in real situations not all biases can be seen from the misfit. Thus, in this section the results are presented considering a possible solution to this problem. Therefore, commonly the contribution of potential bias is included in the estimation of the retrieval errors (e.g., see Dubovik et al., 2000; Sinyuk et al., 2020). Using similar logic, in the present methodology was added an extra term, Eq. 31, that accounts for propagation of possible bias from the measurements. The propagation is accounted for linear approximation in similar manner as the systematic term in Eq. 28 accounts for bias from misfit. Thus, this section analyses the potential effect of realistic biases and their overall importance for reliable estimations of the retrieval errors in practice.

The potential effect of the systematic errors is analyzed in series of the numerical tests with possible assumed systematic errors. Following previous studies by Dubovik et al. (2000); Torres et al. (2017); Sinyuk et al. (2020) in ground-based photometric and radiometric data, we consider two types of potential main biases in measured AOD and sky-radiances. These biases could be originated from miss-calibrations of direct Sun or diffuse sky sensors (Eck et al., 1999). The biases are assumed wavelength independent, since spectral systematic deviations are easier to identify in direct analysis of observation, they are likely to be manifested in misfit and may compensate each other influence on the retrievals. Specifically, two possible levels of biases nominal and maximum are considered as follows:

i) in AOD, nominal bias of $\pm 0.01$ and maximum bias of $\pm 0.02$, and

ii) in radiances nominal bias of $\pm 3\%$ and maximum bias of $\pm 5\%$.

To evaluate the effects of biases, the above values were added to synthetic direct measurements of AOD and sky-radiance by AERONET like ground-based radiometer. These data were inverted by GRASP code and the retrieved values of aerosol parameters were compared to the values assumed in synthetic simulation as a 'truth'. In addition, the deviations of retrieved





values from the 'true' ones are compared to the errors estimates generated by GRASP based on Eq. 31 using known values of
705 added biases. In similar manner, the influence of the potential systematic errors in aerosol retrieval from combined observations
of ground-based radiometer and lidar was analyzed. In these tests, the biases in lidar attenuation measurements were assumed
for each wavelength following the studies by Lopatin et al. (2013, 2021): $\varepsilon_{355} = \pm 0.2$, $\varepsilon_{532} = \pm 0.15$ and $\varepsilon_{1064} = \pm 0.1$. It
should be noted that conducted synthetic tests not only allow to verify the accuracy of the systematic error estimates by
GRASP and also to analyze the effects of biases on the retrievals for different retrieval scenarios in diverse situations.

**4.2.1 Effects of measurement bias in retrieval of single aerosol component from radiometer measurements**

In this section the study is focused on the analysis of the effects of the biases and on estimating contribution of systematic
errors in retrievals of aerosol from ground-based observation by radiometer. In similar manner as in the analysis of random
errors, first was consider the observations dominated by two types of aerosols: BB and dust. The effect of measurement biases
is expected to be manifested in the situations with low and moderate aerosol loading, therefore, the analysis is focused on the
715 scenarios with AOD(440) = 0.1, 0.3 and 0.6.

Two situations were considered:

i) when a single bias in AOD or radiances is present;

ii) when the biases can be present in both AOD and radiances simultaneously. In this case, the different combinations of
positive and negative biases in AOD and radiances are considered.

The estimations of the errors introduced by the biases were calculated as:

$$\sigma_{bias}^2 = \sigma_{lm}^2 + \sigma_{misfit}^2 + \frac{1}{N} \sum_{k=1}^{N} \sigma_k^2 \qquad (35)$$

where $\sigma_{lm}^2$ corresponds to contributions from systematic errors introduced by the Levenberg-Marquardt procedure and
$\sigma_{misfit}^2$ are the errors manifested by the miss-fit estimated by Eq. 28, and each $\sigma_k^2$ is the contribution adding $+$ bias and
$-$ bias in the measurements.

Figures 20 to 23 illustrate the results of the analysis for the different retrieved properties for situation with bias of $\pm 0.01$ and
$\pm 0.02$ in AOD only. These results show specific effects from AOD bias. The figures have two blocks: on the left the retrievals
with added positive bias in AOD and on the right retrievals with negative bias are illustrated. In both cases, the error bars
represent the systematic component adding the positive or negative bias respectively. In all the figures, the solid lines show
the assumed value of the parameters in the simulation, the dotted lines show the retrieved values and the magnitudes of the
730 estimated bias are shown by the shaded areas. It should be noted that for the case of BB with AOD(440) = 0.1, the results with
negative bias are not shown. This is because, the AOD for BB decreases very strongly with the wavelength and for the case of
AOD(440) = 0.1, the AOD at $1020\,nm$ is $\sim 0.01$.

The figures show different and clear tendencies which are in agreement with general expectations and with the tendencies
already observed in previous studies by Dubovik et al. (2000) and Torres et al. (2014). For example, it can be seen that bias



in AOD most strongly affects the estimate of the parameters characterizing aerosol absorption such as imaginary part of the refractive index and single scattering absorption. This is an anticorrelation: the positive bias results in overestimation of absorption (higher RRI and lower SSA) and the negative in underestimation absorption (lower RRI and higher SSA) respectively. The result was quite expected since radiance values in this first experience do not vary. Thus, if we keep the scattering component (which is derived from radiances) but we enlarge the extinction component (by enlarging the AOD), necessary the retrieval

understands that the absorption should be larger (imaginary part of the refractive index). Conversely, if we reduce the value of extinction the retrieval would reduce the value of absorption. Also, the strongest effect is observed for optically thin situations when a small absolute error in optical thickness becomes comparable with the magnitude of aerosol optical thickness. This is especially clear for BB observations where AOD(1020) is always rather small as earlier discussed also by Dubovik et al. (2000). For observations of dust aerosol, the effect of biases in AOD are significantly smaller than for BB. It can be explained

since dust has a small value of Ångstrom exponent and therefore larger values of AOD at longer wavelengths. For the retrieval of the size distribution, the bias in AOD has rather minor effect, though we found a general overestimation for positive BIAS values and underestimation for negative values.

Overall, the estimated systematic error agrees well with actual manifestations of the bias in the retrieval. The quantitative estimations are also quite convincing and shown in Figs. 20 and 22 for biases of $\pm 0.01$. In some cases, some underestimations

of the bias effects can be observed. For example, the largest differences are identified for the case of higher value of bias ($\pm 0.02$) shown in Figs. 21 and 23 while a significant increase in the systematic component of the retrieved error is also well captured by the error estimates.

It can be seen that among all considered aerosol parameters, the main differences between the bias effects and the obtained error estimates are observed for Real part of Refractive Index (RRI). In these cases, the bias is not fully covered by the

755 systematic component of the retrieved error results. Similarly, apparent underestimation of RRI errors was also seen by Sinyuk et al. (2020), who attributed these underestimations to different factors as, for example, the effect of not accounted pointing bias. In the present simulations, there is no pointing bias considered, and discrepancy is likely coming from the fact that Eqs. 24 and 31 relies on the derivatives estimated in the vicinity of the solution and based on linear approximation. Indeed, the dependence of both AOD and radiances scattered by aerosol is very complex and non-linear. Therefore, both taking the

760 derivatives in the vicinity of obtained solution instead of vicinity of 'true values', as well as, non-liner character of AOD and radiances may explain the differences. At the same time, it is important to note that, as can be seen from analysis of the random component of RRI error (Section 4.1) the random error effect is likely to dominate over effect of AOD bias, and therefore, the estimation of the total error (described below in this section) seems to allow us to make an objective and complete observation on this parameter.

Figures 24 to 27 show the effects of the biases in the radiances of two magnitudes of $\pm 3\%$ and $\pm 5\%$ for the observations of BB and dust. In general, it can be seen from the results that in both cases, BB and dust, the retrievals are less affected by bias in radiances than by the biases in the AOD, even when the bias in radiances is $\pm 5\%$. Similar tendency was also reported in studies by Dubovik et al. (2000), Torres et al. (2014) and Sinyuk et al. (2020). At the same time, it should be noted that the present analysis is focused on measurement configuration corresponding to solar almucantar the SZA is 75 degrees (see





Table 1) when the measurements include are taken in a wide range of scattering angles. In this respect, Dubovik et al. (2000) showed that the effect of sky radiance bias increases when the range of observed scattering angles is limited, i.e. in almucantar observation corresponding to SZA less than 60 degrees. Moreover, according to recent tendencies in observational practices, the use of such measurements is limited and most analyses are focused on observational scenarios with sufficient range of observed scattering angles. For example, AERONET start to establish so-called 'hybrid' observational scenario during high

SZA times (Giles et al., 2019). In regard to the performance of the error estimation, the effect of the bias in the sky radiances seems to be well captured by the GRASP error estimates.

It should be also noted that we observe an anticorrelation between the radiances BIAS and the retrieval of the imaginary part of the refractive index. This effect is opposite to the one observe in the case of AOD and with differences significantly smaller. Thus, when the BIAS are positive ($+3\%$ and $+5\%$) there is a decrease in the imaginary part of the refractive index. The fact

that the value of AOD remains the same and there is an increase in the value of the scattering is interpreted by the code as a decrease of the aerosol absorption. Conversely, the negative BIAS in the radiances produce an increase in the imaginary part of the refractive index which can be explained by the same reason.

Figure 28 shows the results of the analysis of the situation when the systematic biases present in both AOD and radiances are simultaneously assumed. The results for BB are on the left and for Dust on the right. The illustrations are shown for the specific

situation with two positive biases: $+0.01$ in AOD and $+5\%$ in radiances. It should be noted, that the tests were produced for both the situations with the biases of the same and opposite signs. This case with the biases of the same signs showed the most interesting results with the strongest manifestation of bias effects and therefore they are presented here. In the situation with biases of opposite signs effects on the retrievals are rather minor due to internal compensations of the influences of the biases. Additionally, the misfit of observations is more pronounced that helps to identify the issues and account for the biases

in the error estimates. Also, the analysis here is focused on the simultaneous biases of moderate values ($\pm 0.01$ in AOD and $\pm 5\%$ in radiances), since the appearance of simultaneous biases, of the highest bias values (i.e., $\pm 0.02$ in the AOD and $\pm 5\%$ in radiancies), lead to very strong effects in the retrievals. Those situations can be easily seen and screen out by quality filters (e.g., by high value of misfit). Also, it is quite unlikely to have such strong systematic errors in practical observations as those by AERONET.

As can be seen from Fig. 28 the biggest differences and highest bias values in the retrieval are found for low AOD (0.1). As seen earlier for this situation the errors for the RRI remain notably underestimated. As already mentioned, the situation is expected to be improved once the effects of both random and systematic errors are considered.

Figure 29 illustrate such situation for the retrieval of BB and dust for the three different aerosol loads (0.1, 0.3 and 0.6) when total error estimate includes both random and systematic components as:

$$\sigma_{tot} = \sqrt{\sigma_{ran}^2 + \sigma_{bias}^2} \qquad (36)$$

where $\sigma_{bias}^2$ is calculated as was indicated in Eq. 35.





It can be seen that the total error estimates capture the deviations for all parameters in the presence of random and systematic noises. These results confirm that the estimations using Eq. 31 based on the additional assumptions of potential presence of bias in the measurements improve the results of error estimated compare to the approach discussed in Section 4.1 when the effects of biased were taken into account only based on the value of the observation misfit. The observed tendencies in the effects of biases on the retrieval are consistent with all the results previously described in earlier studies. The obtained results are expected to be representative for most of practical situations, while some additional tests and analysis could certainly be useful. Therefore, in the examples presented below and for the real cases analyzed, the total error will be used as described in Eq. 36. It means, the representation of the error will take into account the contribution of the random and systematic component. This last component contains the contribution of Levenberg-Marquardt and the misfit and the measurements, in which the contributions of $\pm$ bias added in the measurements are considered (Eq. 35). These values of the assumed biases in our applications are consistent with AERONET as was aforementioned: $\pm 0.01$ in AOD and $\pm 5\%$ in radiances.

### 4.2.2 Effects of measurement bias in retrieval of mixed aerosol properties from measurements of radiometer only

The present section tries to understand how the bias affects when inhomogeneous aerosol are observed. The example is not commonly considered in practical application, e.g. in AERONET operational processing. At the same time, since GRASP can consider this type of bi-component inversion that are fundamentally of high interest, we are analyzing this situation in the presence of biases. In the Section 4.2.1 we have shown different examples, considering bias in each measurement separately, and we have also illustrated the complete example with presence of both random errors and bias in all the measurements. Here we illustrate directly the results considering the presence of both random and bias in all the measurements since this is complete situation that is most close to the most practical situation.

Different tests were performed for this study. In particular, we focus on the case of BB-Dust since the Section 4.2.1 has already demonstrated the bias affects when each type of aerosols is observed separately. The effects of each bias separately were analyzed while the corresponding illustrations are not shown since the results presented similar tendencies to those previously discussed are observed for observation of each type of aerosol separately in the last section. Figure 30 illustrates the examples of BB-Dust when $\tau(440) = 1.0$, for different aerosol loads ($\tau_f = 0.8$ and $\tau_c = 0.2$, $\tau_f = \tau_c = 0.5$ and $\tau_f = 0.2$ and $\tau_c = 0.8$) assuming bias and random noises. The shaded areas represent the estimated total errors, as shown in Equation 36. An important observation is that the error estimates for all retrieved and derived parameters well characterize the actual errors. As it can be gained from the figure, the retrieval of the properties of minor component appears as the most challenging. As a matter of fact, the biggest errors in the retrieval are observed for the fine mode properties, particularly, in the case of $\tau_f = 0.2$ and $\tau_c = 0.8$. The largest discrepancies of estimated errors with the actual ones are observed in this situation. On the other hand, the properties of coarse mode are well represented in almost all cases showing a good accuracy compared to the properties of the fine mode even in the most challenging cases with the smallest presence of coarse mode. This can probably be explained by the fact that desert dust AOD has rather moderate spectral changes.

On the other hand, in this section are also provided some illustrations for the lidar ratio in order to demonstrate how the retrievals and the error estimates are affected by the bias in the measurements. Figure 30 illustrates the lidar ratios in this





situation of mixed aerosol. The retrieval results and estimation of LR errors are rather satisfactory, with exceptions of low AOD cases, mainly in the case where the fine mode has only very minor presence (of $\tau_f = 0.2$ and $\tau_c = 0.8$). Also, it should be emphasize that the error estimated for total SSA and RI are rather adequate while in the Section 4.1, where only random errors were considered, the results showed some apparent underestimations.

### 4.2.3 Effects of measurement bias in retrieval of mixed aerosol properties from observations of radiometer in combination with lidar

This section considers the same example as in Section 4.2.2 and analyses the effects of measurement biases in the synergy retrieval using co-incident measurements sun/sky photometer and lidar measurements. At the same time, the results are presented for most practical situation when both random and biased are present in measurements and accounted in the error estimates.

Figure 31 shows the results for the example of BB-Dust described in the previous section for $\tau(440) = 1.0$ and assuming the presence of both bias and random noises in all the synthetic measurements for AOD, radiometer and lidar. As can be seen, the results of these different tests illustrate the positive influence of using radiometer and lidar synergy. The error estimates seem to be rather accurate too. For example, the most notable enhancement is in the lidar ratio accuracy especially when the mode fine is the smallest, i.e. for the case with: $\tau_f = 0.2$ and $\tau_c = 0.8$. This behavior was also seen by Lopatin et al. (2013)

who explained that these were expected results since lidar ratio has a high sensitivity to lidar signal. Nevertheless, we can see some improvements in retrieval of complex refractive index using both lidar and photometer data.

    In regard to the accuracy of the error estimation, in the Section 4.1, we have illustrated the retrieval of error estimates for LR and showed some apparent underestimation when only random errors were considered. Figure 31 illustrates an important improvement in the estimation of the errors, once both random noise and bias considered and Eq. 31 were used for accounting

the effect of the systematic component.

    Thus, using the synergy of both instruments can provide more accurate retrievals of LR and the error can be estimated rather accurately using the developed methodology for both aerosol components even for aerosol mode with the lower presence. Figure 32 illustrates the retrievals of the vertical aerosol profile in all three cases. The results show similar tendencies as in Section 4.1.

### 4.3 Illustration and description of the correlation matrices

The values of non-diagonal elements of covariance provide important and interesting information about the retrieved parameters. For example, if the values $\rho_{ii'} \neq 0$ are close to $\pm 1$ the similitude of the influences of the parameters $a_i$ and $a_{i'}$ on the inverted measurements $\mathbf{f}^*$ may explain the large variances of the retrieval error for these parameters. Also, knowledge about $\rho_{ii'} \neq 0$ is highly useful for the situation when several parameters from a set simultaneously retrieved parameters $a_i$ need

be jointly used in the applications. This can be easily seen from Eq. 29. For example, let us consider the estimates of two parameters $a_1$ and $a_2$ which have errors $\Delta a_1$ and $\Delta a_2$ characterized by covariance matrix:





$$\mathbf{C}_\Delta = \begin{pmatrix} \sigma_1^2 & \sigma_1\sigma_2\rho_{12} \\ \sigma_1\sigma_2\rho_{12} & \sigma_2^2 \end{pmatrix} \tag{37}$$

where $\mathbf{a}$ is a vector defined as $\mathbf{a} = (a_1, a_2)^T$. Correspondingly if in applications one needs to use the characteristics $m$ that is a liner function of $m = K_1 a_1 + K_2 a_2$, the variance $\sigma_m^2$ can be obtained from Eq. 29 as:

$$\sigma_m^2 = \mathbf{K}\mathbf{C}_{\Delta a}\mathbf{K}^T = \begin{pmatrix} K_1 & K_2 \end{pmatrix} \begin{pmatrix} \sigma_1^2 & \sigma_1\sigma_2\rho_{12} \\ \sigma_1\sigma_2\rho_{12} & \sigma_2^2 \end{pmatrix} \begin{pmatrix} K_1 \\ K_2 \end{pmatrix}$$

$$= K_1^2\sigma_1^2 + K_2^2\sigma_2^2 + 2K_1 K_2 \sigma_1\sigma_2\rho_{12} \tag{38}$$

From this equation, the importance of the correlation coefficient $\rho_{12}$ is quite evident. Specifically, if $\rho_{12} = 0$ then the variance $\sigma_m^2$ is just a simple sum $K_1^2\sigma_1^2 + K_2^2\sigma_2^2$. Therefore, the error propagation from $\Delta a_i$ to $\Delta m$ is straightforward, and namely only the values of sensitivities $K_i^2$ determine the contribution of $\Delta a_i$ (decreasing or increasing) to $\Delta m$.

When $\rho_{12} = 0$ the situation is more complex. However, the estimation of main tendencies can be simplified in some cases. For instance:

$$\text{if} \quad \sigma_1^2 = \sigma_2^2 \quad \text{and} \quad K_1 = K_2, \quad \text{then} \quad \sigma_m^2 = 2K_1^2\sigma_1^2(1 + \rho_{12}) \tag{39}$$

or

$$\text{if} \quad \sigma_1^2 = \sigma_2^2 \quad \text{and} \quad K_1 = -K_2, \quad \text{then} \quad \sigma_m^2 = 2K_1^2\sigma_1^2(1 - \rho_{12}) \tag{40}$$

From these equations, it can be seen that if correlation coefficient $\rho_{12} \to 1$ or $\rho_{12} \to (-1)$ then depending on the case in Eqs. 39 and 40 $\sigma_m^2$ can be close to zero or up $4K_1^2\sigma_1^2$. Therefore, the knowledge about the non-zero correlation coefficients $\rho_{ii'} \neq 0$ is very important to understand how the error is propagated to derived parameters obtained from the primary set $\mathbf{a}$.

In practical cases, when the derived parameter $m$ is a function of a large number parameters $a_i$, the contributions to $\sigma_m^2$ become increasingly very complex with the increase of the number of involved parameter $a_i$. Therefore, unfortunately, the general qualitative analysis, similarly to the one demonstrated by Eqs. 38-40, becomes very difficult and often practically impossible. Nonetheless, as it will be shown below the visualization of the correlation matrices Eq. 33 can be very useful for analysis of the retrieval tendencies.

### 4.3.1 Retrieval of single aerosol component from radiometer measurements

Figure 33 shows the correlation matrices of random retrieval errors for BB (spherical particles) and dust (non-spherical particles) for conventional AERONET-like inversion. The first 22 parameters (22x22) represent the SD. It is followed by two





blocks of 4x4. These two blocks are related to the RRI and IRI for 4 wavelengths ($440\,nm$ to $1020\,nm$). The last parameter is the sphericity fraction (1 x 1). The colors represent the values of correlation coefficients, where the red color denotes positive correlations, blue color indicates negative correlations. The density of the colors indicates values of the correlation coefficients changing from zero (the white color) to dense red or blue colors corresponding to values 1 and $-1$ accordingly.

The correlation for biomass burning case is shown in Fig. 33a. As it can be seen, in general size bins retrievals have rather
moderate correlation between them, though large positive correlations between the retrieval errors of neighbours can be observed. This is more evident for size bins at the smallest and largest particle sizes. This indicates that size distribution values for those size have tendency to be overestimated or underestimated together, which can be mostly explained by the use of typical smoothness constraints imposed to the size distributions. The errors of RRI and IRI are negatively correlated with the SD parameters. The correlations seem especially pronounced between RRI and values of size distribution for the fine mode.
Correspondingly the overestimations of size distribution values may tend to be accompanied by underestimation of RRI and vice versa. The errors of the fraction of spherical particle seem to show positive correlation with the SD retrieval errors. This correlation is more evident when there is a fine mode dominated aerosol, since scattering of fine mode particles has a quite similar shape for sphere and spheroids. Therefore when there is fine mode domination is more difficult to differentiate between sphere or spheroids. The positive sign of correlataion can be explained by the fact that extinction cross sections for the equiv-
alent radii are a bit higher for spheroids. Thus, a higher percentage of spheres can be optically compensated by an increase in the volume concentration, without a big impact in the total residual. The fraction of spherical particle shows a negative strong correlation with the errors of the refractive index. Strong positive correlations can be seen between spectral values of RRI. The positive correlation are presents but lower between spectral values of IRI. As notice already this likely relates with the use of rather strong smoothness constraint of spectral variability of RRI and weaker constraint of spectral variability of IRI in the
retrieval (Dubovik and King, 2000). The essential positive correlation can be noticed also between errors of RRI and IRI.

Figure 33b shows the correlation matrix for retrieval of dust aerosol. The structure of the correlation for SD exhibits some differences compare to the BB case. Specifically, the positive correlations between neighboring size bins for the smallest and largest particle sizes are even more pronounced. Also, somewhat stronger negative correlations can be seen the intermediate sized. The strong negative correlation between RRI and SD retrieval errors remains only between concentrations of very small
particles and values of RRI at shortest wavelengths. The notable positive correlation presents only between spectral values of RRI at the shortest wavelengths and between spectral values of RRI at the longest wavelengths. At the same time, overall the correlation of retrieval errors of both RRI and IRI between themselves and with other parameters decreases compared to the case of BB. The errors of the fraction of spherical particle for dust case correlate much less with the errors of the other parameter compared to BB case. This can be explained by the fact the light scattering of large particles is significantly more
sensitive to deviation of aerosol particles from spheres compared to spheroids, than the light scattering of fine fraction particles (Dubovik et al., 2006). Therefore, when coarse particles dominate the discrimination between spheres and spheroids becomes more evident.

Thus, the analysis of the correlation matrices itself provide very useful inside that helps to understand and interpret retrieval results. For example, such artifacts as appearance of 'tails' (unrealistically high concentrations) at extremes of size distribution



has been noticed and widely discussed (Dubovik et al., 2002a, 2006; Torres et al., 2014). Such retrieval artifacts as underestimation RRI accompanied by overestimation size distribution of very fine particles has been widely discussed in studies by Dubovik et al. (2000, 2002b, a). These artifacts were strongly reduced by accounting for particle non-sphericity of desert dust particles (Dubovik et al., 2002b, 2006), but nonetheless the less pronounced appearance of such artifacts remains in AERONET like retrieval (Torres et al., 2014, 2017). These artifacts are clearly related with the observed above presence of strong negative

correlation between values of RRI and size distribution of very fine particles. It should be noted that the presence of high correlations is an indication that adding information about one of the correlated parameters should improve retrieval not only the constrained parameter itself but also the parameters that strongly correlated with this parameter. For example, addition of polarimetric observations to the traditional set of AERONET observations results in clear improvement in the retrieval of RRI and size distribution of very fine particles (Li et al., 2009; Fedarenka et al., 2016). Indeed, the degree of linear polarization is

known to be very sensitive to amount and especially the RRI of fine particles (Dubovik et al., 2006). This is why, the addition of polarimetric observations help to reduce the correlations between the errors of RRI and size distribution of fine particles that helps to the overall improvement of the retrieval accuracy of these particles.

### 4.3.2 Retrieval of mixed aerosol component from measurements of radiometer only

In this section, the correlation matrix for bi-component aerosol retrieved from synthetic observations of sun/sky radiometer of

940 two aerosol mixtures is illustrated : BB-dust and Urban-dust. The structure of this matrix consists in 25 parameters related to the SD that are separated into two blocks: 10 parameters for fine mode and 15 parameters for coarse mode. The following 4 blocks of 4x4 are related to the RRI and IRI of fine and coarse modes at 4 wavelengths. These blocks are followed by a single value of the sphericity fraction.

The area of the correlation matrix that contains SD, RRI, IRI and sphericity fraction is quite similar to the correlation matrix

obtained for aerosol AERONET like retrieval. The main difference is the separation into two modes since strong negative correlations can be observed between the corresponding parameters of fine and coarse mode. For example, strong negative correlations can be observed between IRI fine and coarse. These correlations mean that overestimating the amount or absorption of one aerosol mode is likely compensated by underestimation of the amount or absorption of another aerosol mode. Another interesting anticorrelation can be observed for the last three bins of SD fine mode and the first three bins of SD coarse mode.

Actually, both volume distributions have these three bins in common. This overlap zone is never easy to properly separate for the code but at the same time, it coincides to a local minimum value of most size distributions found in the real retrievals.

### 4.3.3 Retrieval of mixed aerosol properties from measurements of radiometer in combination with lidar

Figure 35 shows the correlation matrix for the case when bi-component aerosol retrieved from synthetic observations sun/sky photometer and lidar of aerosol mixtures. In the figure the different blocks are identified. The first 25 parameters represent the

955 SD that are separated into two blocks: 10 parameters for fine mode and 15 parameters for coarse mode. The following 4 blocks of 7x7 are related to the RRI and IRI of fine and coarse modes at 7 wavelengths. These blocks are followed by a single value





of the sphericity fraction. Two last and largest blocks of 60x60 parameters each correspond to the AVP values of two modes given at 60 different altitudes.

The area of correlation matrix that contains SD, RRI, IRI and sphericity fraction is quite similar to the correlation matrix
described in previous section considered the aerosol mixture retrieval from AERONET like observation only.

As it could be expected, the block of the correlations of AVP retrieval shows strong negative correlations between errors of the retrieved parameters of fine and coarse mode. Thus, an overestimation of one mode is higly correlated with an underestimation of another AVP mode. Furthermore, an strong positive correlations can be observed between AVP values corresponding to the same fine or coarse mode; i.e. the AVP values of each mode have tendency to be simultaneously overestimated or un-
derestimated. This is related to the limited sensitivity of used lidar data for distinguishing the contributions of different modes, and also, to the use of smoothness constraints on vertical variations of AVP of each fraction. On the other hand, nearly zero correlation can be seen for AVP parameters at the altitudes with significant presence of one or both aerosol components. Correspondingly, there is a high sensitivity of both lidar and radiometer observation to the aerosol parameters at those altitudes. It should be noted that all above discussed retrieval suggested from the analysis of correlation matrices where actually observed
in the retrievals from real data as discussion by Lopatin et al. (2013, 2021).

### 4.4 Illustration of GRASP error estimates with real observations

This section illustrates the GRASP error estimates performance for the retrieval from real data. With that purpose the lidar and sun/sky photometer measurements collected at Aeroparque ($34° \, 33' \, 51''S$, $58° \, 25' \, 02''W$) and Villa Martelli ($34° \, 33'21''S$, $58° \, 30' \, 23''W$) stations, in Buenos Aires, Argentina have been used. These instruments are part of LALINET (Latin America
Lidar Network, Guerrero-Rascado et al. (2016)) and AERONET networks. Both sites are located in an industrialized city dominated by continental and urban/industrial aerosols, and during winter and spring are affected by biomass burning from north and center of the country and neighboring countries mainly Brazil. Aeroparque station is located at the airport Jorge Newbery within the limits of the city. This station does not have a co-located sun-photometer but its location is 7 km from the Villa Martelli station where the sun-photometer is installed. On the other hand, Villa Martelli station is found in the limit of
Buenos Aires city, in a highly populated and industrialized area.

The observation from two different biomass burning events in Argentina were selected for the illustrations. Specifically, three days were chosen with different aerosol loads and SZA. The lidar range-corrected signal (RCS) corresponding to each event are shown in Figure 36. They have been calculated from the lidar signal, with background and dark current correction, multiplied by the height squared. In addition, the back-trajectories calculated from the HYSPLIT (Hybrid Single-Particle Lagrangian
Integrated Trajectory; Stein et al. (2015); Rolph et al. (2017)) models are presented in order to confirm where the air masses come from (Fig. 37).

The two first cases selected correspond to an important event of biomass burning occurred in bordering countries in the north of Argentina in August 2014, particularly south of Brazil and Paraguay. It was detected in Buenos Aires, between 19 to 23 August. For the illustrations are used the measurements corresponding on August 19 which present low aerosol load at
$440 \, nm$ ($\sim 0.11$) and SZA $> 50$ degrees. Figure 36a indicated the presence of several layers of aerosols up to $1.5 \, km$. The



lidar measurements on August 19 were taken between 12:15 UTC and 12:45 UTC, and AERONET measurements correspond to 11:25 UTC. Figure 37 shows the HYSPLIT back trajectories that validate the source of the air masses. The measurements corresponding to August 22 are shown in Figure 36b, where several layers of aerosol up to 3 km are observed. For this day the aerosol load increases (AOD at $440\,nm \sim 0.31$) and the SZA is $< 50$ degrees. The inversion was realized with the average
lidar data between 15 UTC to 15:20 UTC, and the AERONET measurements were considered to 16:59 UTC. The satellite image corresponding to August 22 (Fig. 37a) shows the presence of aerosols that extend from north of Argentina towards the center, passing through the province of Buenos Aires. Moreover, MODIS hotspots are detected in the satellite image. The source of the air masses can be validated from the HYSPLIT back trajectories (Fig. 37c). The last selected case corresponds to the biomass burning event on 25 September 2017, occurred in the north of Argentina and bordering countries (Figure 36c). In
this work, lidar measurements from Aeroparque station between 19:20 UTC to 20:10 UTC, and the AERONET measurements corresponding to 19:20 UTC, whose AOD value at $440\,nm$ is 0.57 and SZA $> 50$ degrees were used.

Figure 38 illustrate the retrieved columnar properties for each day obtained by GRASP from a combination of radiometer and lidar data and the comparison with the corresponding standard AERONET retrievals. The results provided by GRASP are represented in solid lines: blue for the fine mode and green the coarse mode. Shaded areas represent the error provided
by GRASP for each retrieved and derived property. Zoomed plots represent the effective refractive index and total SSA for GRASP (black solid line) and AERONET (black dashed line). From the illustrations can see almost all the GRASP retrieved properties in the three cases present good agreement with AERONET retrievals.

The error tendencies for SD that can be seen from Figure 38 agree with those identified above in present study and with results presented in previous sections and also with the results in some works as Dubovik et al. (2000) and Lopatin et al.
(2013). For example, the retrieval errors clearly increase at the extremes of SD. Moreover, one clear and known tendency can be mentioned. The size distribution shift towards higher radii in the three cases could be explained by the use of lidar data in the inversions that provide additional information at scattering angles of 180 degrees (Lopatin et al., 2013; Bovchaliuk et al., 2016; Benavent-Oltra et al., 2017). As it can be seen in the Figure 38, these deviations in almost all cases are included in the error (shaded areas).

The errors of RRI, IRI and SSA were retrieved for each mode separately by GRASP and they are significantly higher than the error for RRI, IRI and SSA of the total components. The effective RRI and IRI and the total SSA obtained by GRASP are in the middle of the retrieved values for fine and coarse mode separately. The total values showed in the zoomed plots agree well with RRI, IRI and SSA provided by AERONET.

On the other hand, the case corresponding to September 25 has an AOD $> 0.4$ and SZA $> 50$ degrees allowing us to have the
1020 uncertainty of the SSA provided by AERONET. Thus, in the Figure 38 for this particular case we can observe the comparison of the uncertainty of SSA from AERONET and the SSA error provided by GRASP. Note, it is one advantage of GRASP that provide the errors for each parameter in all the situations. Regarding the values of SSA and the tendencies of their variability, the results show SSA representative of biomass burning cases. Namely the values of SSA decreasing with the wavelength agree with AERONET climatology by Dubovik et al. (2002a). As expected, based on the results of previous studies by Dubovik et al.
(2000), Lopatin et al. (2013) and Tsekeri et al. (2017) the best agreements are obtained as the aerosol load increases. More





specifically, we observe the estimated errors of total SSA, in the two first cases increase. This can be associated with not favorable configurations of observation, on August 19 the AOD at $440\,nm$ is $\sim 0.11$ (SZA is $> 50$ degrees) and on August 22 the AOD at $440\,nm \sim 0.31$ (SZA is $< 50$ degrees). Thus, the measurements in situations with low amount of aerosol and with small SZA may not contain enough information to adequately retrieve the SSA (Dubovik et al., 2000; Lopatin et al., 2013;

Torres et al., 2014)). Therefore, the case of September 25 corresponds to the most favorable situation for realizing reliable aerosol retrieval since AOD at $440\,nm$ value is $> 0.4$ and SZA is $> 50$ degrees. Indeed, the GRASP and AERONET retrievals have the best agreement for this day.

Figure 39 shows the retrieved vertical distributions of fine and coarse modes. The vertical structure of the aerosols of different types is clearly discriminated and shows good agreements with the back trajectory analysis for each day. Furthermore, the error

estimates show good agreements with previous results provided in last sections for simulated cases.

Moreover, Figure 40 show the retrieved LR and their error estimates using GRASP algorithm. The zoomed plots show the total LR provided by GRASP (black solid line) and by AERONET (black dashed line). The associated errors are represented in shaded areas for GRASP. As can be see only is possible to compare the values with AERONET retrieval for the Fig.40 which corresponds to higher aerosol load and SZA$> 50°$ on September 25, 2017.

Thus, the retrieved parameters and error estimates from GRASP application to the real data and their comparisons to the AERONET retrieval results showed an encouraging agreement between columnar properties of aerosol. At the same time, GRASP provide the error estimates for the retrieved properties in both fine and coarse mode and also for the total components. Moreover, GRASP has also the advantage of provide the dynamic error estimates in all the configurations. As was seen AERONET error estimates are only provided in some particular situations, when the AOD at 440 is greater than $0.4$ and

SZA$> 50$ degrees.

## 5 Conclusions

In this work we reviewed the approach realized in GRASP algorithm for estimating error of the parameters retrieved from remote sensing observations. The employed approach relies on rigorously realized concept of statistical estimations and tends to account for the propagation of both random and systematic errors. Then we evaluated the performance of the GRASP error

estimates for aerosol parameters retrieved from ground-based observations. We considered AERONET like retrievals from observations by Sun/sky-scanning radiometer and GRASP synergy aerosol retrieval from joint observations by radiometer and lidar. GRASP generates the full covariance matrices that expected to be used for generating error bars for retrieved parameters and provide an interesting inside for understanding retrieval tendencies. Therefore, we studied the quantitative reliability of the obtained covariance diagonal elements and analyzed the structure of correlation coefficient of covariance matrices.

The performance of GRASP estimates of error variances in presence of random errors was evaluated in series of numerical tests and illustrated the capabilities of GRASP algorithm to provide rigorous estimates of dynamic retrieval errors. In frame of these tests the synthetic proxy observations perturbed by 300 random noise generated realization were inverted using GRASP algorithm. Then, retrieved parameters were compared to those used for generation of the synthetic data and the obtained error





estimates were compared with actual deviations of the retrieved parameters from assumed values. This analysis was realized
for synthetic observations for three different types of aerosols, as well as, for the mixture of them. Observations of Dust were
modeled using AERONET retrieval climatology at Solar Village (Saudi Arabia) site. The AERONET retrieval climatologies
from African savanna (Zambia) and the GSFC (Maryland, USA) were used simulating Urban and BB aerosol observations
respectively. The Urban-Dust aerosols and BB-Dust mixture were considered for modeling properties of mixed aerosols. For
each observed aerosol type or mixture of them different aerosol loads were tested. First, we modelled observation of aerosols of
only one type aerosol for $\tau(440) = 0.3$, $0.6$ and $0.9$. For aerosol mixtures we also considered scenarios with different aerosol
loads, while we present most of the illustrations for the situation with the large aerosol load ($\tau(440) = 1.0$, combining different
aerosol loads for each mode $\tau_f = 0.2$ and $\tau_c = 0.8$, $\tau_f = \tau_c = 0.5$ and $\tau_f = 0.8$ and $\tau_c = 0.2$). The data were perturbed by
random noise before applying the retrieval algorithm. For all these simulations it was used the SZA at 75 degrees.

The tests evaluated the situations when only radiometer data were inverted and then radiometer data were inverted jointly
with co-incident lidar data. Two GRASP retrieval set ups were tested: (i) then the retrieval assumes that aerosol is composed
by homogeneous particles and parameters of only one aerosol component are retrieved and (ii) then the aerosol is assumed
as external mixture of two aerosol components and parameters of each component are retrieved separately. In case the when
lidar data were used the vertical profiles of concentration were also retrieved for each aerosol component. The illustrations
for aerosol retrieval from all simulated data sets were provided using both approaches. However, the full statistical analysis
was provided only the two conventional retrieval scenarios: - AERONET like single component retrieval from radiometric
observations and - GRASP bi-component aerosol retrieval from combinations of radiometer and lidar data.

The results of the tests showed that the complete set of aerosol parameters for each aerosol component can be robustly
derived with acceptable accuracy in almost all considered situations. The retrieval of bi-component aerosol was evaluated
using radiometer only simulated measurements and then by adding lidar observations. These tests allowed us to observe that
by using the synergy of two instruments, some improvements in retrieval of aerosol properties of each component of observed
aerosol mixture and in the estimations of the retrieval errors. The test for selected cases with different presence of different
aerosol components ($\tau_f = 0.2$, $\tau_c = 0.8$ and $\tau_f = 0.8$, $\tau_c = 0.2$) showed that optical properties of the dominant mode can be
retrieved significantly more accurately as can be expected. It is interesting to note that in all situation using only a radiometer
data or adding lidar simulated measurements, such properties as total SSA and effective refractive index can be retrieved rather
accurately, even in cases where the retrieval of properties of each mode separately is questionable.

The results of the statistical tests with randomly generated noise showed that GRASP error estimates in most cases are
comparable or exceed the actual errors by the 20 to 30% and therefore can be safely used for assuring uncertainties of actual
retrieval products. In addition, the observation of typical error values was summarized for different situations and retrieval
scenarios. Namely, the study confirmed that the detailed properties of aerosol mixtures can be rather reliably retrieved from a
combination of radiometer and lidar data provided that there is sufficient amount of both aerosol components. For example,
for the case when total $\tau(440) = 1.0$ with comparable presence of both components $\tau_f(440) = \tau_c(440) = 0.5$, and SZA is 75
degrees, the mean values for RRI errors are $\sim 0.05$ for BB and Urban and vary between 0.07 to 0.09 for Dust, IRI errors are
around 60% for BB, 75% for Urban and 50% for Dust. SSA errors vary between 0.024 to 0.061 for Urban, 0.042 to 0.086 for



BB and 0.04 to 0.009 for Dust, showing a clear tendency to increase with the wavelengths for values of fine mode and decrease for coarse mode. However, even for this case the separation of the LR values for both modes showed high uncertainties at short wavelengths in particular for the fine mode, while the values of the total LR errors were found reasonable in the range of $20\%$ to $55\%$. The relative error estimates for AVP for both aerosol mixtures (Urban-Dust and BB-Dust) cases where ranging for fine mode vary between $50-70\%$ and for coarse Dust mode between $50-55\%$.

The effects of the systematic errors on the retrievals were also analyzed in a series of limited dedicated numerical tests. The results of the tests were used for adjusting the GRASP retrieval estimates to the potential effects of the systematic errors. The results show enhancements of total error estimates from the assumption of bias in the equation of systematic components.

In addition, to the evaluation of error bars estimates and effects of systematic errors, in this paper we illustrated and discussed the correlation structures of the error covariance matrices for all main considered retrieval scenarios. The results showed that analysis of the correlation structure can be very useful for understanding the observed retrieval tendencies and optimizing retrieval. For example, for conventional AERONET like aerosol retrievals from radiometer data only, the strong negative correlation between errors of real part of the refractive index and size distribution values for small sizes. This suggests agrees well with the tendency commonly observed in actual retrieval when the underestimations of the real part are coincident with overestimation of fine mode size distribution. Also, the presence of high positive correlation between the errors of size distribution for extreme sizes and between the errors of refractive index at different wavelengths agrees well with known possibilities of possible overestimations of aerosol concentrations for very small or very large particles and joint overestimations/underestimations of refractive value at different wavelengths. For bi-component retrievals strong negative correlations can be observed between nearly all corresponding parameter of fine and coarse mode. This means, for example, that the overestimation of amount or absorption of one aerosol mode is likely compensated by underestimation of amount or absorption of another aerosol mode. The decrease of some of these correlations was observed when inverted radiometer data were inverted simultaneously with the lidar data. The high positive correlations were seen for the errors of the vertical profile of the fine and coarse concentrations with the exception of the values for the altitudes where one or both of aerosol modes had substantial loads. These and other less obvious, but quite interesting, correlations structures and tendencies can be identified using the analysis of the correlation matrix structure. Thus, the availability and analysis of not only error variances but also correlation patterns appear to be a useful and promising approach for optimizing observation schemes and retrieval setups.

Finally, the utilization of GRASP for deriving detailed aerosol properties and estimation of their errors was demonstrated for the coincident lidar and sun photometer observations from Buenos Aires, Argentina. The GRASP retrievals and the error estimates of the columnar aerosol properties showed to be fully adequate in comparative analysis with the aerosol products available from AERONET operational retrievals. The retrieval of vertical profiles of fine and coarse aerosol modes showed results consistent with the expectation and the predictions of back-trajectories analysis.

Thus, results presented in this work showed a promising potential for utilization GRASP retrieved dynamic error estimates for the detailed retrieved aerosol parameters from measurements of ground-based radiometers and lidars considering different geometries and presence of diverse aerosol loads. These studies are expected to be completed in future by more extensive analysis of the error estimates for the such detailed parameters as vertical profiles of SSA and LR.



*Code availability.* GRASP is an open-source software, available upon registration from https://access-request.grasp-cloud.com/service/
gitlab (GRASP Open repository, 2022)

*Data availability.* Data available upon request. AERONET data used for this study can be downloaded from NASA website https://aeronet.
gsfc.nasa.gov/. LALINET data are available upon request.

*Author contributions.* OD contributed to the development of the overall algorithm methodology, research planning and article writing. BT
contributed in the results discussions and the article writing. BT, TL, DF, AL, PL, CC, MEH contributed to the algorithms designs, devel-
opment and software support. PL, CC, JABO, JLB contributed in the results discussions and polishing the article text. PR contributed to
the data preparation and analysis. MEH prepared the manuscript including the co-authors contributions and performed the main part of the
presented research.

*Competing interests.* The authors declare that they have no conflict of interest.

*Acknowledgements.* The authors acknowledge the support from the H2020 Marie Skłodowska-Curie RISE Actions (grant no. 778349) and
the authors acknowledge the support from the European Metrology Program for Innovation and Research (EMPIR) within the joint re-
search project EMPIR 19ENV04 MAPP 'Metrology for aerosol optical properties'. The EMPIR is jointly funded by the EMPIR partici-
pating countries within EURAMET and the European Union. T. Lapyonok was supported from the Chemical and Physical Properties of
the Atmosphere Project funded by the French National Research Agency through the Programme d'Investissement d'Avenir under contract
ANR-11-LABX-0 0 05-01, the Regional Council 'Hauts-de-France', and the 'European Funds for Regional Economic Development', from
Research grant (no. URF/1/2180- 01-01) 'Combined Radiative and Air Quality Effects of Anthropogenic'. The authors also acknowledge to
AERONET and LALINET for the scientific and technical support. We acknowledge the use of imagery from the NASA Worldview applica-
tion (https://worldview.earthdata.nasa.gov), part of the NASA Earth Observing System Data and Information System (EOSDIS). The authors
also would like to acknowledge the NOAA Air Resources Laboratory (ARL) for the provision of the HYSPLIT transport and dispersion
model (https://www.ready.noaa.gov) used in this publication.



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





**Table 1.** Summary of general input data and the set of parameters retrieved by GRASP algorithm used in this work for two configurations: Sun/sky radiometer only and Sun/sky radiometer plus lidar.

| Sun/sky radiometer Only | Sun/sky radiometer plus lidar |
|---|---|
| *Input:* | *Input:* |
| | Sun/sky-radiometer data: |
| - AOD[*] | - AOD[*] |
| - Calibrated radiances[*] | - Calibrated radiances[*] |
| | Lidar data: |
| | - Range corrected profiles (RCS[**]) normalized |
| | at 60 log-spaced bins at different heights |
| [*] at $440\,nm$, $675\,nm$, $870\,nm$ and $1020\,nm$ | [**] at $355\,nm$, $532\,nm$ and $1064\,nm$ |
| *Retrieved aerosol properties in the total atmospheric column*: | *Retrieved aerosol properties (Column-integrated and vertical distributions)*: |
| - $dV(r_i)/d\ln r_i$ | - $dV(r_i)/d\ln r_i$ (in total atmospheric column) |
| - $C_{sph}$ | - $C_{sph}$ (in total atmospheric column) |
| - $n(\lambda_i)$ | - $n(\lambda_i)$ (in total atmospheric column) |
| - $k(\lambda_i)$ | - $k(\lambda_i)$ (in total atmospheric column) |
| | - $C_V(h)$ (vertical distribution) |
| *Retrievals provided for total, fine and coarse modes.* | *Retrievals provided for total, fine and coarse modes.* |

[*] Azimuth angles, for sky radiances in the almucantar geometry, relative to sun (in degrees): 3.0, 3.5, 4.0, 5.0, 6.0, 7.0, 8.0, 10.0, 12.0, 14.0, 16.0, 18.0, 20.0, 25.0, 30.0, 35.0, 40.0, 45.0, 50.0, 60.0, 70.0, 80.0, 90.0, 100.0, 110.0, 120.0, 140.0, 160.0, 180.



**Table 2.** Errors provided by GRASP for the RRI, IRI and SSA are represented by the mean values of each boxplot for their respective wavelength. Absolute errors are given for RRI and SSA, and relative errors for IRI. Mean values of actual errors are provided in parenthesis.

|  | **BB** | | | **Urban** | | | **Dust** | | |
|---|---|---|---|---|---|---|---|---|---|
|  | **RRI** | **IRI[%]** | **SSA** | **RRI** | **IRI[%]** | **SSA** | **RRI** | **IRI[%]** | **SSA** |
| **440** | 0.079 | 24.0 | 0.028 | 0.056 | 54.1 | 0.013 | 0.201 | 24.4 | 0.014 |
|  | (0.014) | (6.66) | (0.005) | (0.016) | (17.6) | (0.004) | (0.03) | (11.52) | (0.006) |
| **675** | 0.082 | 10.9 | 0.033 | 0.053 | 33.3 | 0.014 | 0.17 | 24.2 | 0.008 |
|  | (0.018) | (7.8) | (0.005) | (0.015) | (19.8) | (0.005) | (0.013) | (11.36) | (0.004) |
| **870** | 0.084 | 11.29 | 0.042 | 0.051 | 31.55 | 0.017 | 0.17 | 28.2 | 0.008 |
|  | (0.019) | (11.61) | (0.009) | (0.015) | (28.57) | (0.009) | (0.011) | (14.3) | (0.003) |
| **1020** | 0.081 | 13.17 | 0.043 | 0.052 | 35.1 | 0.021 | 0.167 | 30.4 | 0.007 |
|  | (0.017) | (13.52) | (0.014) | (0.015) | (27.3) | (0.011) | (0.011) | (11.7) | (0.002) |





**Table 3.** The mean values of RRI, IRI, SSA and LR retrieval errors estimated by GRASP for the synthetic test for a mixture of Urban-Dust aerosol mixture. The mean values represent distributions obtained using 300 realizations of added random errors for the situation with total $\tau(440) = 1.0$ with $\tau_f = \tau_c = 0.5$ and SZA $= 75°$. The absolute errors are provided for RRI and SSA, and relative errors for IRI and LR. Mean values for the actual errors are provided in parenthesis.

| | 355 nm | 440 nm | 532 nm | 675 nm | 870 nm | 1020 nm | 1064 nm |
|---|---|---|---|---|---|---|---|
| | **Urban-Dust** | | | | | | |
| **RRI$_f$** | 0.050 | 0.049 | 0.048 | 0.045 | 0.045 | 0.045 | 0.045 |
| | (0.033) | (0.030) | (0.029) | (0.031) | (0.036) | (0.036) | (0.036) |
| **RRI$_c$** | 0.076 | 0.073 | 0.071 | 0.080 | 0.085 | 0.086 | 0.086 |
| | (0.028) | (0.028) | (0.028) | (0.021) | (0.017) | (0.015) | (0.015) |
| **IRI$_f$[%]** | 73.31 | 71.18 | 70.67 | 70.33 | 71.34 | 72.18 | 72.26 |
| | (103.03) | (103.4) | (94.19) | (81.46) | (76.62) | (76.78) | (76.79) |
| **IRI$_c$[%]** | 50.59 | 45.00 | 44.58 | 45.12 | 49.01 | 51.22 | 51.51 |
| | (36.17) | (26.45) | (22.32) | (18.50) | (18.23) | (17.92) | (17.89) |
| **SSA$_f$** | 0.024 | 0.026 | 0.029 | 0.033 | 0.045 | 0.057 | 0.061 |
| | (0.017) | (0.019) | (0.020) | (0.022) | (0.027) | (0.034) | (0.037) |
| **SSA$_c$** | 0.039 | 0.031 | 0.024 | 0.015 | 0.011 | 0.009 | 0.009 |
| | (0.042) | (0.025) | (0.014) | (0.007) | (0.004) | (0.003) | (0.003) |
| **SSA$_T$** | 0.017 | 0.015 | 0.014 | 0.010 | 0.008 | 0.008 | 0.008 |
| | (0.009) | (0.004) | (0.004) | (0.005) | (0.004) | (0.003) | (0.003) |
| **LR$_f$[%]** | 14.93 | 12.11 | 10.41 | 8.88 | 8.92 | 9.79 | 9.96 |
| | (40.45) | (29.39) | (15.37) | (7.58) | (8.44) | (8.96) | (8.63) |
| **LR$_c$[%]** | 37.04 | 35.71 | 35.23 | 42.39 | 47.77 | 48.99 | 49.32 |
| | (17.42) | (12.01) | (9.48) | (11.81) | (14.90) | (15.17) | (14.99) |
| **LR$_T$[%]** | 23.31 | 26.46 | 28.21 | 35.61 | 41.45 | 43.36 | 43.97 |
| | (9.01) | (7.48) | (6.85) | (9.71) | (13.3) | (13.9) | (13.9) |





**Table 4.** The mean values of RRI, IRI, SSA and LR retrieval errors estimated by GRASP for the synthetic test for a mixture of BB-Dust aerosol mixture. The mean values represent distributions obtained using 300 realizations of added random errors for the situation with total $\tau(440) = 1.0$ with $\tau_f = \tau_c = 0.5$ and SZA = 75°. The absolute errors are provided for RRI and SSA, and relative errors for IRI and LR. Mean values for the actual errors are provided in parenthesis.

| | BB-Dust | | | | | | |
|---|---|---|---|---|---|---|---|
| | 355 nm | 440 nm | 532 nm | 675 nm | 870 nm | 1020 nm | 1064 nm |
| **RRI**$_f$ | 0.052 | 0.050 | 0.048 | 0.044 | 0.045 | 0.046 | 0.046 |
| | (0.064) | (0.067) | (0.061) | (0.052) | (0.046) | (0.047) | (0.047) |
| **RRI**$_c$ | 0.084 | 0.083 | 0.079 | 0.085 | 0.089 | 0.091 | 0.091 |
| | (0.019) | (0.019) | (0.019) | (0.017) | (0.016) | (0.014) | (0.014) |
| **IRI**$_f$**[%]** | 60.51 | 57.61 | 57.91 | 58.17 | 60.33 | 61.48 | 61.59 |
| | (35.65) | (35.40) | (37.56) | (41.63) | (43.74) | (44.14) | (44.13) |
| **IRI**$_c$**[%]** | 48.75 | 44.57 | 43.81 | 43.89 | 47.43 | 49.56 | 49.87 |
| | (33.75) | (41.39) | (48.20) | (60.36) | (62.61) | (55.55) | (55.44) |
| **SSA**$_f$ | 0.041 | 0.044 | 0.048 | 0.052 | 0.067 | 0.085 | 0.089 |
| | (0.031) | (0.031) | (0.033) | (0.040) | (0.050) | (0.057) | (0.059) |
| **SSA**$_c$ | 0.049 | 0.040 | 0.032 | 0.021 | 0.015 | 0.013 | 0.013 |
| | (0.032) | (0.031) | (0.027) | (0.019) | (0.013) | (0.010) | (0.009) |
| **SSA**$_T$ | 0.022 | 0.016 | 0.016 | 0.011 | 0.010 | 0.009 | 0.009 |
| | (0.011) | (0.004) | (0.005) | (0.004) | (0.003) | (0.003) | (0.003) |
| **LR**$_f$**[%]** | 17.99 | 15.17 | 12.53 | 10.74 | 11.75 | 13.48 | 13.82 |
| | (33.43) | (16.46) | (11.29) | (18.05) | (21.76) | (21.43) | (21.30) |
| **LR**$_c$**[%]** | 49.38 | 48.73 | 47.91 | 54.82 | 58.57 | 59.54 | 59.76 |
| | (17.24) | (13.65) | (11.60) | (15.04) | (19.94) | (22.16) | (22.23) |
| **LR**$_T$**[%]** | 30.30 | 34.11 | 36.09 | 44.35 | 50.82 | 53.51 | 54.18 |
| | (8.83) | (9.03) | (10.52) | (15.65) | (20.38) | (22.13) | (22.14) |





**Table 5.** The mean values of AVP retrieval errors estimated by GRASP for the synthetic test for a mixture of Urban-Dust aerosol mixture. The mean values represent distributions obtained using 300 realizations of added random errors for the situation with total $\tau(440) = 1.0$ with $\tau_f = \tau_c = 0.5$ and SZA $= 75°$. The shown relative errors for AVP [1/km] are represented by the mean values for three layers. Mean values for the actual errors are provided in parenthesis.

|  |  | until 1.5 km | 1.5-3.5 km | above 3.5 km |
|---|---|---|---|---|
| **Urban-Dust** | **AVP$_f$ [%]** | 61.91 | 66.90 | 59.95 |
|  |  | (25.79) | (27.73) | (22.06) |
|  | **AVP$_c$ [%]** | 57.25 | 54.91 | 56.34 |
|  |  | (14.43) | (4.11) | (23.51) |
| **BB-Dust** | **AVP$_f$ [%]** | 61.46 | 68.50 | 59.17 |
|  |  | (25.80) | (30.53) | (22.28) |
|  | **AVP$_c$ [%]** | 55.74 | 53.90 | 55.30 |
|  |  | (14.36) | (4.08) | (22.91) |





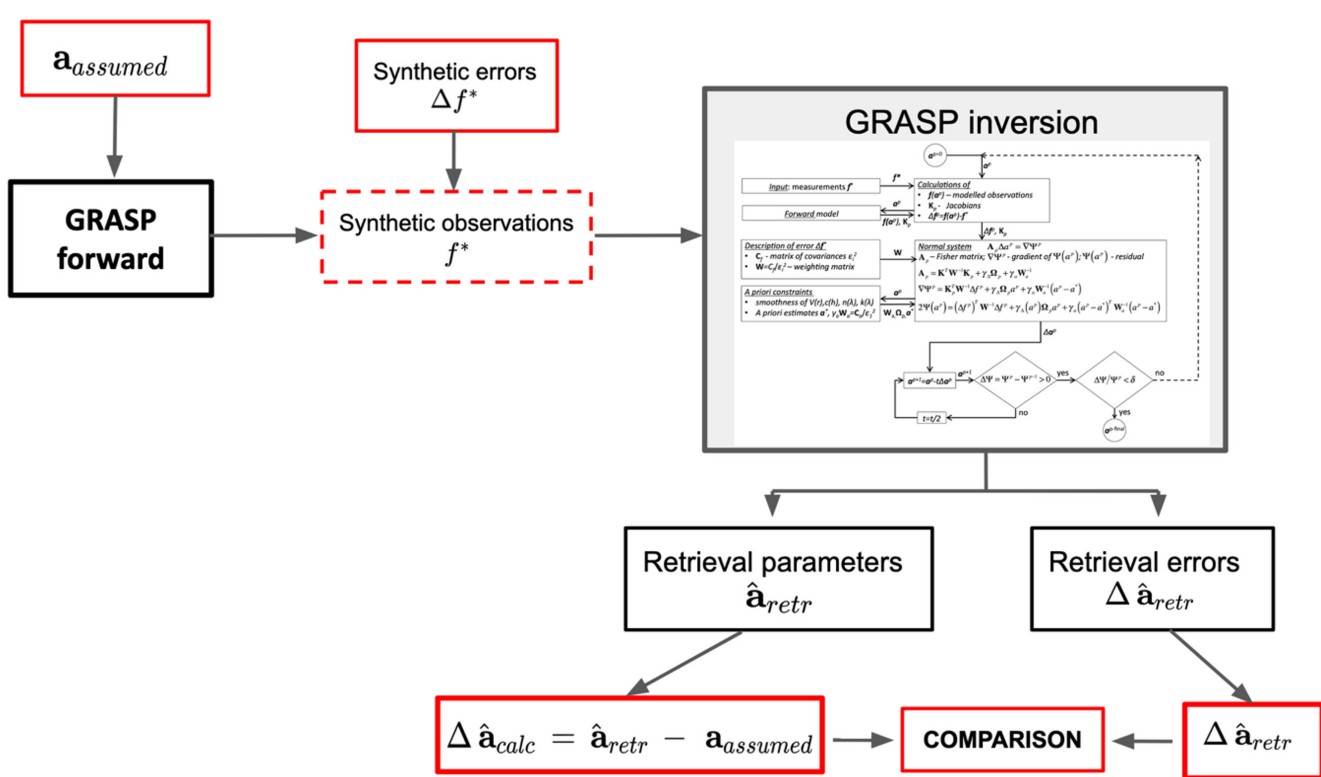

**Figure 1.** General scheme for the validation of the error estimates.

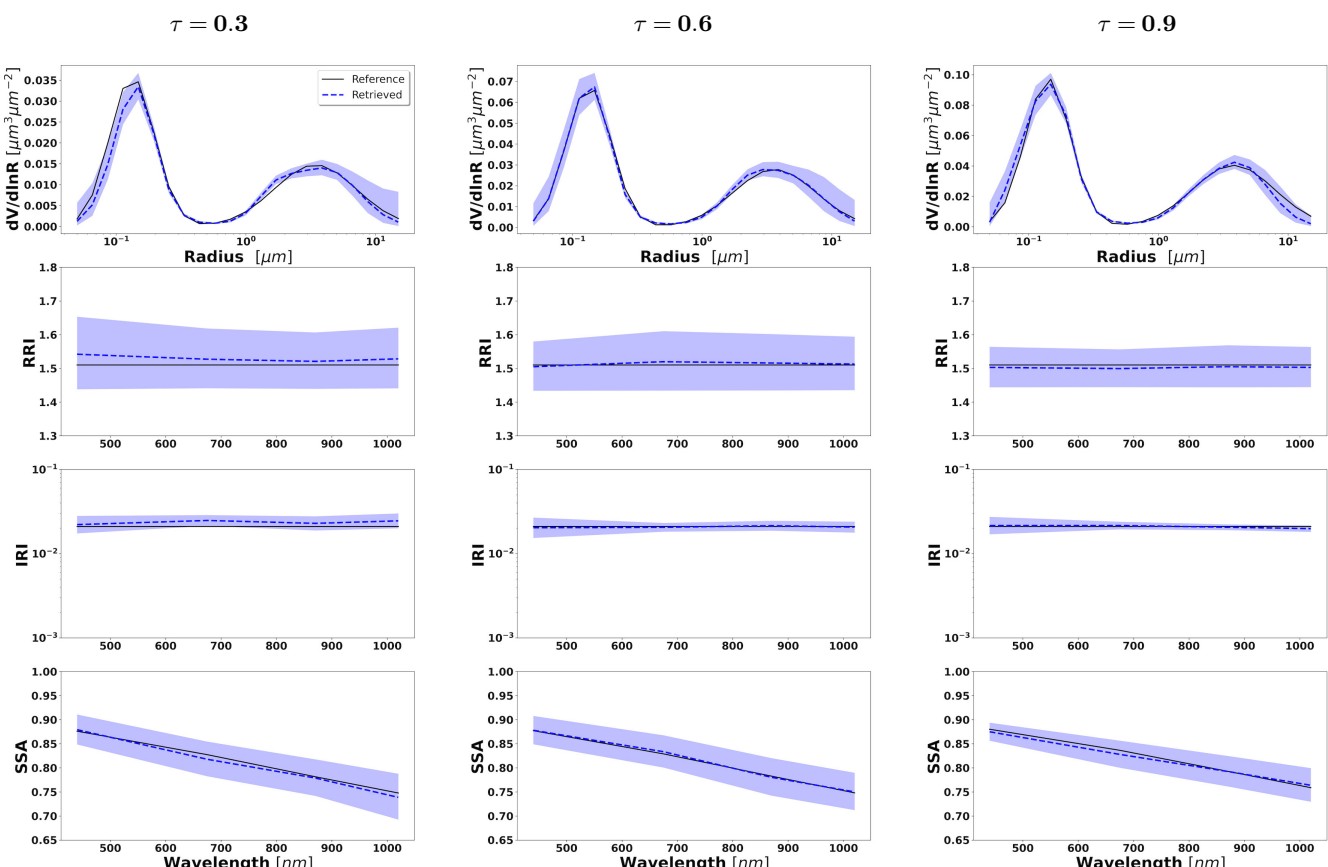

**Figure 2.** Aerosol properties retrieved from simulated for sun-photometer data with random noise added for BB aerosol for $\tau(440) = 0.3$, 0.6 and 0.9 (left to right). The solid lines indicate the simulated properties (SD, RRI, IRI and SSA), the dashed lines are the retrieved parameters. The shaded areas indicate error estimated by GRASP algorithm.

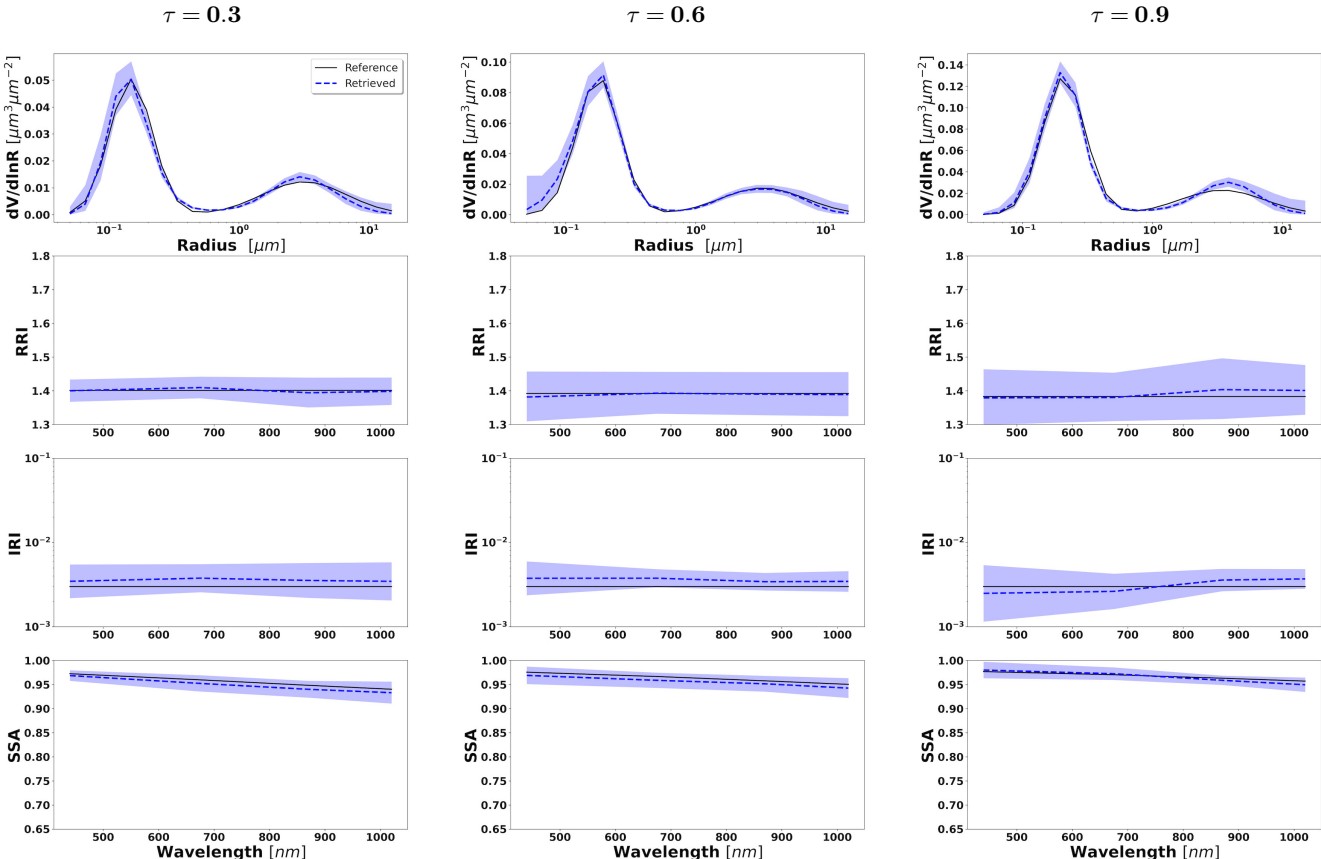

**Figure 3.** Aerosol properties retrieved from simulated for sun-photometer data with random noise added for urban aerosol for $\tau(440) = 0.3$, 0.6 and 0.9 (left to right). The solid lines indicate the simulated properties (SD, RRI, IRI and SSA), the dashed lines are the retrieved parameters. The shaded areas indicate error estimated by GRASP algorithm.



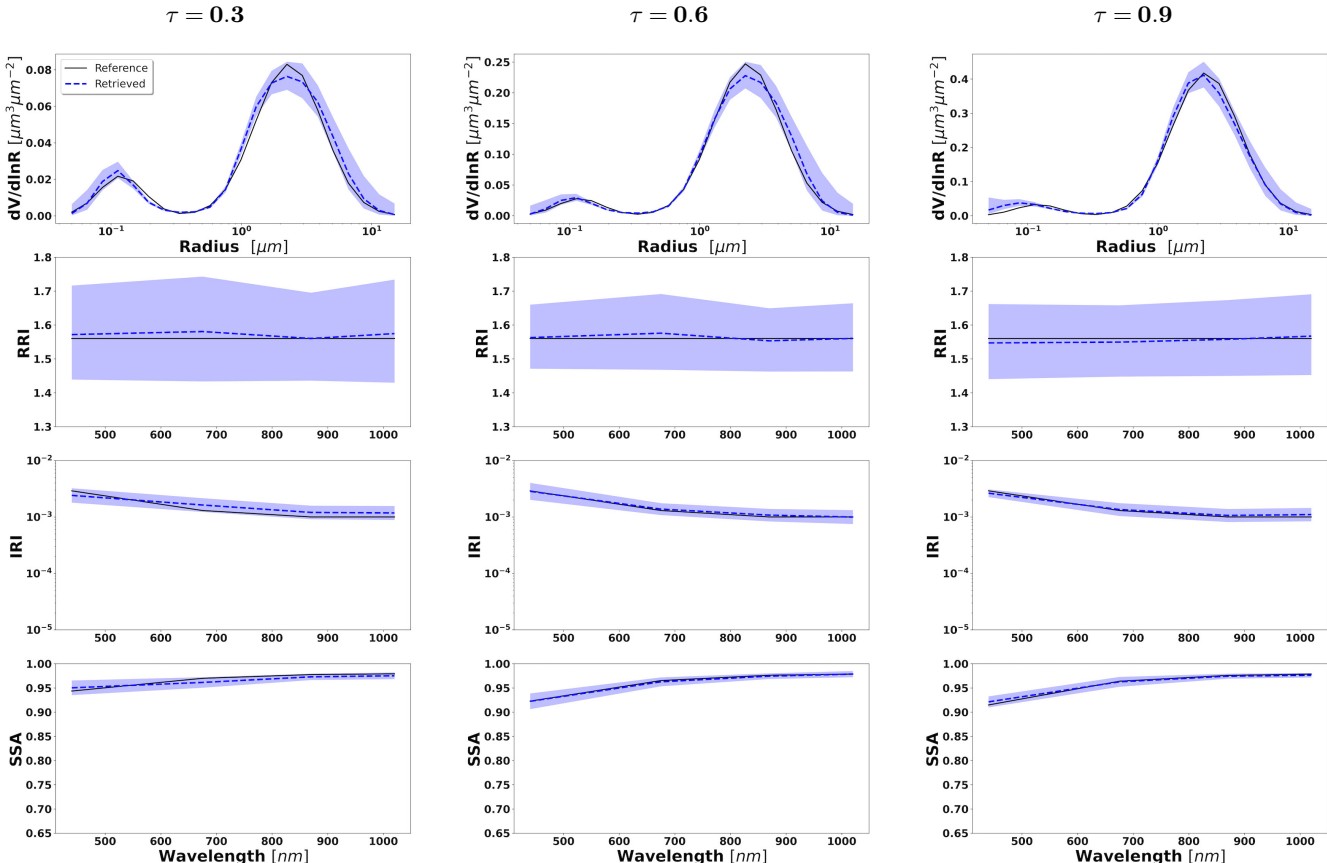

**Figure 4.** Aerosol properties retrieved from simulated for sun-photometer data with random noise added for dust aerosol for $\tau(440) = 0.3$, 0.6 and 0.9 (left to right). The solid lines indicate the simulated properties (SD, RRI, IRI and SSA), the dashed lines are the retrieved parameters. The shaded areas indicate error estimated by GRASP algorithm.

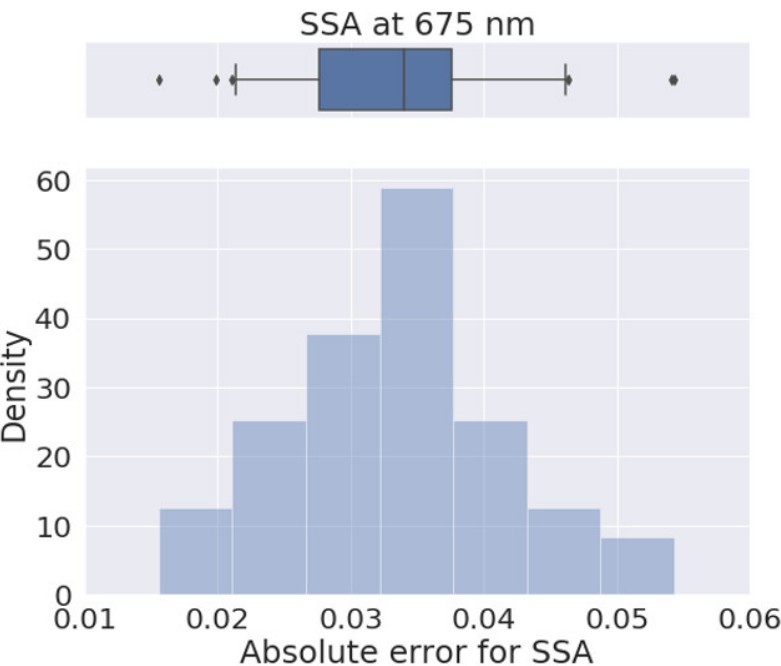

**Figure 5.** The comparison of the variance SSA(675) values estimated by GRASP algorithm with actual errors obtained for extensive tests with randomly added modeled errors. Upper panel: the box represents 50% of the data with the whiskers representing 5th and 95th percentiles of the data and the solid line in the boxplot representing the median value.

**Figure 6.** The comparison of estimated and actual error distributions for spectrally dependent aerosol parameters retrieved from sun/sky photometer simulated measurements (a case with $\tau(440) = 0.6$). The distributions were obtained using 300 realizations of added random errors. The median values of the errors are shown by a line in the boxplot along with the 25-75th percentiles indicate by a box and 5-95th percentiles indicated using whiskers. The mean values are represented by the black dot. The red color shows the error estimates provided by GRASP and the blue shows the calculated actual errors (Eq. 34).



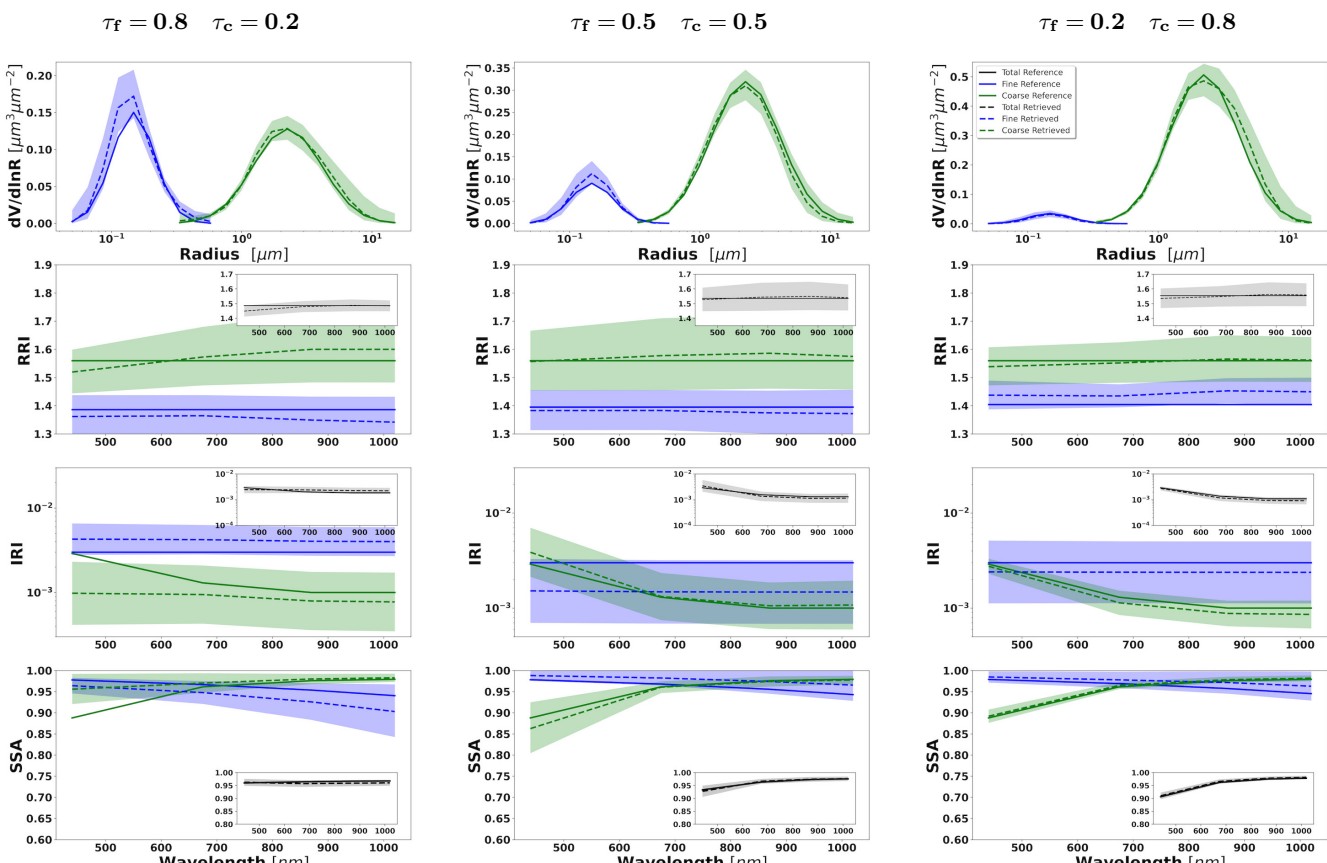

**Figure 7.** Aerosol properties retrieved from simulated sun-photometer data with random noise added for a mixture of Urban-Dust aerosols. The solid lines indicate the simulated properties (SD, RRI, IRI and SSA), the dashed lines are the retrieved parameters. The shaded areas indicate error estimated by GRASP algorithm. The zoomed plots represent the effective refractive index and total SSA.

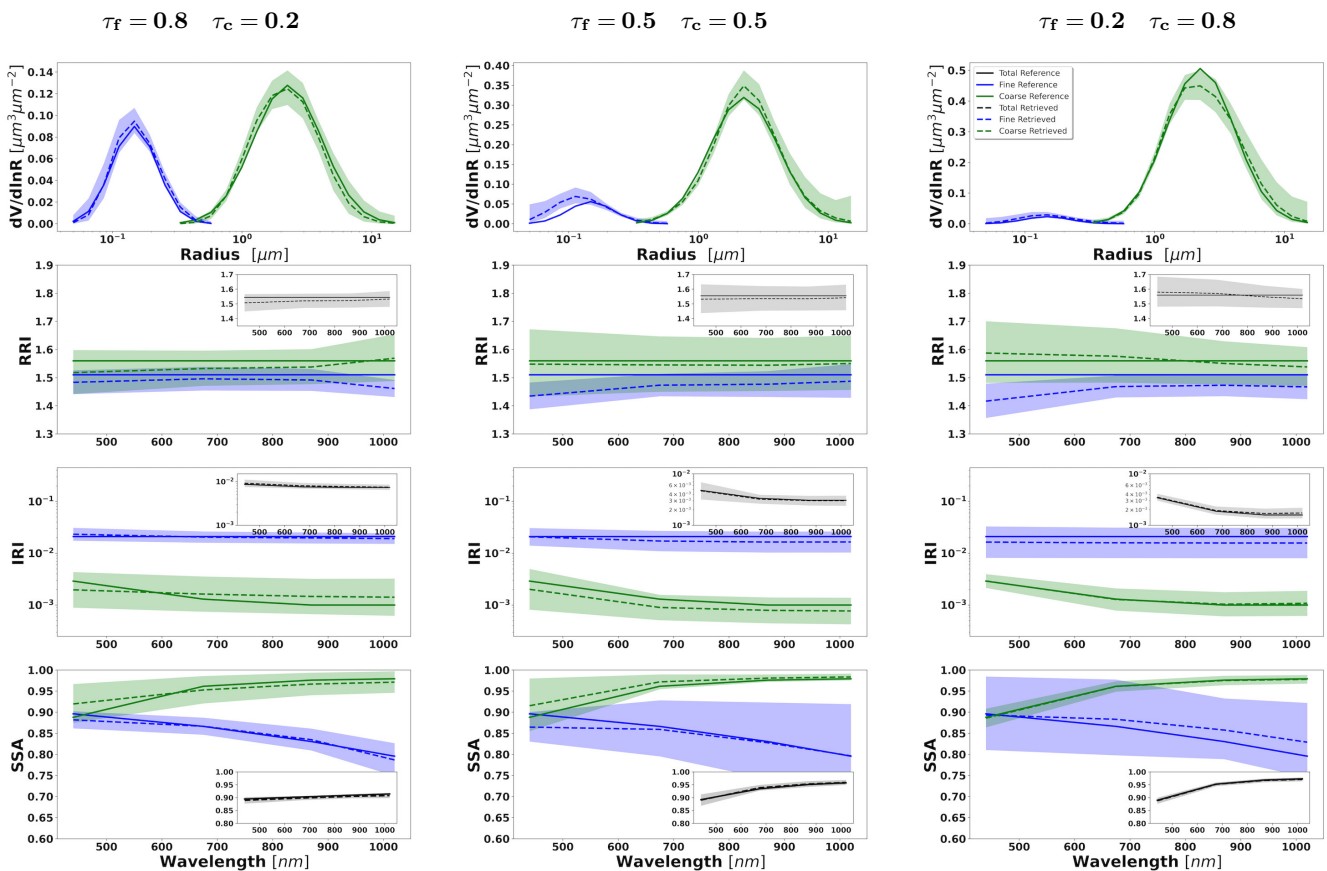

**Figure 8.** Aerosol properties retrieved from simulated sun-photometer data with random noise added for a mixture of BB-Dust aerosols. The solid lines indicate the simulated properties (SD, RRI, IRI and SSA), the dashed lines are the retrieved parameters. The shaded areas indicate error estimated by GRASP algorithm. The zoomed plots represent the effective refractive index and total SSA.

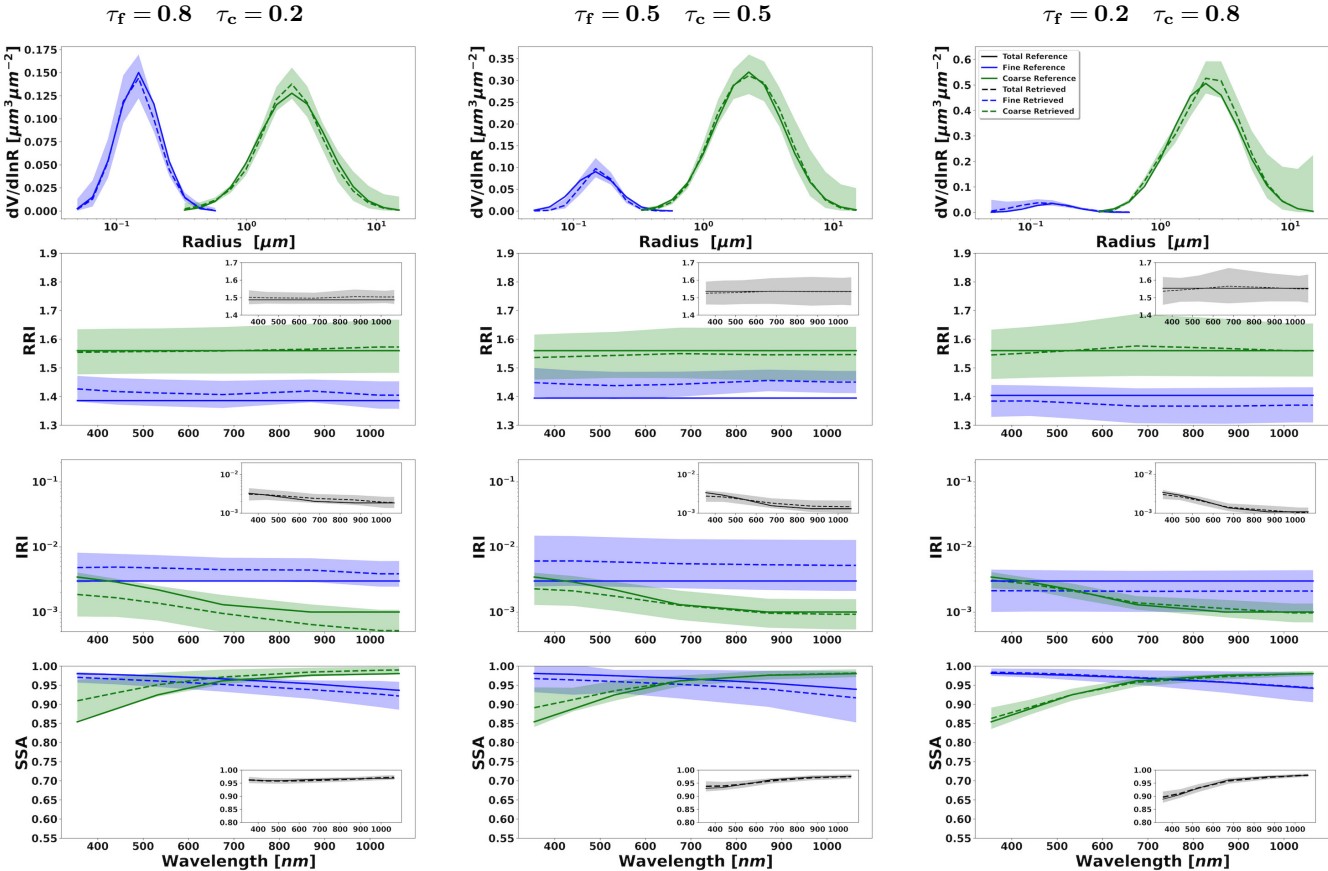

**Figure 9.** Aerosol properties retrieved from simulated sun-photometer and lidar data with random noise added for a mixture of Urban-Dust aerosols. The solid lines indicate the simulated properties (SD, RRI, IRI and SSA), the dashed lines are the retrieved parameters. The shaded areas indicate error estimated by GRASP algorithm. The zoomed plots represent the effective refractive index and total SSA.

**Figure 10.** Aerosol properties retrieved from simulated sun-photometer and lidar data with random noise added for a mixture of BB-Dust aerosols. The solid lines indicate the simulated properties (SD, RRI, IRI and SSA), the dashed lines are the retrieved parameters. The shaded areas indicate error estimated by GRASP algorithm. The zoomed plots represent the effective refractive index and total SSA.



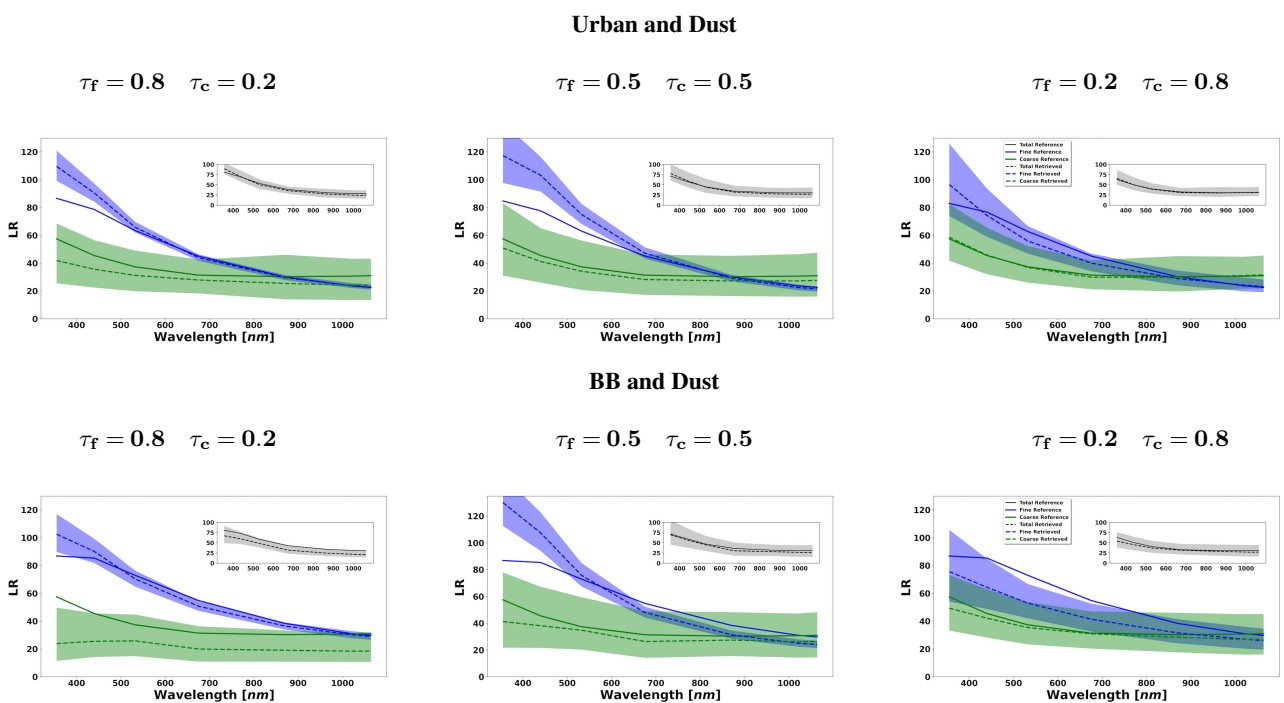

**Figure 11.** The aerosol lidar ratio (LR) retrieved from simulated sun-photometer and lidar data with random noise added for a mixture Urban-Dust aerosols (above) and BB-Dust (below). The solid lines indicate the simulated properties (SD, RRI, IRI and SSA), the dashed lines are the retrieved parameters. The shaded areas indicate error estimated by GRASP algorithm. The zoomed plots represent the results for total LR retrievals.




**Figure 12.** The aerosol AVP retrieved from simulated sun-photometer and lidar data with random noise added for a mixture of Urban-Dust aerosols (above) and BB-Dust (below). The solid lines indicate the simulated properties (AVP), the dashed lines are the retrieved parameters. The shaded areas indicate error estimated by GRASP algorithm.



Atmospheric
Measurement
Techniques



**Figure 13.** The comparison of estimated and actual error distributions for spectrally dependent aerosol parameters retrieved from measurements by sun/sky photometer simulated and lidar for a mixture of Urban-Dust aerosol. The distributions were obtained using 300 realizations of added random errors. The median values of the errors are shown by a line in the boxplot along with the 25-75th percentiles indicate by a box and 5-95th percentiles indicated using whiskers. The mean values are represented by the black dot. The red color shows the error estimates provided by GRASP and the blue shows the calculated actual errors (Eq. 34).







**Figure 14.** The comparison of estimated and actual error distributions for spectrally dependent aerosol parameters retrieved from measurements by sun/sky photometer simulated and lidar for a mixture of BB-Dust aerosol. The distributions were obtained using 300 realizations of added random errors. The median values of the errors are shown by a line in the boxplot along with the 25-75th percentiles indicate by a box and 5-95th percentiles indicated using whiskers. The mean values are represented by the black dot. The red color shows the error estimates provided by GRASP and the blue shows the calculated actual errors (Eq. 34).

**Figure 15.** The comparison of estimated and actual error distributions for aerosol SSA retrieved from measurements by sun/sky photometer simulated and lidar for a mixture of Urban-Dust aerosols. The distributions were obtained using 300 realizations of added random errors. The median values of the errors are shown by a line in the boxplot along with the 25-75th percentiles indicate by a box and 5-95th percentiles indicated using whiskers. The mean values are represented by the black dot. The red color shows the error estimates provided by GRASP and the blue shows the calculated actual errors (Eq. 34).



**Figure 16.** The comparison of estimated and actual error distributions for aerosol SSA retrieved from measurements by sun/sky photometer simulated and lidar for a mixture of BB-Dust aerosols. The distributions were obtained using 300 realizations of added random errors. The median values of the errors are shown by a line in the boxplot along with the 25-75th percentiles indicate by a box and 5-95th percentiles indicated using whiskers. The mean values are represented by the black dot. The red color shows the error estimates provided by GRASP and the blue shows the calculated actual errors (Eq. 34).

**Figure 17.** The comparison of estimated and actual error distributions for aerosol LR retrieved from measurements by sun/sky photometer simulated and lidar for a mixture of Urban-Dust aerosols. The distributions were obtained using 300 realizations of added random errors. The median values of the errors are shown by a line in the boxplot along with the 25-75th percentiles indicate by a box and 5-95th percentiles indicated using whiskers. The mean values are represented by the black dot. The red color shows the error estimates provided by GRASP and the blue shows the calculated actual errors (Eq. 34).



**Figure 18.** The comparison of estimated and actual error distributions for aerosol LR retrieved from measurements by sun/sky photometer simulated and lidar for a mixture of BB-Dust aerosols. The distributions were obtained using 300 realizations of added random errors. The median values of the errors are shown by a line in the boxplot along with the 25-75th percentiles indicate by a box and 5-95th percentiles indicated using whiskers. The mean values are represented by the black dot. The red color shows the error estimates provided by GRASP and the blue shows the calculated actual errors (Eq. 34).





**Urban and Dust**

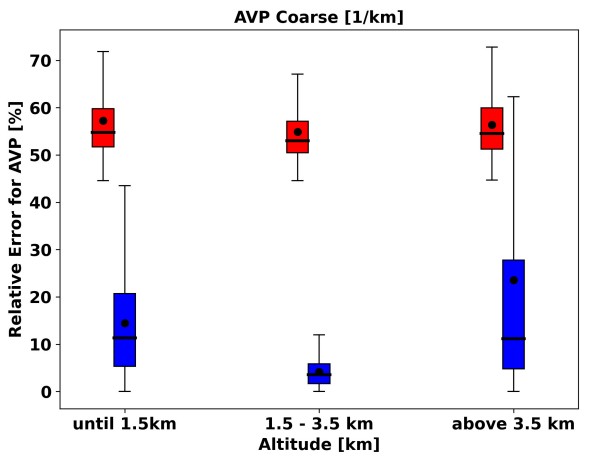

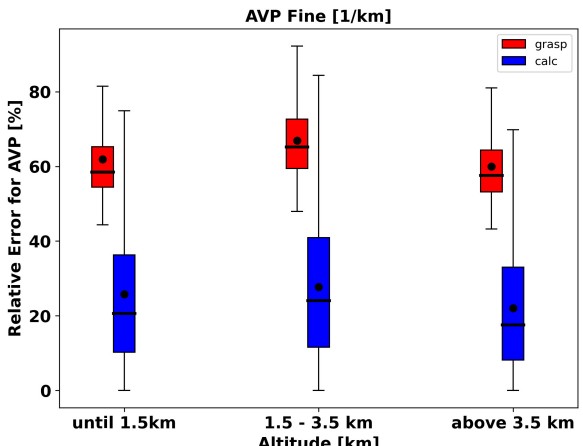

**BB and Dust**

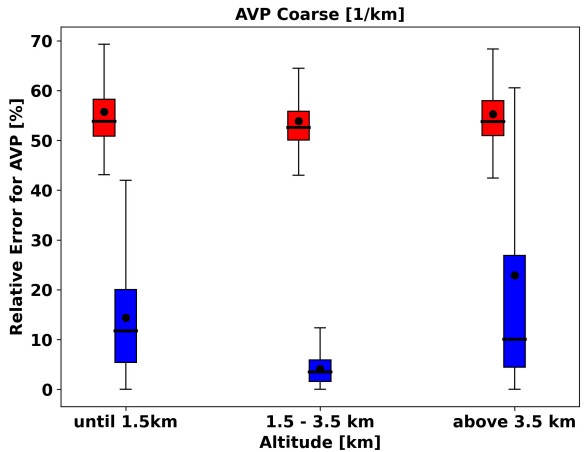

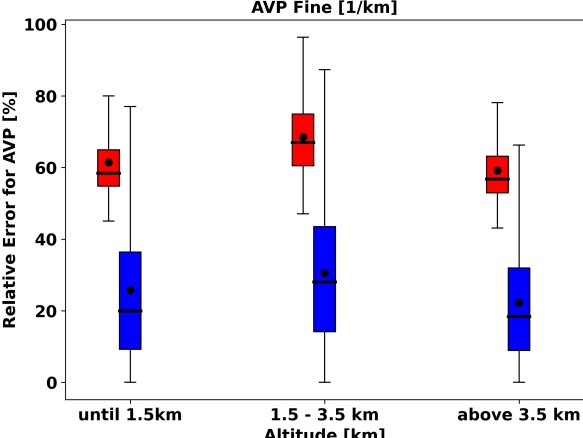

**Figure 19.** The comparison of estimated and actual error distributions for AVP retrieved from measurements by sun/sky photometer simulated and lidar for a mixture of Urban-Dust aerosols (top) and BB-Dust (bottom). The distributions were obtained using 300 realizations of added random errors. The mean values are represented by the black dot and the median values of the errors are shown by a line in the boxplot along with the $25 - 75th$ percentiles indicate by a box and $5 - 95th$ percentiles indicated using whiskers. The mean values are represented by the black dot. The red color shows the error estimates provided by GRASP and the blue shows the calculated actual errors (Eq. 34).

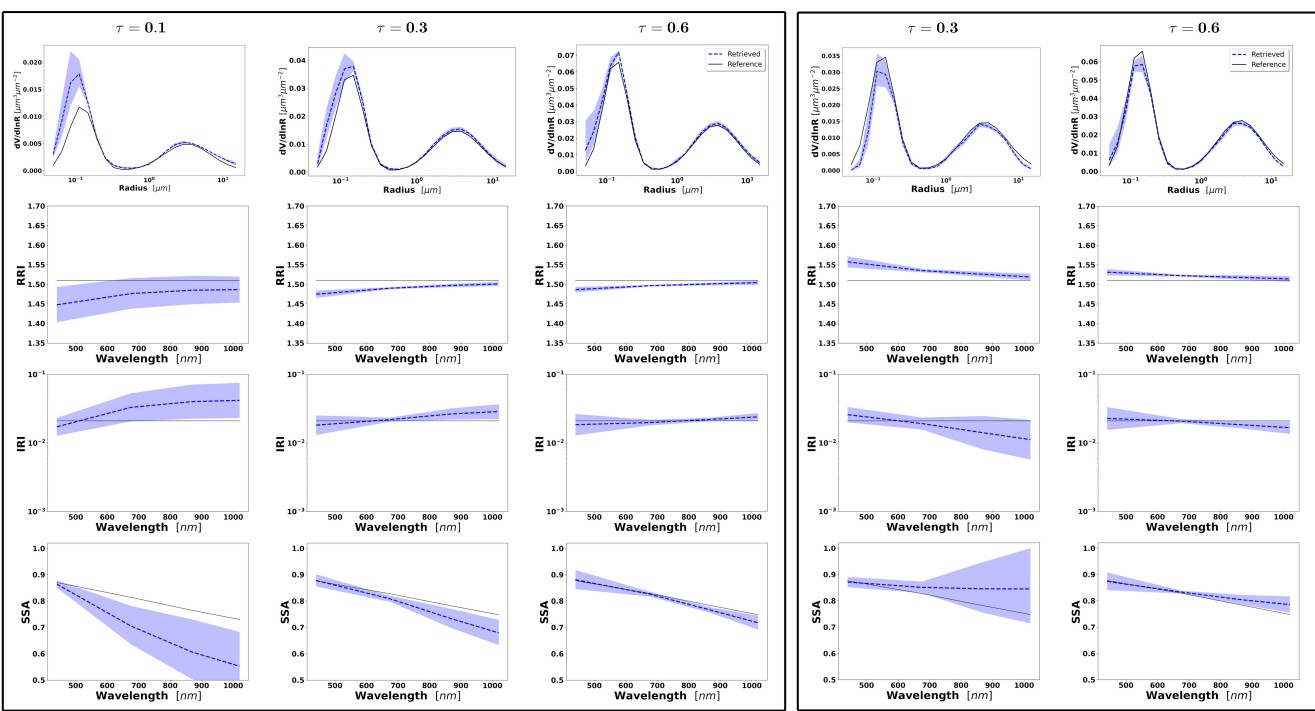

**Figure 20.** Aerosol properties retrieved from simulated sun/sky photometer data with assumed bias in AOD simulated data for BB aerosol for $\tau(440) = 0.1$, $0.3$ and $0.6$ (left to right). Retrievals after adding positive bias $+0.01$ are represented in the block on the left and negative bias $-0.01$ in the block on the right. The solid lines are the simulated properties (SD, RRI, IRI and SSA), the dashed lines are the retrieved parameters. The shaded area indicates systematic errors estimated by GRASP algorithm.





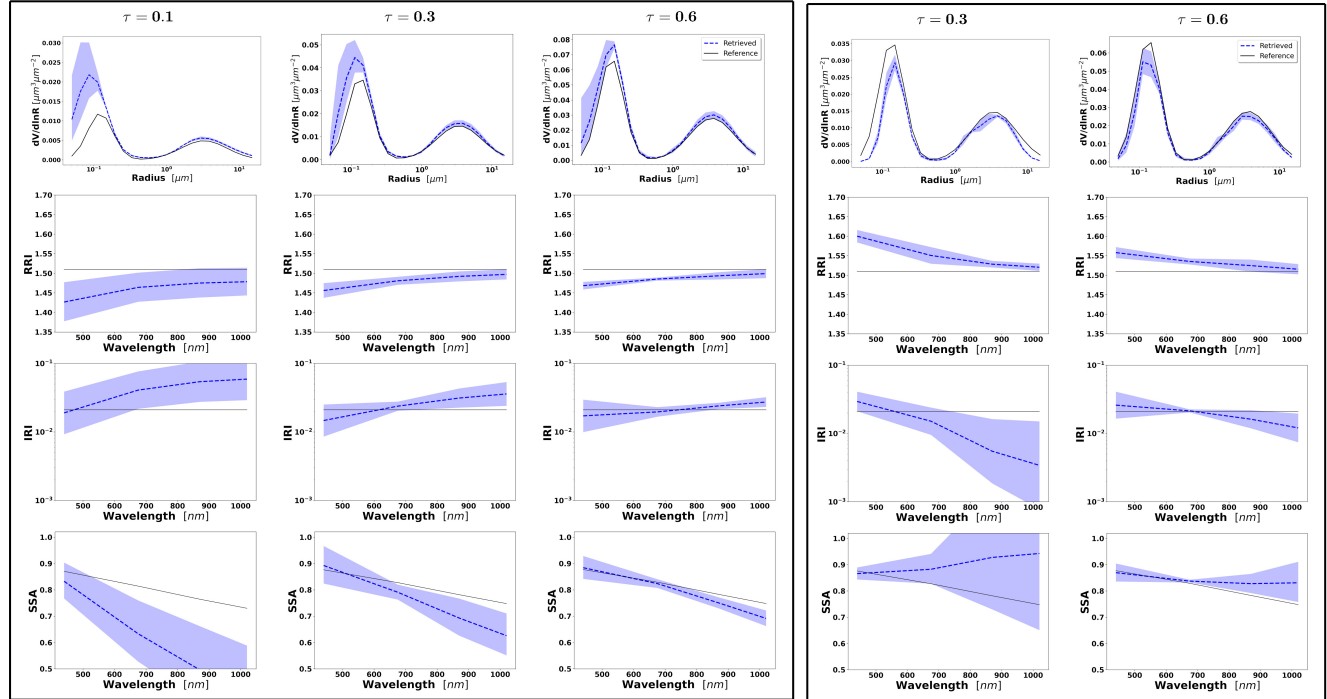

**Figure 21.** Aerosol properties retrieved from simulated sun/sky photometer data with assumed bias in AOD simulated data for BB aerosol for $\tau(440) = 0.1$, 0.3 and 0.6 (left to right). Retrievals after adding positive bias $+0.02$ are represented in the block on the left and negative bias $-0.02$ in the block on the right. The solid lines are the simulated properties (SD, RRI, IRI and SSA), the dashed lines are the retrieved parameters. The shaded area indicates systematic errors estimated by GRASP algorithm.

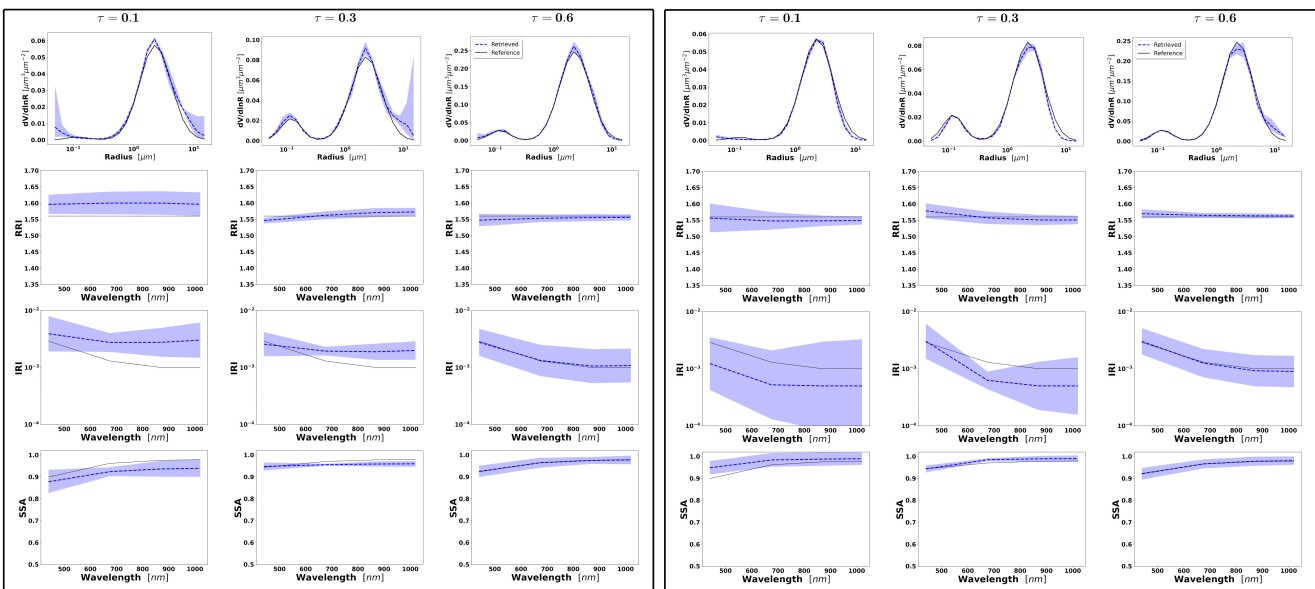

**Figure 22.** Aerosol properties retrieved from simulated sun/sky photometer data with assumed bias in AOD simulated data for dust aerosol for $\tau(440) = 0.1, 0.3$ and $0.6$ (left to right). Retrievals after adding positive bias $+0.01$ are represented in the block on the left and negative bias $-0.01$ in the block on the right. The solid lines are the simulated properties (SD, RRI, IRI and SSA), the dashed lines are the retrieved parameters. The shaded area indicates systematic errors estimated by GRASP algorithm.



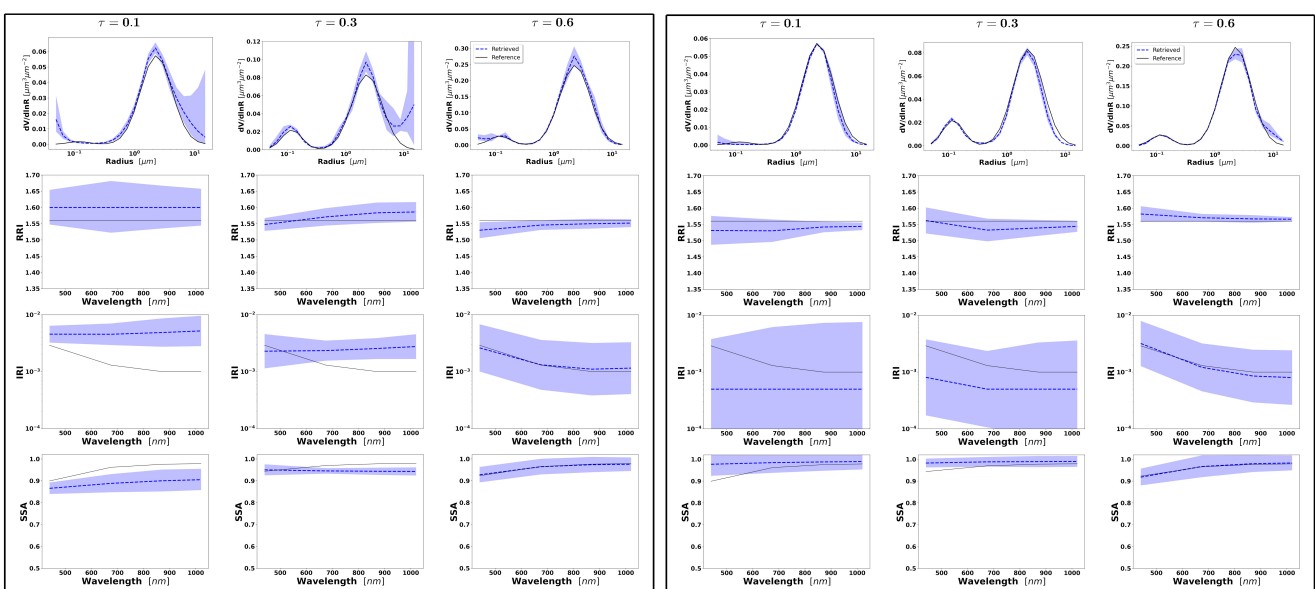

**Figure 23.** Aerosol properties retrieved from simulated sun/sky photometer data with assumed bias in AOD simulated data for dust aerosol for $\tau(440) = 0.1$, $0.3$ and $0.6$ (left to right). Retrievals after adding positive bias $+0.02$ are represented in the block on the left and negative bias $-0.02$ in the block on the right. The solid lines are the simulated properties (SD, RRI, IRI and SSA), the dashed lines are the retrieved parameters. The shaded area indicates systematic errors estimated by GRASP algorithm.



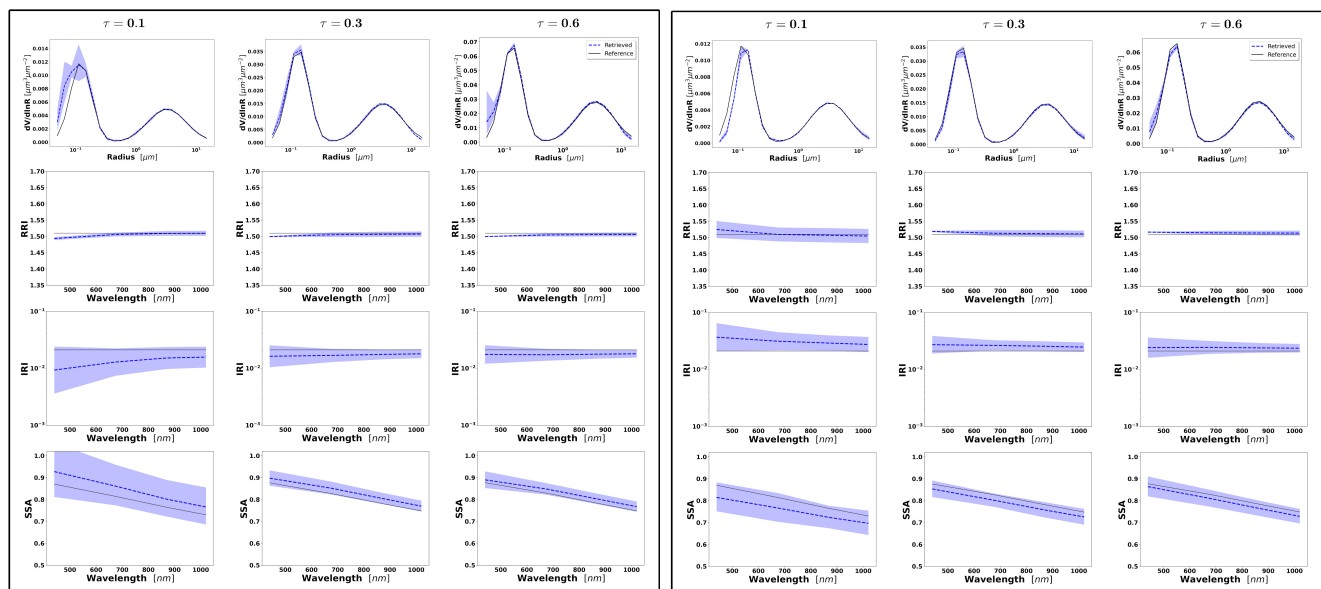

**Figure 24.** Aerosol properties retrieved from simulated sun/sky photometer data with assumed bias in RAD simulated data for BB aerosol for $\tau(440) = 0.1$, 0.3 and 0.6 (left to right). Retrievals after adding positive bias $+3\%$ are represented in the block on the left and negative bias $-3\%$ in the block on the right. The solid lines are the simulated properties (SD, RRI, IRI and SSA), the dashed lines are the retrieved parameters. The shaded area indicates systematic errors estimated by GRASP algorithm.



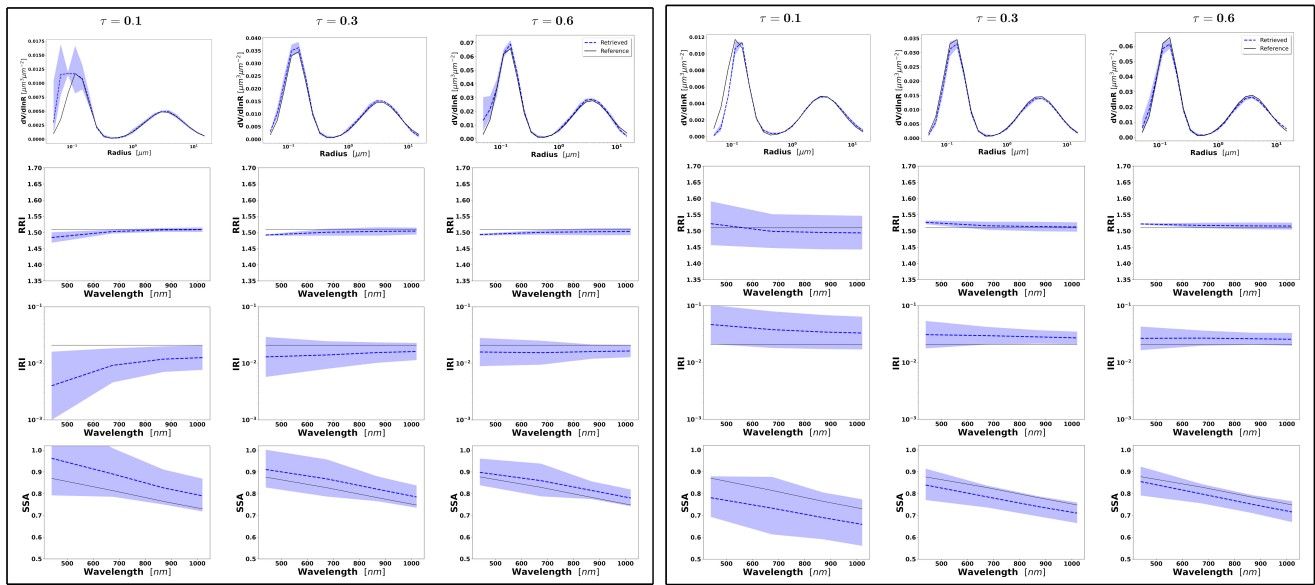

**Figure 25.** Aerosol properties retrieved from simulated sun-photometer data with assumed bias in RAD simulated data for BB aerosol for $\tau(440) = 0.1$, 0.3 and 0.6 (left to right). Retrievals after adding positive bias $+5\%$ are represented in the block on the left and negative bias $-5\%$ in the block on the right. The solid lines are the simulated properties (SD, RRI, IRI and SSA), the dashed lines are the retrieved parameters. The shaded area indicates systematic errors estimated by GRASP algorithm.





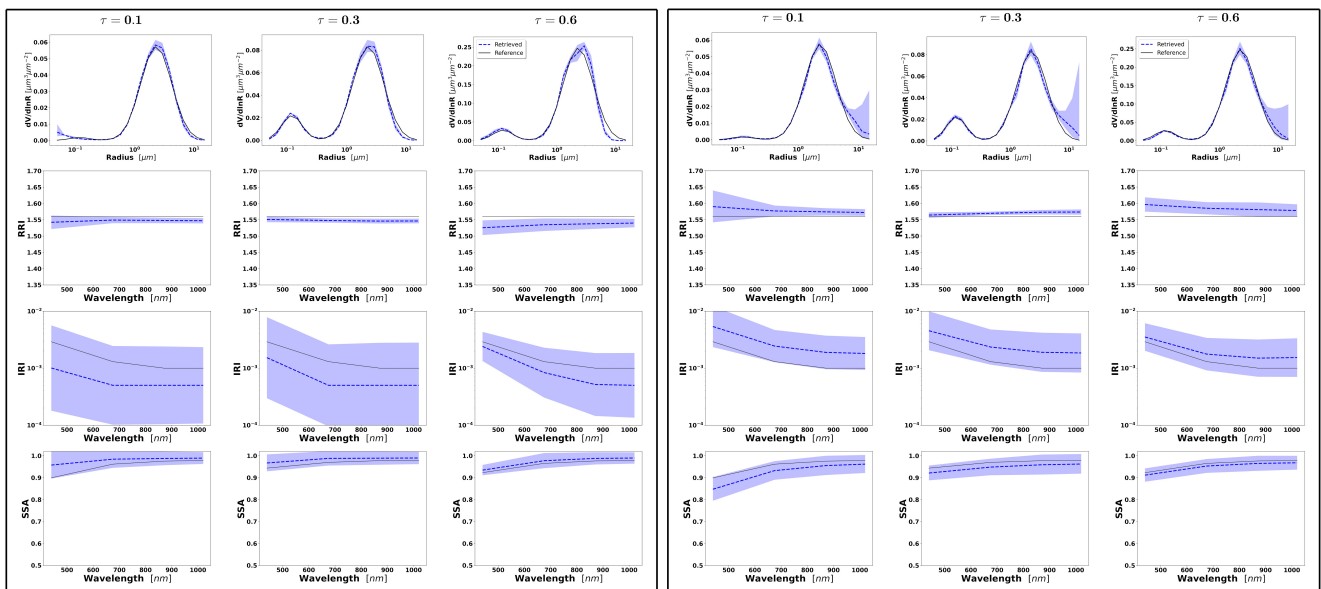

**Figure 26.** Aerosol properties retrieved from simulated sun/sky photometer data with assumed bias in RAD simulated data for Dust aerosol for $\tau(440) = 0.1$, 0.3 and 0.6 (left to right). Retrievals after adding positive bias $+3\%$ are represented in the block on the left and negative bias $-3\%$ in the block on the right. The solid lines are the simulated properties (SD, RRI, IRI and SSA), the dashed lines are the retrieved parameters. The shaded area indicates systematic errors estimated by GRASP algorithm.

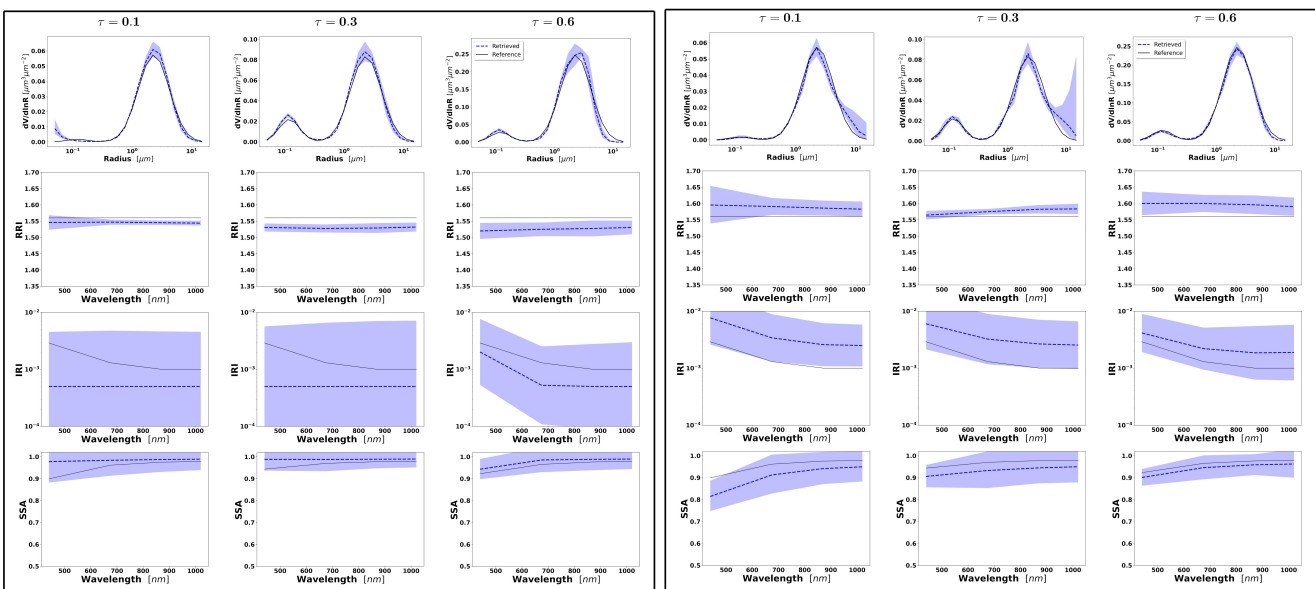

**Figure 27.** Aerosol properties retrieved from simulated sun/sky photometer data with assumed bias in RAD simulated data for Dust aerosol for $\tau(440) = 0.1$, 0.3 and 0.6 (left to right). Retrievals after adding positive bias $+5\%$ are represented in the block on the left and negative bias $-5\%$ in the block on the right. The solid lines are the simulated properties (SD, RRI, IRI and SSA), the dashed lines are the retrieved parameters. The shaded area indicates systematic errors estimated by GRASP algorithm.



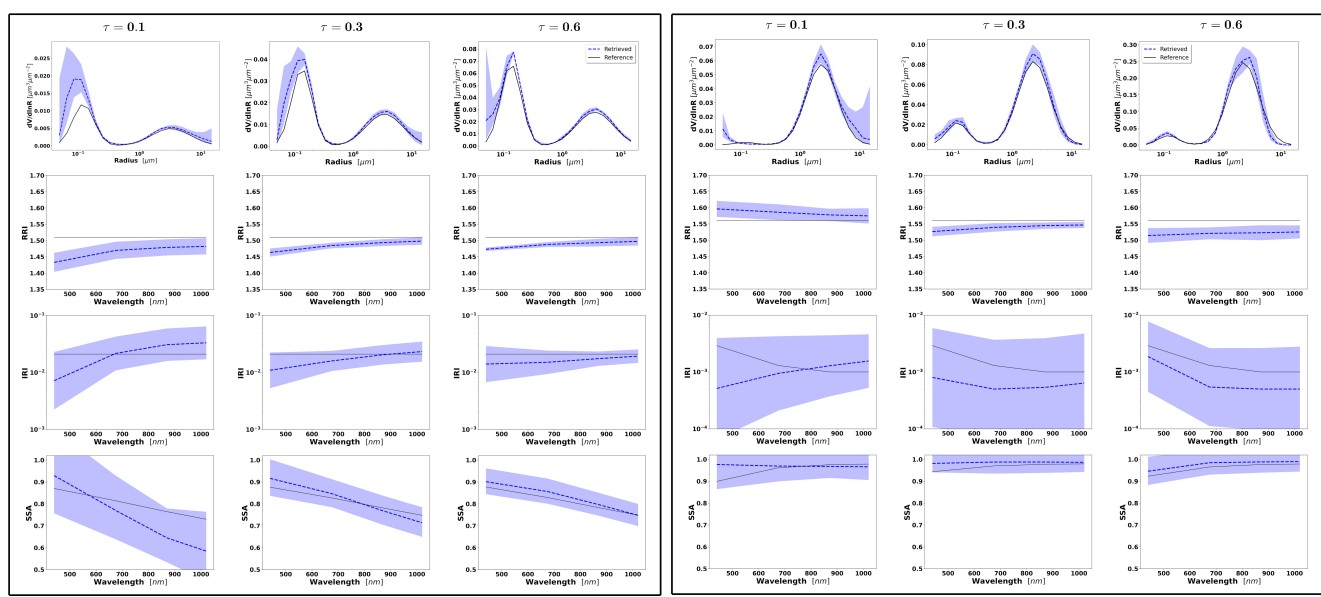

**Figure 28.** Aerosol properties retrieved from simulated sun/sky photometer data with assumed bias in AOD and radiances simulated data for BB (left) and Dust (right) aerosol for $\tau(440) = 0.1$, 0.3 and 0.6 (left to right). Retrievals after adding positive bias $+0.01$ in AOD and $+5\%$ in radiances in both cases. The solid lines are the simulated properties (SD, RRI, IRI and SSA), the dashed lines are the retrieved parameters. The shaded area indicates systematic error estimated by GRASP algorithm.



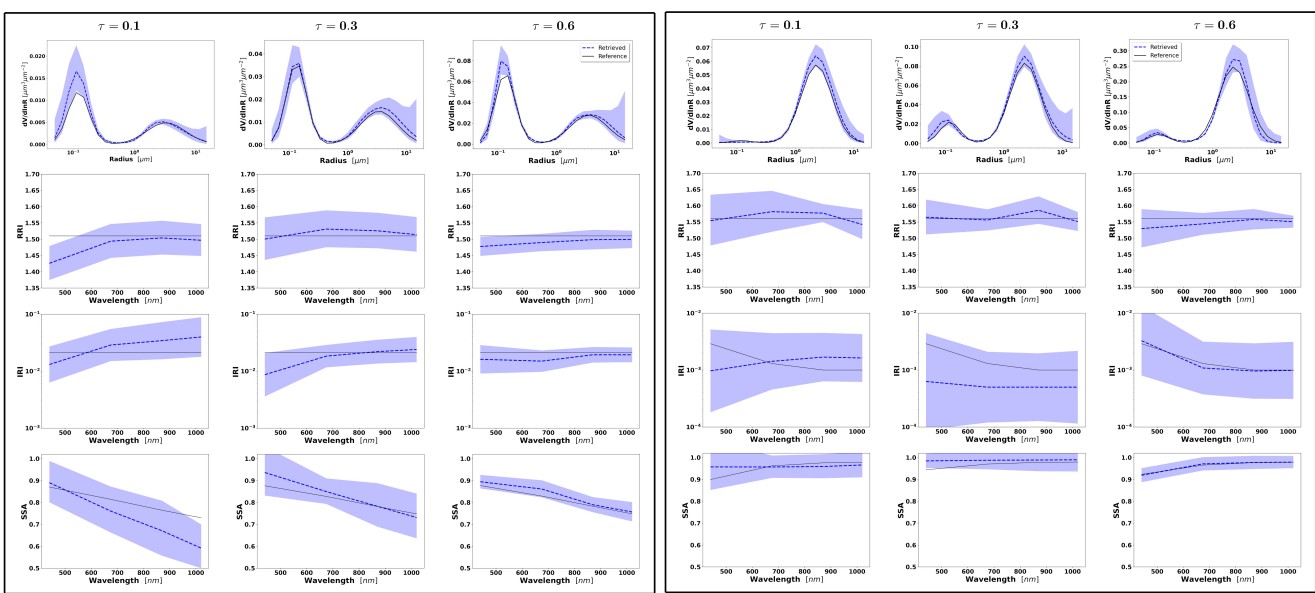

**Figure 29.** Aerosol properties retrieved from simulated sun/sky photometer data with assumed random noise and bias in AOD and radiances simulated data for BB (left) and dust (right) aerosol for $\tau(440) = 0.1$, 0.3 and 0.6 (left to right). Retrievals after adding positive bias $+0.01$ in AOD and $+5\%$ in radiances in both cases. The solid lines are the simulated properties (SD, RRI, IRI and SSA), the dashed lines are the retrieved parameters. The shaded area indicates total errors provided by GRASP algorithm.



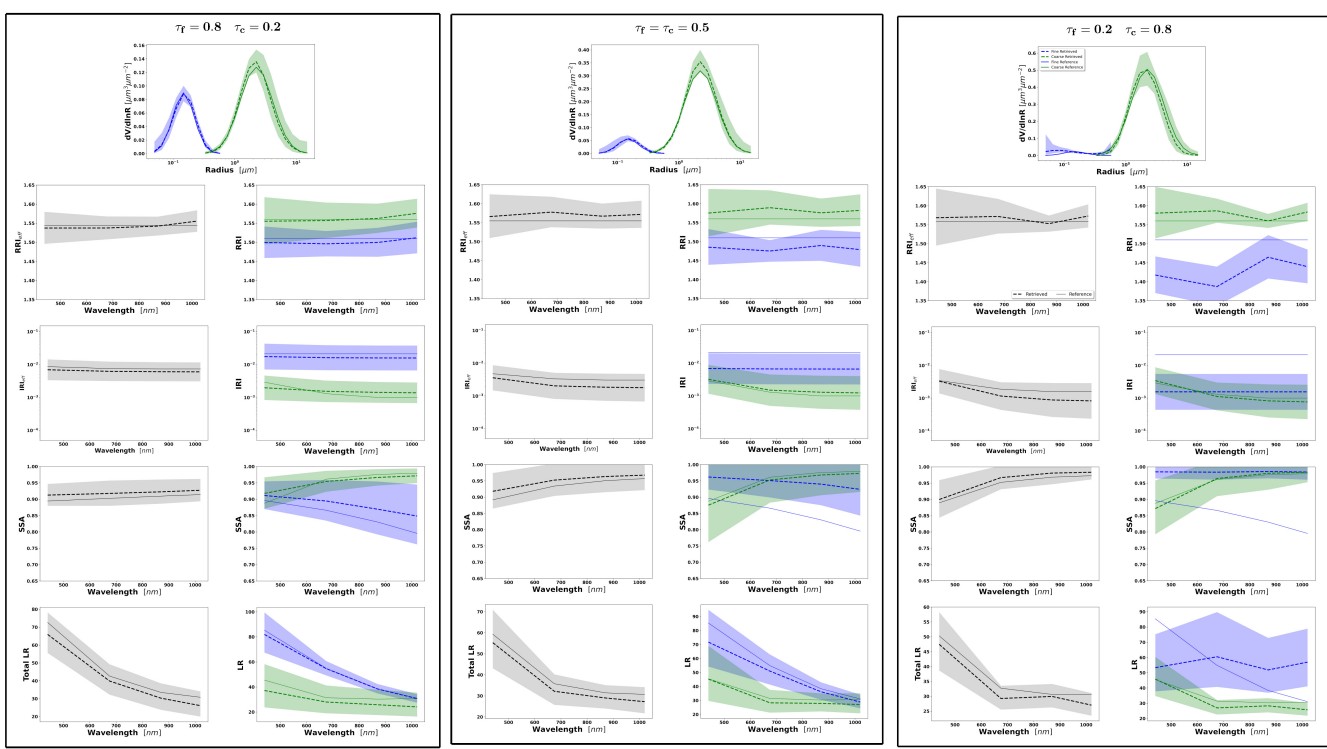

**Figure 30.** Aerosol properties retrieved from simulated sun/sky photometer data with assumed random noise and bias in AOD and radiances simulated data for BB-Dust for $\tau(440) = 1.0$. Retrievals after adding positive bias $+0.01$ in AOD and $+5\%$ in RAD in both cases. The solid lines are the simulated properties (SD, RRI, IRI, SSA and LR), the dashed lines are the retrieved parameters. The shaded area indicates total errors provided by GRASP algorithm.



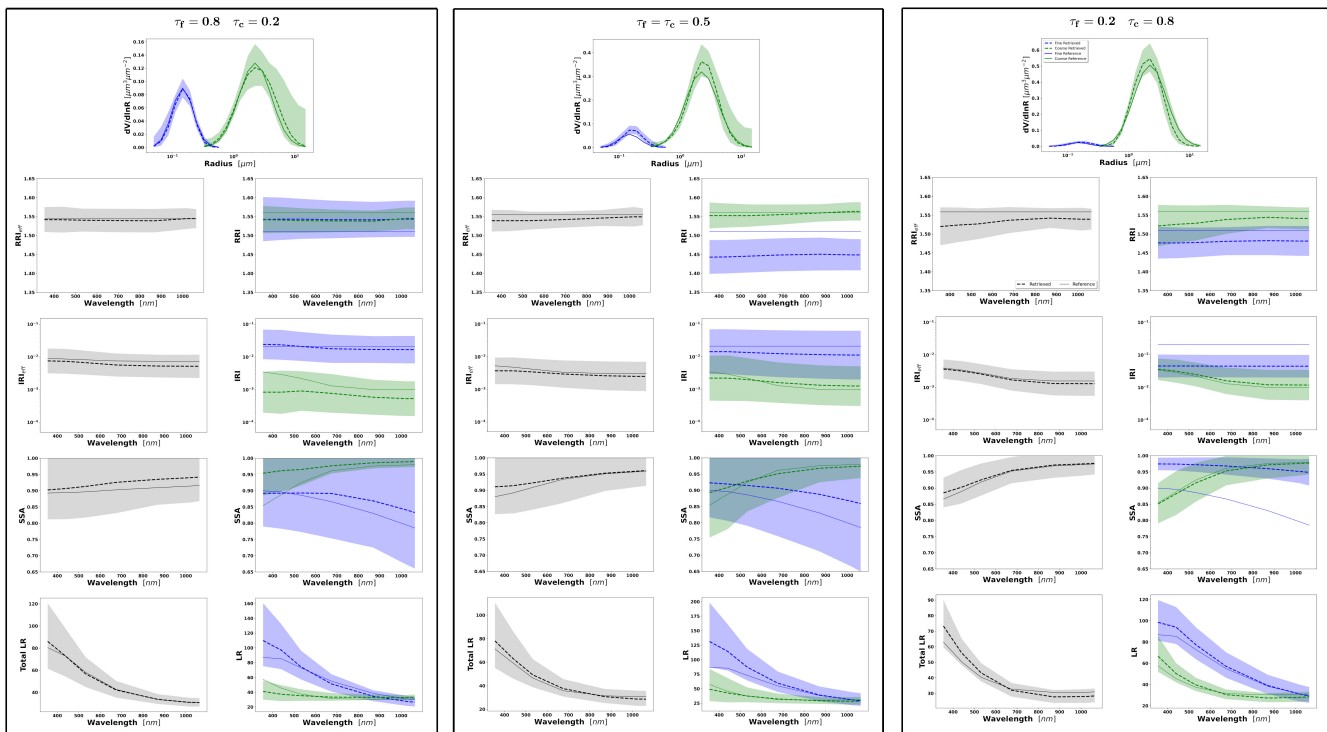

**Figure 31.** Aerosol properties retrieved from simulated sun/sky photometer and lidar data with assumed random noise and bias in AOD, radiances and lidar simulated data for BB-Dust for $\tau(440) = 1.0$. Retrievals after adding positive bias $+0.01$ in AOD, $+5\%$ in RAD and $+0.2$, $+0.15$ and $+0.1$ for each lidar wavelength: $355\,nm$, $532\,nm$ and $1064\,nm$ respectively. The solid lines are the simulated properties (SD, RRI, IRI, SSA and LR), the dashed lines are the retrieved parameters. The shaded area indicates total errors estimated by GRASP algorithm.

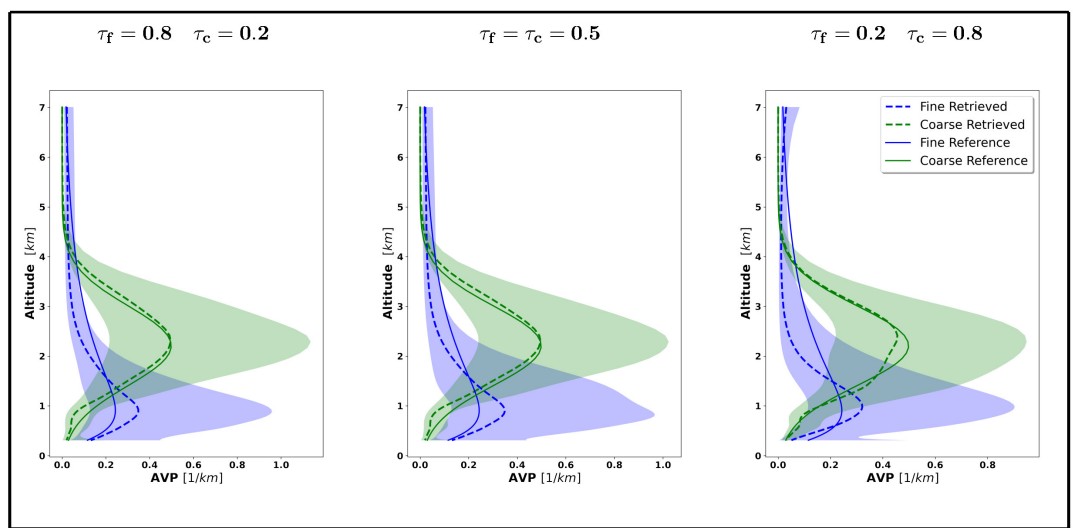

**Figure 32.** Aerosol vertical profiles (AVP) from simulated sun/sky photometer and lidar data with assumed random noise and bias in AOD, radiances and lidar simulated data for BB-Dust for $\tau(440) = 1.0$. Retrievals after adding positive bias $+0.01$ in AOD, $+5\%$ in RAD and $+0.2$, $+0.15$ and $+0.1$ for each lidar wavelength: $355\,nm$, $532\,nm$ and $1064\,nm$ respectively. The solid lines are the simulated properties (SD, RRI, IRI, SSA and LR), the dashed lines are the retrieved parameters. The shaded area indicates total errors estimated by GRASP algorithm.



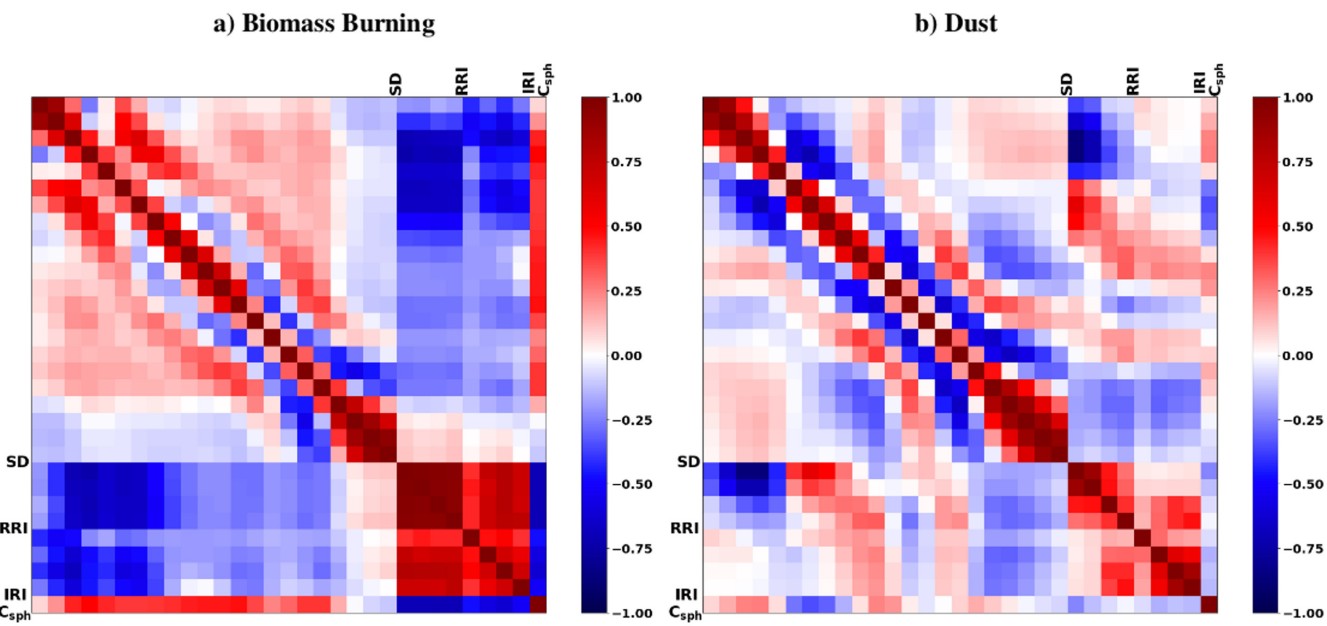

**Figure 33.** Correlation matrices of the estimated errors for aerosol retrieval from Sun/sky-radiometer observations a) for biomass burning aerosols, and b) for desert dust using GRASP algorithm. The values close to 1 or −1 mean stronger correlations between the properties, positive or negative, respectively.



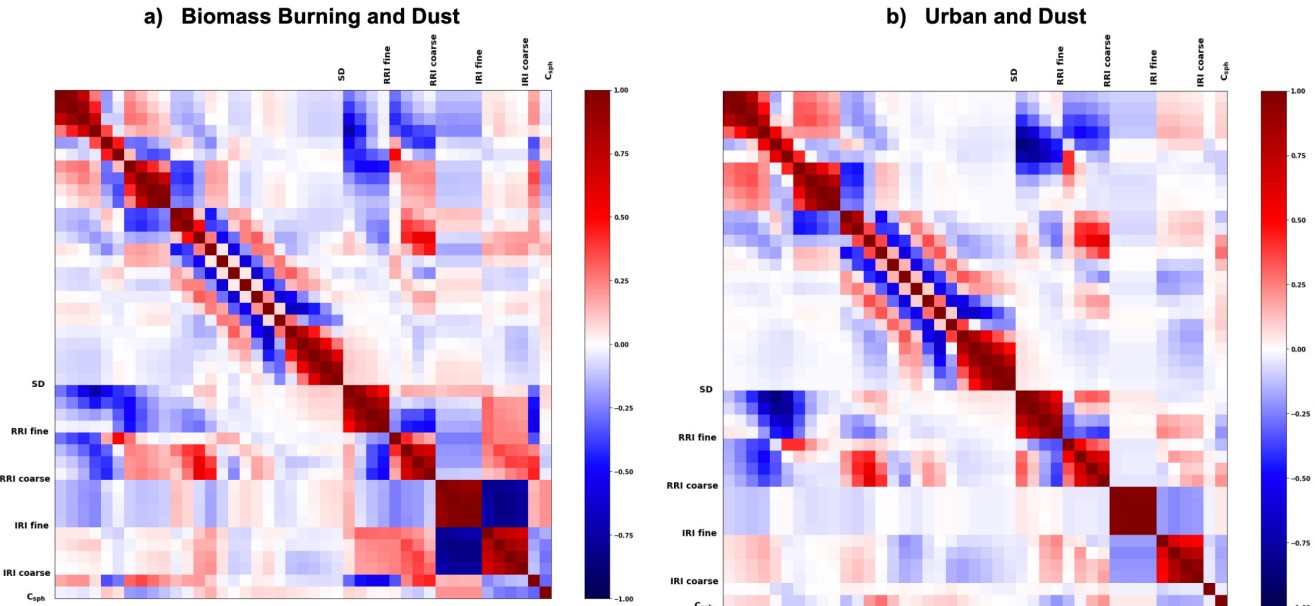

**Figure 34.** Correlation matrices of the estimated errors for mixed aerosol retrieval from Sun/sky-radiometer observations a) for the mixture of biomass burning and dust aerosols, and b) for the mixture of urban and dust aerosols using GRASP algorithm. The values close to 1 or −1 mean stronger correlations between the properties, positive or negative, respectively.



**Figure 35.** Correlation matrices of the estimated errors for aerosol retrieval from joint Sun/sky-radiometer and lidar observations for a mixture of urban and desert dust using GRASP algorithm. The values close to 1 or −1 mean stronger correlations between the properties, positive or negative, respectively.



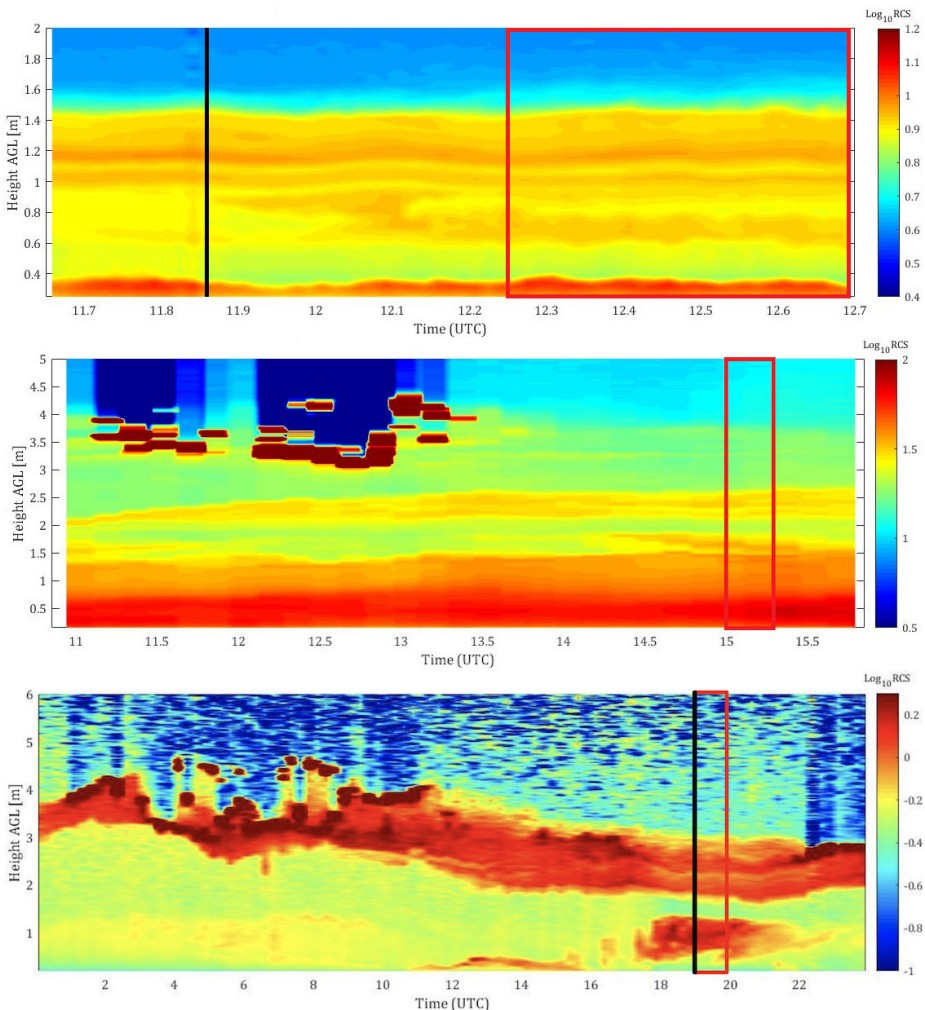

**Figure 36.** RSC at 1064 nm in arbitrary units from Villa Martelli, Argentina on 19 August 2014 (a) and 22 August 2014 (b), and from Aeroparque station (c), Argentina on 25 September 2017. The two red lines indicate the analyzed lidar data interval. The black solid line indicates the sun-photometer measurements.



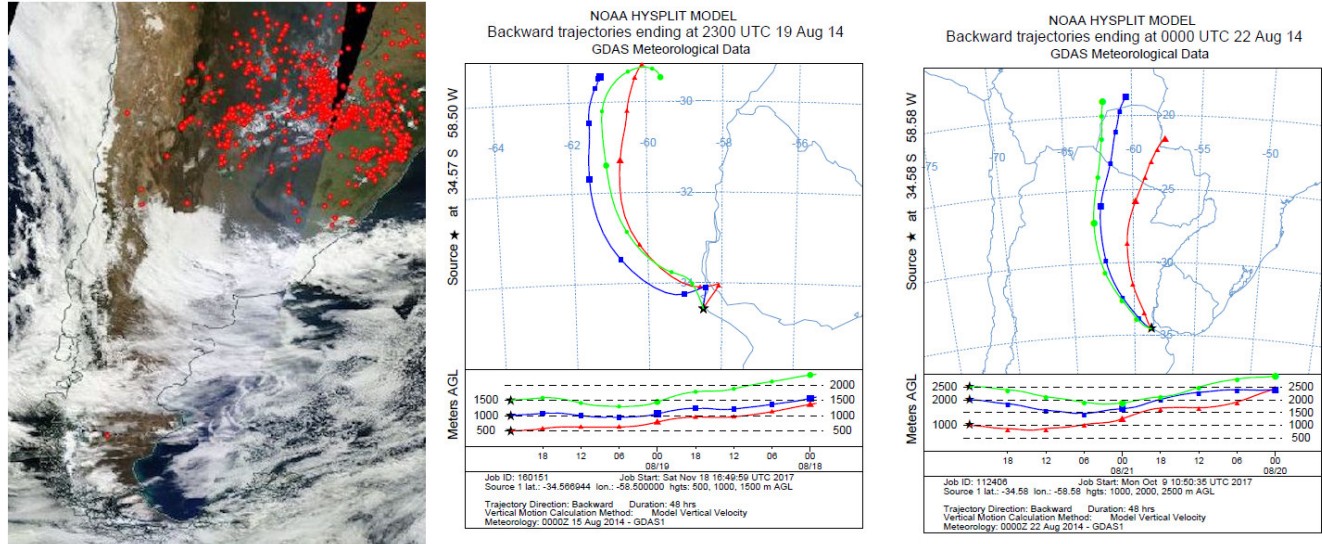

**Figure 37.** a) Satellite image with hot spots corresponding to August 22, 2014 (NASA Worldview), b) air mass back trajectories for the Villa Martelli measurement site on August 19, 2014 and c) air mass back trajectories for the Villa Martelli measurement site on August 22, 2014.

**Figure 38.** Comparison of columnar properties retrieved by GRASP from a combination of sun/sky photometer and lidar data and retrieved conventionally by AERONET. SD, RRI, IRI and SSA retrieved by GRASP are shown in solid lines: blue (fine mode) and green (coarse mode). The shaded area in colors blue and green represent the total error provided by GRASP and black shaded areas are the uncertainties provided by AERONET. Zoomed panels show the RI effective and total SSA provided by GRASP (black solid line) and AERONET (black dashed line). Their associated errors are represented in grey sahded area for GRASP and with error bars for AERONET.



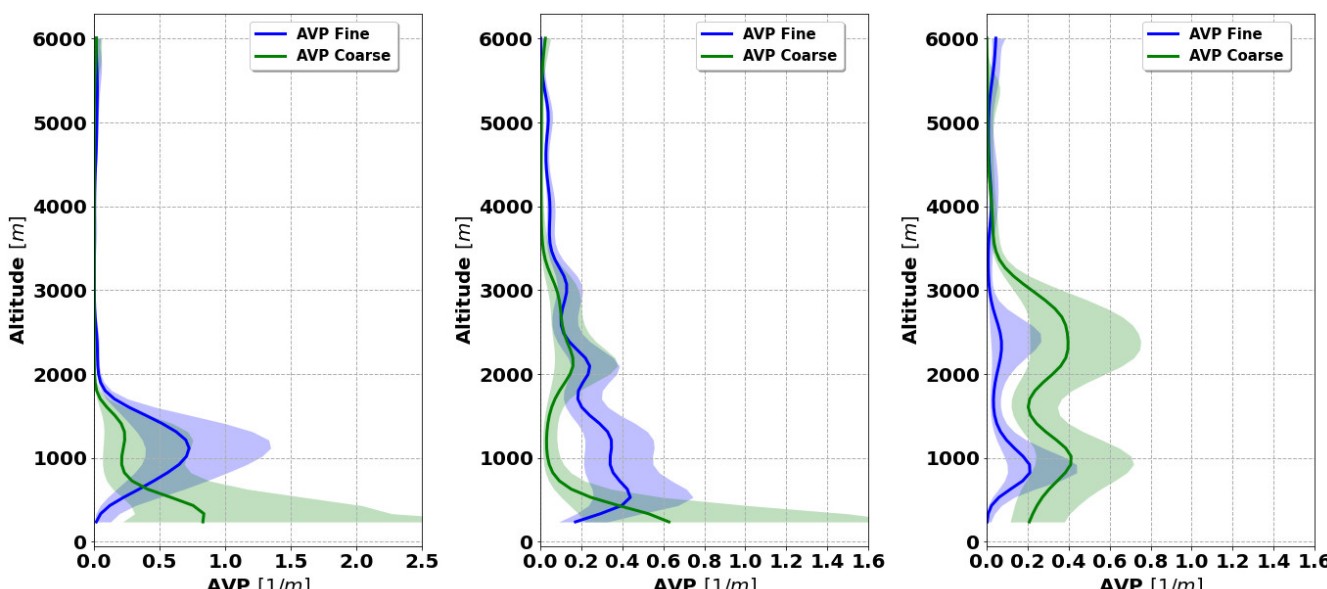

**Figure 39.** Retrieved aerosol vertical profiles (AVP) by GRASP from a combination of sun/sky photometer and lidar data. Blue solid line represents the AVP fine mode and green the AVP coarse mode. The shaded areas correspond to the total error provided by GRASP.





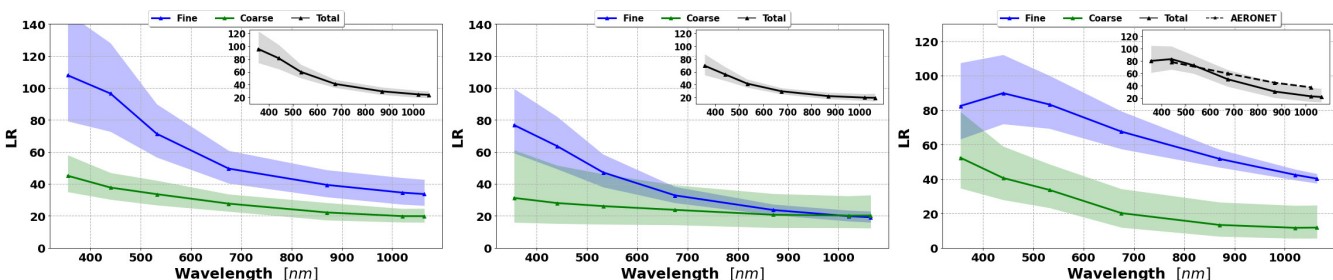

**Figure 40.** Retrieved LR by GRASP from a combination of sun/sky photometer and lidar data. Blue solid line represents the LR fine mode and green the LR coarse mode. Zommed plots show: in black solid lines the LR provided by GRASP and in dashed line the LR provided by AERONET. Their GRASP associated errors are represented in shaded areas.