# Peer review of "Estimates of the remote sensing retrieval errors by GRASP algorithm: application to ground-based observations, concept and validation"

_Atmospheric Measurement Techniques, 2022_

## Author Comment (AC1)

Dear Reviewer #1,

First of all, we would like to thank you for the positive and very interesting comments that certainly are very helpful to improve the manuscript. Please, find the detailed answers below. The original text of your comments is highlighted in blue color.

Manuscript discusses estimation of errors of aerosol and surface parameters retrieved from remote sensing observation by using covariance matrix of retrieved parameters. This approach is an alternative to more commonly used sensitivity analysis approach which is based on perturbing synthetic measurements and radiative transfer model parameters by assumed biases and then inverting perturbed measurements to observe deviation of retrieved parameters from those used in generation of synthetic observation. The advantage of sensitivity approach is its intuitive simplicity and clarity. Its disadvantage, however, is a long time of error estimation which prevents using of this approach in real time. From other hand, covariance matrix of retrieved parameters is calculated after each inversion which allows error estimation in real time. The diagonal elements of covariance matrix provide standard deviation of retrieved parameters and non-diagonal elements are proportional to correlation coefficients between them. In spite of their rare use, the non-diagonal elements of covariance matrix have a valuable information about retrieval tendencies which can be used for better understanding and improving of information content of measurements.

The main focus of the manuscript is development, testing, validation and applying covariance matrix approach to real observations. Also, manuscript includes the first to my knowledge, analysis of full covariance matrix including non-diagonal elements. The manuscript describes in detail the inversion procedure and error estimated used by GRASP algorithm. It also presents analysis of performance of the approach by example of two types of ground-based measurements: sun photometer measurements of aerosol optical depth and sky radiances and synergy of sun photometer observation and multi wavelength lidar. This is very detailed analysis done by numerical tests using synthetic measurements and perturbing them by random noise and biases. After numerical tests, approach is applied to real observations combination of sun photometer and lidar at Aeroparque and Villa Martelli stations in Buenos Aires, Argentina. Finally, the manuscript describes GRASP error estimates applied to POLDER/GRASP retrievals. Each case considered to illustrate the approach performance is supplemented by analysis of full covariance matrix which allows fast and efficient analysis of information content of the given set of observations.

I believe that the subject of the manuscript is in scope of AMT. It can be published after the following comments will be addressed.

Major comments.

1. **My major comment is related to using linear approximation in errors estimation because forward model can be highly non-linear in vicinity of solution. This issue is discussed in the manuscript and named as a reason for overestimation of errors by 20 to 30%. I wandering whether any specific research was done to better understand the effect of non-linearity of forward model on error estimate? For example, how variation of increment in calculating derivatives can help diminish the effect of non-linearity? I do not require this research to be added to the current version of the manuscript but recommend doing this analysis in the future research. It would be good adding a couple of sentences outlying doing this analysis in perspective.**

Answering to this comment, we would like to note that the numerical tests in synthetic noise were done with the purpose to check how the developed error estimates agree with "actual" errors obtained in the numerical experiment. The results of these tests showed rather good agreement with ~20-30% of overestimation. There could be many different reasons for this result, but we consider that the use of linear approximation is likely the main reason causing this effect. At the same time, the more complex non-linear formalism for error estimation is practically non-existent. Also, we consider that the overestimation can be considered as high boundary for expected errors that is acceptable for our applications since purely numerical experiments tend to provide more accurate retrieval results than it usually seen in practice. Therefore, we concluded that the developed error estimates are satisfactory for our purposes.

Regarding, sensitivity of our error estimates to different settings of the retrieval, specifically, to value of increment in calculating derivatives, we have done some tests and didn't find any clear tendency. Nonetheless, we agree that some sensitivity exists it is possible that our tests were too limited and we plan to do some analysis of this aspect in future.

2. **On page 11 it is stated "Indeed, this optimization makes the iterations converge from given initial guess to git the data even if the basic linear system is singular. Therefore, ones Levenberg-Marquart optimization is used there is an evident dependence on the initial guess that can bias the solution". This is true if not a priori information is used. Underdetermined linear system has multiple solution and minimized quadratic form has wide maximum. In this case there is dependence on initial guess. However, the goal of adding a priori information is to improve condition of linear system for it not being singular. In this case the dependence on initial guess will decrease or didapper. Is this somehow accounted for in including dependence on initial guess in error estimates?**

We fully, agree that a priori information is used in order to make solution unique and if it is added fully adequately no dependence on initial guess should be observed. At the same time, in practice we can state that retrievals when state vector includes large number of unknowns such dependence is often appears in some extent. Moreover, if the retrieval is not optimally set, such dependence can be rather significant while unnoticed because the retrieval continues to converge to local minima thanks to use of Levenberg-Marquart optimization. Therefore, in order to account for such an effect, we have added the Levenberg-Marquardt contribution into the formalism. According to our evaluation, this term is nearly negligible if there is no dependence on initial guess, while increases if such dependence appears. In order to make this

consideration clearer for the reader we have added additional explanations in the revised manuscript on the p. 12.

3. **On page 21 it is stated "In this regard, while the retrieval of multi-component is not a part of the standard AERONET inversion, GRASP algorithm allows the retrieval of several aerosol components from dives remote sensing observations including the case of aerosol retrieval from radiometer measurements only". As far as I know, AERONET retrieval code has an option to retrieve several aerosol components. In particular, at the start of the AERONET project the standard product included aerosol parameters for bi-component mixture. Later it was decided to retrieve only one component. Therefore, I always was under impression that information content of sun photometer observation is not sufficient to separate different aerosol types in external mixture. Mostly because of possible correlation between aerosol parameters of different components. Could you please explain, how this separation is achieved in GRASP algorithm? At what conditions or using some additional constraints?**

We are fully agreeing that retrieval of multi-component aerosol from AERONET observation is very uncertain. At the same, some sensitivity to presence of multi-component aerosols exists, especially in some particular situations when two aerosol components have comparable influence on AERONET measurements (the situation used in our tests). Therefore, the possibility of multi-component retrieval and distinguishing the two components is often asked by aerosol community in spite of the earlier demonstration (e.g. in Dubovik et al., 2000). Therefore, we have chosen such approach for illustrative purposes. It seemed a very good case to demonstrate that if constraints are not sufficient the errors can be unacceptably high and correlated. Also, it was a very good case to show that when some estimates are highly uncertain and strongly correlated they still can be used for accurate estimation of their functions. For example, we showed that such property as total SSA of mixed aerosol can be rather accurately obtained from retrieved SSA of fine and coarse modes. We have also illustrated the improvements in multi-component retrieval and error reduction when is included extra information, such as lidar measurements. That retrieval is often used when co-located AERONET and lidar data are inverted together (Lopatin et al., 2013, 2021)

By the way, in the relation to the discussion above, it can be noticed that the retrieval of multi-component aerosol from AERONET is often non-unique and therefore solution depends on initial guess and the L-M. term added in error estimate is useful.

We believe that this is very nice question that can be asked by many readers, therefore we also added a paragraph discussing this aspect, page 22 in the revised manuscript.

Monir comments.

1. **In Eq (30), what exactly bias proxy set means? Is it set of assumed/modeled biases?**

Yes. This means proxy of possible biases. They can come from the knowledge of the measurements, for example, due to calibration uncertainty in AERONET AOD that known at the level of, ~0.01, the bias in AOD can be + or – 0.01.

2. **In EQ. (35), I wandering would it be more correct averaging standard deviations corresponding to + and – biases of the same type prior to the averaging over contribution from different sources of uncertainties? In this case the multiplier in front of sum would be 2N.**

In the Eq. (35), N is equal 2 because N represents, the systematic component after the addition of both, positive and negative bias.

---

## Author Comment (AC2)

Dear Dr. Xu,

First of all, we would like to thank you for the positive and very interesting comments that certainly are very helpful to improve the manuscript. Please, find the detailed answers below. The original text of your comments is highlighted in blue color.

By performing truth-in-truth-out simulations, the paper by Herrera et al. did a careful work on assessing GRASP retrieval errors. The topic of study is important and technical routine was well presented. Following an introduction of the error estimate model, examples of error estimate for ground based observation including AERONET and elastic lidar were demonstrated through 40 figures and in-depth analysws in the paper. The main conclusion matches their numerical results and technical approach is well displayed and explained. I have the following comments for the authors to consider:

**1. The imposition of a priori constraints (e.g. the smoothness constraints) in GRASP algorithm can very effectively mitigate the subjection of optimization towards locally optimized solutions. However, such a possibility of getting locally optimized solutions still exist when the parameter space gets large and/or the measurement uncertainties increase. In addition, the some modeling assumption of aerosol microphysics (e.g. type, shape, aerosol size components, …) and atmosphere structure (e.g. plane-parallel atmosphere) behind the retrieval can bring in certain levels of errors as well. If I understand correctly the authors' paper, these error sources are not attempted to be analzyed in the present work. So the authors might want add some caveat information to this regard either in the introduction or sumary of the paper.**

We fully agree that there many retrieval assumptions as those on aerosol microphysics, atmosphere structure and used a priori constrains, etc. affect the accuracy of retrieval results. This is common situation in all retrieval algorithms. At the same time, those uncertainties belong to so-called systematic errors that do not have random character. Commonly, in the retrieval approaches aimed to provide statistically optimum solution, the systematic errors are considered to be negligible and the optimization is realized only for random error component. This is a case for GRASP algorithm as it was stated in this and other key GRASP paper. Certainly, one can never assure with 100% certainty that all systematic errors are truly negligible. Therefore, usually the used forward model is to be tested extensively. In these regards, our studies are based on the results of quite large number previous studies including Dubovik et al. (2000, 2002, 2006), Dubovik and King (2000), Sinyuk et al. (2007, 2020), Torres et al. (2017), etc. All those studies are resulted in the fact that all main biases were addressed rather adequately (e.g., non-sphericity) and quantified. Therefore, at present we can state that there is a rather solid certainty that there no clear significant systematic errors in forward model of GRASP and AERONET. There is also clear understanding of inevitable presence of operational systematic biases as those related calibration of the instruments. The values of these uncertainties are well known on the quantitative level and, therefore, the effect of these bias is analyzed extensively in the paper. Finally, we realize that some unexpected biases can inevitably appear in the retrieval and there is no guaranteed approach for detecting all biases, except the fact that some biases (and especially strong ones) can manifest themselves via misfit of measurements or misfit of a priori constraints. Therefore, the value of misfit (that is also includes effect of random errors) is rigorously incorporated in the used methodology for estimating the dynamic uncertainties.

**2. As illustrated via the box plots (Figs.6 and 13-19), the comparison of GRASP estimate errors of various aerosol properties against the actual errors indicated general agreement, which is impressive ! Interestingly, there are some systematic differences. It is quite commendable that the authors are looking into the impact of correlation of retrieval errors. I'm curious whether there are other possible causes behind these systematic differences. For example: the authors made an interesting observation - "The results of the statistical tests with randomly generated noise showed that GRASP error estimates in most cases are comparable or exceed the actual errors by the 20 to 30% and therefore can be safely used for assuring uncertainties of actual retrieval products." Could the overestimate of the error due to the chance of double counting the effect of random error in both "$C_{\delta a\_ran}$" and "$a_{bias}*a_{bias}$" terms via Eq.(22) ?**

Actually, there is no "double counting" of random errors in Eq.(22), because the second term is zero in case if no systematic errors present. The most probable reason for overestimating the actual errors is the fact that the used equations are defined in linear approximation that tends overestimate the actual errors if inverted functions are strongly non-linear. This is especially true in a case when one retrieved a very large number of parameters, in such situation the actual covariance matrices practically can't not be explicitly derived (in fact, the covariance matrices can be estimated but it is rather tedious and challenging task) and sophisticated solvers are used that not fully transparent, i.e. full control of all uncertainties is hardly possible in practice. It also can be noted, the "actual errors" produced in our studies are generated using solvers that may in some situation diminish the error in synthetic experiments. This is why, we consider our results as satisfactory at present, while we certainly plan to continue the efforts in improving them.

**3. As described by the authors in Table 1, the authors assume almucantar geometry in their simulation. I wonder whether there is any dependence of the accuracy of GRASP error estimate model on other observation geometries (e.g. principal-plane scan) ?**

In current paper only measurement in the almucantar geometry were analyzed because these are the most popular measurements and the analysis of only these geometries is already quite extensive. Nonetheless, the applicability of the error estimate formalism has general character and the developed formalism should be equally applicable for PPL measurements. On the basis of known experiences, it is possible to expect in general the measurements errors may be a bit higher for retrieved aerosol parameters from PPL measurements. Indeed, based on the result of a detailed comparative analysis of PPL and ALM retrievals performed by Torres et al. (2014) some differences in error tendencies can be expected. For example, the PPL geometry is generally more sensitive to measurements errors (e.g., influence of pointing errors) and also to retrieval assumptions (as aerosol vertical distribution).

**4. The abstract and the conclusion of the paper may be shortened by summarizing the main work and finding. For example, the first two paragraphs can be moved to the introduction part of the paper.**

In principle we agree that the abstract is somewhat long. However, after making some unsuccessful efforts to reduce it following you suggestion, we have decided to keep the abstract as it is. The main rational for that decision was related to fact that the paper itself is rather

long, and shortening the abstract would make it more difficult for the readers to get sufficient intro to all aspects considered in the paper

---

## Author Comment (AC3)

Dear Dr. Veselovskii,

First of all, we would like to thank you for the positive and very interesting comments that certainly are very helpful to improve the manuscript. Please, find the detailed answers below. The original text of your comments is highlighted in blue color.

Authors provide very detailed description of application of GRASP algorithm to the Sun photometer data inversion, and to the combination of this instrument with multiwavelength lidar as well. The main goal of this study is estimation of uncertainties of inversion, which is definitely very important scientific task. The manuscript is very well written, though is rather long with large amount of illustrations. On another hand, it provides the reader with all necessary information to understand the inversion technique and expected retrieval uncertainties. So, I think such length is acceptable.

I general, I think that this is very deep research, which is suitable for publishing in AMT. I have just several technical comments, concerning combining the lidar and Sun photometer.

We thank for the very positive and encouraging evaluation of our paper.

Regarding the questions, we have concluded that all questions below related to Section 3.3 are caused by the lack of introductory explanations for the considered retrievals from the combined co-incident measurement by Sun/sky-radiometer and lidar. In fact, this section considered the approach developed earlier by Lopatin et al. (2013) and widely used for joint processing of AERONET radiometer and lidar data. In frame of this approach the aerosol is modelled as external mixture of two components. These components are characterized by height independent microphysical properties in including size distribution (represented by several size bins) and spectrally dependent complex refractive index. Also, each component is described by the detailed vertical profile of volume concentration. Therefore, the retrieval provides height independent columnar properties of each component (size distribution and complex refractive index) and two profiles of fine and coarse mode volume concentrations. It is expected that this model is sufficient to adequately describe both radiometric and lidar observations.

The above explanations are added in the introduction of Section 3.3 of the revised manuscript (p. 19).

1) **Section 3.3. It would be useful to provide modal radii of the fine and coarse modes for aerosol types used in the model. Modal radii depend on the relative humidity and may change with height. Can it influence the uncertainty?**

We have added more information about the table containing the parameters of modelled size distributions and complex refractive indices in the text of the revised paper. Regarding the effects of the humidity and hygroscopic growth, we fully agree that this presents in the reality and may result in the vertical variability of both size and refractive index. Our model doesn't allow such variability for properties of each mode. This is certainly a limitation. At the same time, we expect that the model is sufficient for adequate mimicking of all lidar measured parameters such high and spectrally dependent extinction, lidar ratios, etc. Certainly, this concept will evolve in future, but for now our studies were based on this concept, especially is widely used by lidar communities such as ACTRIS.

**2) Fig.9. I am a bit confused. What is height distribution of particle concentration used in this modeling? For what height the results are shown? Does uncertainty depend on height?**

As explained above, the approach assumes that the microphysical properties of each mode (size distribution, complex refractive index and shape) are height independent, while vertical profiles of concentrations vary with altitude as is shown in Figure 12. Correspondingly, Figure 9 and 11 show only the column integrated properties while Figure 12 illustrates the vertical profile concentration which vary with the altitude for the two modes, fine and coarse mode.

**3) Fig.11. The height distributions for dust and smoke are the same? In real situation these are always different, so would be good to discuss how it will influence the modeling. Again, for what height lidar ratios are shown?**

Once again, Fig. 11 shows the column LR for fine and coarse modes independently that are assumed vertically constant. The height distribution for fine and coarse mode are not the same as they are illustrated in the Figure 12. The fine mode represents the background aerosol with specific vertical distribution, while coarse mode distribution had a thick layer approximately at 2.5 km. As a result, the total lidar ratio certainly changes vertically.

**4) Fig.12. The results are shown for the mixture of two components. Any chance to retrieve the profiles of two components separately?**

The Figures 9, 11 and 12 were simultaneously retrieved and they represent the columnar properties for two aerosol modes (Fig. 9 and 11) and the aerosol vertical profiles of each modes (Fig. 12). In the tests, we generated sun/sky photometer and lidar measurements and we retrieved simultaneously different aerosol properties, such us, SD, RRI, IRI, sphericity fraction and also the aerosol vertical distribution for both, fine and coarse mode separately (once again, these properties of each mode are assumed vertically constant). In addition, we provided the derived parameters, SSA and LR, for fine and coarse mode. With no doubts, total SSA and LR are strongly vertically dependent, as illustrated by the figure below for one example of the real case (Fig. 38) where we can clearly identify this dependency.

[Figure]

Figure: The vertical profiles of SSA and lidar ratios for total (fine +
coarse modes) aerosol for the case illustrated by Fig. 38.